# Glioblastoma Therapy: Past, Present and Future

**DOI:** 10.3390/ijms25052529

**Published:** 2024-02-21

**Authors:** Elena Obrador, Paz Moreno-Murciano, María Oriol-Caballo, Rafael López-Blanch, Begoña Pineda, Julia Lara Gutiérrez-Arroyo, Alba Loras, Luis G. Gonzalez-Bonet, Conrado Martinez-Cadenas, José M. Estrela, María Ángeles Marqués-Torrejón

**Affiliations:** 1Scientia BioTech S.L., 46002 Valencia, Spain; paz.moreno72@gmail.com (P.M.-M.); Maria.Oriol@uv.es (M.O.-C.); loblanch@alumni.uv.es (R.L.-B.); jose.m.estrela@uv.es (J.M.E.); 2Department of Physiology, Faculty of Medicine and Odontology, University of Valencia, 46010 Valencia, Spain; begona.pineda@uv.es; 3Department of Medicine, Jaume I University of Castellon, 12071 Castellon, Spain; gutierrj@uji.es (J.L.G.-A.); loras@uji.es (A.L.); ccadenas@uji.es (C.M.-C.); 4Department of Neurosurgery, Castellon General University Hospital, 12004 Castellon, Spain; lbonet@uji.es; 5Department of Physiology, Faculty of Pharmacy, University of Valencia, 46100 Burjassot, Spain

**Keywords:** glioblastoma, targeted therapy, immunotherapy, nanotherapy, non-ionizing radiation

## Abstract

Glioblastoma (GB) stands out as the most prevalent and lethal form of brain cancer. Although great efforts have been made by clinicians and researchers, no significant improvement in survival has been achieved since the Stupp protocol became the standard of care (SOC) in 2005. Despite multimodality treatments, recurrence is almost universal with survival rates under 2 years after diagnosis. Here, we discuss the recent progress in our understanding of GB pathophysiology, in particular, the importance of glioma stem cells (GSCs), the tumor microenvironment conditions, and epigenetic mechanisms involved in GB growth, aggressiveness and recurrence. The discussion on therapeutic strategies first covers the SOC treatment and targeted therapies that have been shown to interfere with different signaling pathways (pRB/CDK4/RB1/P16^ink4^, TP53/MDM2/P14^arf^, PI3k/Akt-PTEN, RAS/RAF/MEK, PARP) involved in GB tumorigenesis, pathophysiology, and treatment resistance acquisition. Below, we analyze several immunotherapeutic approaches (i.e., checkpoint inhibitors, vaccines, CAR-modified NK or T cells, oncolytic virotherapy) that have been used in an attempt to enhance the immune response against GB, and thereby avoid recidivism or increase survival of GB patients. Finally, we present treatment attempts made using nanotherapies (nanometric structures having active anti-GB agents such as antibodies, chemotherapeutic/anti-angiogenic drugs or sensitizers, radionuclides, and molecules that target GB cellular receptors or open the blood–brain barrier) and non-ionizing energies (laser interstitial thermal therapy, high/low intensity focused ultrasounds, photodynamic/sonodynamic therapies and electroporation). The aim of this review is to discuss the advances and limitations of the current therapies and to present novel approaches that are under development or following clinical trials.

## 1. Introduction

### 1.1. The Nature and Prognosis of Glioblastoma

Glioblastoma (GB), categorized as a grade IV astrocytoma, is the most prevalent, aggressive, and lethal primary brain tumor in adults. In 2021, the World Health Organization (WHO) introduced significant changes in the criteria for the diagnosis of gliomas, focusing on the importance of genetic and molecular alterations. According to these new criteria (Figure 1), GB should be diagnosed in adults as an isocitrate dehydrogenase wild-type (IDHwt) diffuse astrocytic glioma if there is microvascular proliferation or necrosis (the conventional criteria), and/or at least one of the following three criteria: concurrent gain of whole chromosome 7 and loss of whole chromosome 10 (+7/−10), telomerase reverse transcriptase (TERT) promoter mutations, and epidermal growth factor receptor (EGFR) amplification [1,2]. The primary driver behind the change in diagnosis criteria is the IDH mutation status, which results in the following modifications: restricting the diagnosis of GB to tumors that do not have IDH mutations (IDHwt); reclassifying tumors previously identified as IDH-mutated GBs as astrocytomas with IDH mutations (grade IV); and establishing the presence of IDH mutations as a requirement for classifying tumors as astrocytomas or oligodendrogliomas Consequently, due to its more favorable prognosis, the previously designated IDH-mutant GB is now categorized within the astrocytomas group, which covers grades II–IV, thus eliminating the term IDH-mutant GB [3]. Moreover, in IDHwt diffuse astrocytomas occurring in younger people, diagnostic consideration should be given to the different types of diffuse pediatric-type gliomas [1]. Gliosarcoma, epithelioid cell GB and giant cell GB are still registered subtypes of GBs, and the term “glioblastoma multiforme” should not be used [4,5]. New clinical trials will need to be designed with these new distinctions in mind [6].

GB typically appears in the cerebral hemispheres, with 95% of these tumors arising in the supratentorial region, especially in the frontal and temporal globes. It infiltrates inside the brain parenchyma and usually does not spread to other parts of the body [7,8,9]. Histologically, GBs are characterized by prominent cellular and nuclear atypia, increased mitotic activity, areas of necrosis, and microvascular proliferation. At least one of these two features must be present for a histologic diagnosis of GB [5]. GB causes death in less than 6 months if untreated [10]. Despite advances in neurosurgery, chemotherapy and radiotherapy, GB remains one of the most treatment-resistant malignancies and its relapse is, in practice, inevitable [7,11,12]. Recurrence often implies a more aggressive form and a median survival of less than 18 months in treated patients [13,14]. Survival beyond 5 years is observed in less than 5.8% of patients [7]. Patients with recurrent GB (rGB) show an approx. 6-month progression-free survival (PFS6) in only 15% of the cases, and overall survival (OS) ranging between 24 and 40 weeks. Survival rate decreases with age [11,15].

As suggested by the moniker “multiforme”, GB has a widespread tumoral heterogeneity and plasticity at the cytopathological, transcriptional, and genomic levels [16,17,18,19,20,21]. Moreover, its highly infiltrative nature and the protection by the blood–brain barrier (BBB) have posed significant treatment challenges [9,22,23]. Glioma stem cells (GSCs) are a small subpopulation of cells within the GB, with genomic instability, self-renewal and tumor-initiating capacity, and the ability to differentiate into different GB subpopulations, being responsible for tumor heterogeneity [24,25,26,27]. Moreover, GSCs are resistant to apoptosis [28,29,30,31], can modulate the components of the tumor microenvironment (TME), are involved in angiogenesis activation and immunosuppression and drive radio/chemoresistance [22,32,33]. The inability of current therapies to eliminate specific GSC subpopulations has been considered a major factor contributing to the inevitable recurrence after treatment [33,34].

Verhaak et al. proposed a four-subtype classification of GB (classical, mesenchymal, proneural, and neural) based on the analysis of mutational changes in 601 genes in the context of The Cancer Genome Atlas (TCGA) [35]. Verhaak’s latest update removed the neural subtype attributing its origin to a peripheral contamination of the tumor samples [20]. The proneural subtype is associated with a more favorable outcome with respect to the mesenchymal, but this difference is relative to the more favorable outcome of IDH-mutant GBs which were consistently classified as proneural GBs [20,36]. Moreover, mesenchymal GSCs are enriched with genes associated with angiogenesis, inflammation, and cell migration/invasion. They tend to develop immunosuppression and exhibit increased radio/chemoresistance, all of which are features linked to a worse prognosis [36,37,38,39]. In any of the cases, the survival difference is minimal because both subtypes can coexist in the same tumor, and dynamic transitions from a proneural to a mesenchymal phenotype can be induced by TNF-α, temozolomide (TMZ), or radiation through an NF-κB-dependent mechanism [20,36,40,41]. Meanwhile, multiplatform analyses of the genetic, epigenetic, and transcriptional profiles have proven useful in refining the classification of gliomas and predicting patient outcomes [42,43,44,45].

### 1.2. Incidence and Risk Factors

GB is the most common (50.1%) among all malignant brain tumors [15]. The annual incidence is low (≈3.19 per 100,000 people in developed countries) but seems to be increasing in some countries owing to aging populations and improvements in diagnosis, among other factors [46]. The median age of diagnosis is approx. 64 and the incidence increases with age reaching its maximum value (15 per 100,000 people) between 75 and 84 years [11,15,47]. It is extremely rare in a pediatric population (0.15 per 100,000), which usually shows longer survivals. The occurrence of GB is 1.6 times more common in males than females and in Caucasians relative to other ethnicities [15,48]. 

Beyond rare cases of genetic susceptibility and high-dose radiation exposure, there are no known GB risk factors. An increased risk is seen in some specific genetic diseases, such as hereditary retinoblastoma or Cowden, Turcot, Lynch, Li-Fraumeni and Maffucci syndromes. However, less than 1% of GB patients have a known hereditary disease. Radiation-induced GB can be diagnosed several years after radiation therapy for another tumor or condition in children [49], but no increased risk was observed in adults exposed to IR [50]. Patients diagnosed with previous non-neurological cancers may have an overall elevated incidence of GB compared to the general population [51].

### 1.3. Criteria to Evaluate Treatment Response and Progression 

The MacDonald criteria [52] have traditionally been used to determine treatment response and progression by assessing contrast-enhancing tumor size [by computed tomography (CT) or magnetic resonance imaging (MRI)] along with clinical evaluation and corticosteroid dosage. These criteria categorized the response into four groups: complete response (CR), partial response (PR), stable disease (SD), and progressive disease (PD). Nevertheless, these criteria have several limitations. One is the temporary increase in tumor enhancement (known as pseudo-progression), which occurs in 20–30% of the patients treated with chemo/radiotherapy and challenges differentiation with a real tumor progression. Another limitation is the high radiographic response rates seen with anti-angiogenic agents and other treatments [53]. To address these issues in 2010, the Response Assessment in Neuro-Oncology (RANO) criteria were developed to address these limitations. Although the RANO criteria improved therapy evaluation in high-grade glioma, the assessment of treatment-related side effects can hinder accurate response evaluation. The appearance of new lesions is considered a criterion for disease progression according to both RANO and MacDonald criteria. However, neuro-oncology patients receiving immunotherapies may experience the transient appearance of new enhancing lesions, either locally or in distant sites. In such cases, it is advisable to evaluate imaging findings within 6 months of starting immunotherapy, including the development of new lesions or radiographic progression, as long as there is no significant clinical deterioration [54].

To tackle the challenges in assessing immunotherapy response for neuro-oncology, the immunotherapy RANO (iRANO) criteria were introduced. The iRANO criteria combine the response assessment framework of RANO with guidelines for confirming disease progression, as originally proposed by the Immunotherapy Response Criteria in Solid Tumors to assist in clinical decision-making. The aim is to minimize premature discontinuation of potentially beneficial therapies while ensuring patient safety [54]. However, in most recurrence cases, there is a mixture of tumor cells and tissue affected by radiation injury. Radiologists strive to identify the predominant component of the lesion to determine prognostic factors and categorize the findings according to the RANO criteria, thus providing the most appropriate treatment for the patient. To overcome the aforementioned limitations in the follow-up, incorporating changes measured by advanced MRI and positron emission tomography (PET) imaging, which may precede anatomical changes in tumor volume, shows promise [55,56]. PET may also help to differentiate actual progression from pseudo progression [57]. Additionally, ^18^F-FMISO-PET can localize regions of hypoxia that are thought to drive radio/chemoresistance in GBs and promote immune suppression [58].

## 2. Lessons Learned in the Pathophysiology of Glioblastoma

### 2.1. Glioma Stem Cells and Tumor Microenvironment

The two prevailing hypotheses for the origin of GB are the GSC and the astrocyte de-differentiation theories [25,26,37]. Neural stem cells (NSCs), as unique stem cell type in the brain, have the ability to self-renew and can differentiate into neurons, astrocytes, and oligodendrocytes (Figure 2) [25,59]. NSCs are most active during development, but small populations remain functional in specific stem-cell niches in the adult brain. Compelling evidence suggests that GSCs may arise from NSCs located in the adult subventricular zone (SVZ) [60,61,62,63,64,65], and a recent article provided molecular genetic confirmation of this hypothesis in a preclinical model [60]. GSCs express the mutated genes TERT, PTEN, EGFR, TP53, and PDGF present in NSCs. In addition, there is an evident functional overlap and similarity between both types of stem cells, reflected in numerous shared gene expression patterns such as CD133, Sox10, nestin, vimentin, musashi, GFAP, and Olig1/2 [65,66,67]. Due to the migration ability of GSCs and the unique environment of SVZ (the vascular system of SVZ is richer than that of other brain regions), treatment-resistant GSCs easily migrate to and colonize the SVZ [68]. Consequently, numerous retrospective studies have confirmed that GBs in close contact with the SVZ possess more aggressive patterns of recurrence and worse clinical outcomes [67,69,70]. Therefore, new therapy strategies are being assayed with the aim of targeting SVZ to eradicate NSCs or GSCs [71].

The origin of GB, based on the stem cell theory, explains the versatility and plasticity of heterogeneous GB tumor populations. However, several studies provide evidence suggesting that partially differentiated glial cells, including oligodendrocyte and astrocyte precursors, may play a role in or be responsible for tumorigenesis [60,72,73]. The astrocyte de-differentiation theory is supported by experiments demonstrating the formation of tumors that are histologically similar to GB after activation of oncogenes and/or suppression of tumor suppressor genes in astrocytes [24,72,74,75]. Nevertheless, this manipulation in astrocytes results in their acquisition of stem-cell-like characteristics. Consequently, both hypotheses are not mutually exclusive and explain the presence of cancer stem cells within the tumor [64,76,77]. Moreover, the dedifferentiation of non-GSCs to GSCs further complicates the GSC-targeted therapy [25,78].

GSCs represent a very low percentage of cells within GBs, and are functionally defined and distinguished from their differentiated tumor progeny at central transcriptional, epigenetic, and metabolic regulatory levels [79,80]. Recognized markers of GSCs include CD133 (PROM1) [81], CD44 [82], SOX2 and nestin [76,83], but none of them are specific markers of GSC. Other putative biomarkers are CD15 (FUT4), A2B5 antigen, CD90 (THY1), integrin ITGA6, CD171 (L1CAM), S100A4, ATP-binding cassette transporters and the combination of CD44 and ID1 (reviewed in [84]). GSCs develop genetic variability and possess self-renewal capacity and specific characteristics that support tumor development, heterogeneity, recurrence, immunosuppression and radio and chemotherapeutic resistance [85]. Therefore, the heterogeneity of GB tumor cells can be attributed to the clonal evolution and differentiation/dedifferentiation capacity of GSC [25,66,72,77,86,87]. The GSCs’ ability to adapt to different niches implies that they can dynamically restructure their transcriptional program, inducing the transient expression of genes with specific functions for each cell state [18,36]. Furthermore, microglia and endothelial cells of the perivascular niche produce numerous growth factors that contribute to the support of proliferation, migration, and differentiation of NSCs and GSCs [27,88,89]. In turn, GSCs release transforming growth factor β (TGFβ) that enhances the tumor vasculature and can even transdifferentiate and generate endothelial cells or pericytes to form new tumor vascular niches [90,91]. GSCs exhibit elevated migratory and invasive potential, eliciting infiltration into healthy tissue, thus limiting the effect of total surgical resection and radiotherapy [92]. Residual cells have the ability to regenerate GB in brain regions distant from the initial tumor by acquiring new and different driver mutations that make them resistant to treatments [41]. Consequently, GSCs are more radioresistant than GB cells [93], can be resilient to TMZ-mediated cell death [94], and have mutations that facilitate recurrence after therapy [95]. DNA damage repair mechanisms, such as ATM, ATR, CHK1, and PARP1, are upregulated in GSCs, and CHK1 is preferentially activated following irradiation [96,97].Consequently, GSCs exhibit rapid G2-M cell cycle checkpoint activation and enhanced DNA repair [98]. The preferential activation of DNA damage checkpoint responses [34] and the increased expression of drug efflux pumps and antiapoptotic proteins [99] contribute to GSC recruitment after treatment. Interestingly, the inhibition of DNA repair protein RAD51 homolog 1 has been found to delay G2 cell cycle arrest, thereby sensitizing GSCs to radiation [100].

Ionizing radiation also enhances the motility, invasiveness and aggressiveness of GSCs. The increased motility and invasiveness result from the activation of the HIF(hypoxia-inducible factor)-1α, whereas aggressiveness is attributable to a pro-neural-to-mesenchymal transition associated with the activation of the STAT3 transcriptional factor [101]. STAT3 is overexpressed in GSCs [87] and plays a crucial role in sustaining stem-like characteristics [102]. Moreover, it enhances the expression of pro-tumorigenic genes related to cell cycle progression, extracellular matrix remodeling, as well as the secretion of cytokines and growth factors [103]. Consequently, STAT3 deletion or inhibition in GB cell lines markedly decreases tumorigeneses in vitro and in vivo [103,104] and has a radiosensitizing effect [93]. WP1066, one of the most promising STAT3 inhibitors, will be investigated in a phase II clinical trial for patients with recurrent malignant glioma [105].

GB cells have the ability to manipulate the TME to favor immunosuppression and to develop a niche sustaining tumor growth, invasion, migration, and survival [28,106]. GB cells can evade immune surveillance through the release of various soluble mediators such as TGFβ, IL-10, and PGE-2. In the presence of TGFβ, CD4+ T cells upregulate FoxP3 and differentiate into Treg cells with potent immunosuppressive potential. This cytokine inhibits the expression of cytolytic gene products (perforin, granzyme A, granzyme B, Fas ligand, and IFN-γ) which are co-responsible for CD8+ T-cell-mediated tumor cytotoxicity. Increased secretion of IL-10 is associated with enhanced expression of anti-inflammatory cytokines, such as IL-4, CCL2, and TGFβ. In the presence of IL-10, TAMs downregulate the expression of antigen-presenting molecules, thereby impairing CD4+ T cell activation. In turn, PGE-2 has been shown as a key mediator of immunosuppressive activity through the expansion of myeloid-derived suppressor cells (MDSCs) [107]. In fact, GSCs and GB cells play the role in recruiting and activating MDSCs [108] and M2 macrophages to drive immune suppression [109,110,111]. Simultaneously, GSCs protect themselves from T-cell-mediated killing by secreting extracellular vesicles containing programmed death ligand 1 (PD-L1) [112,113]. 

Consequently, new therapies that effectively target this important population may help to prevent recurrence and improve patient survival, and for sure, no single therapeutic modality will be effective against such a heterogeneous population of cells.

### 2.2. Metabolic Features Favoring Growth and Resistance

Metabolic reprogramming plays a crucial role in enabling GB invasive cells to generate the energy required for colonizing the surrounding brain tissue and adapting to hypoxic microenvironments [114,115]. The metabolism of GB is characterized by the upregulation of the PI3K/Akt/mTOR signaling pathway, a high rate of glycolysis, and increased lipid storage [116,117]. Aerobic glycolysis along with glucose consumption and lactate production supports rapid GB growth and correlates with a lower survival rate [118]. Nevertheless, GB cells adapt their metabolism according to glucose availability, which gives them extra resistance to hypoxia or altered redox situations. Selective pressure on GB cells makes them overexpress glucose transporters (GLUT1 and, particularly, GLUT3). GLUT3 has a five-fold higher affinity for glucose compared to GLUT1, thus facilitating glucose uptake in environments with lower glucose concentrations. Additionally, the acquisition of a stem cell state is associated with a significant increase in GLUT3 expression in induced pluripotent cells, and this overexpression correlates with poor glioma patient survival [119,120]. When glucose levels are low, HIF-1α guarantees the upregulation of GLUT3 and hexokinase-2, increasing the glycolytic pathway [121,122]. 

The activation of sterol regulatory element-binding protein-1, a crucial transcription factor controlling fatty acid and cholesterol synthesis, as well as cholesterol uptake, enables GB to obtain significant quantities of lipids essential for its rapid growth [123]. GSCs exhibit high expression of mediators of lipid metabolism, such as brain-fatty-acid-binding protein (FABP7), which leads to an increase in lipid contents that are specifically metabolized under glucose-deprived conditions [116]. GB cells direct significant amounts of lipids into specialized storage organelles known as lipid droplets, thus avoiding lipotoxicity. This process involves the overexpression of diacylglycerol acyltransferase-1 and sterol-O-acyltransferase-1, which convert surplus fatty acids and cholesterol into triacylglycerol and cholesteryl esters, respectively, increasing the storage as neutral lipids within lipid droplets [123]. 

Amino acids play a crucial role as important fuels for GB growth. Gene expression profiling has shown an upregulation of the L-Gln importer ASCT2 in GB compared to low-grade gliomas, and L-Gln deprivation has slowed tumor growth in some in vitro studies [124]. The L-Gln-derived glutamate and glucose-derived pyruvate are substrates for the glutamate-pyruvate transaminase 2 (GPT2), which synthetizes α-ketoglutarate. Through GPT2 upregulation, the anaplerotic replenishment of the TCA cycle is possible; otherwise, it is impaired by augmented pyruvate conversion to lactate. In other words, the Warburg effect, manifested as increased lactate release, drives L-Gln addiction in order to maintain the TCA cycle function [124]. Moreover, L-Gln has been shown to promote the mTOR-dependent signaling pathway, a potent driver of GB growth and progression [125,126]. Other amino acids are also utilized to fuel bioenergetic reactions and the synthesis of macromolecules in GBs [114]. L-Asp has been shown to be a limiting metabolite for GB cellular proliferation in hypoxic conditions [127]. L-Arg is involved in GB cell adhesion, and thereby in tumor cell migration and invasion [128]. L-Trp and L-Arg metabolism have also been linked to decreased detection by neighboring immune cells, creating a favorable environment [129].

In gliomas, autocrine glutamatergic signaling has been identified as a promoter of invasion [130]. GB cells release high levels of glutamate, which not only enhances tumor invasiveness but also promotes the turnover of GSCs [131]. In other words, GB cells create a positive feedback system whereby an excess of glutamate promotes their own growth and secondarily causes excitotoxicity-induced cell death in surrounding brain tissue [132]. It is probable that such tissue damage contributes to cerebral edema and the neurotoxicity associated with a growing GB. Consequently, the inhibition of glutamatergic signaling has been proposed as a strategy to mitigate GB-induced neurotoxicity [133].

Moreover, the invasive nature of GB is modulated by cell-to-cell crosstalk within the TME and altered expression of specific genes, such as ANXA2 (encoding the protein annexin A2, a Ca^2+^-dependent phospholipid-binding protein that helps to organize exocytosis of intracellular proteins to the extracellular domain) [134], GBP2 (encoding the guanylate-binding protein 2, which binds to guanine nucleotides and works in intracellular signaling) [135], FN1 (encoding fibronectin, which binds to integrins and facilitates adhesion, growth, migration, and differentiation) [136], PHIP (encoding the pleckstrin homology domain interacting protein, which regulates growth and survival of GB cells) [137], and SLC2A3 (encoding the glucose transporter 3) [114,138]. 

### 2.3. Ion Channels

Different studies have demonstrated the upregulation of Ca^2+^ selective ion channels in GB, contributing to invasion, proliferation and resistance to apoptosis [139]. By blocking L-type voltage-gated Ca^2+^ channels, cell invasion is inhibited as filopodia (also known as tumor microtubes, TMs) formation is blocked [140]. Indeed, GB cells possess the ability to extract specific signals from healthy neurons using TMs [141]. Furthermore, inhibition of T-type Ca^2+^ channels has been shown to induce apoptosis in GB cells [142]. Therefore, blocking Ca^2+^ could prevent tumorigenesis through several mechanisms, i.e., cell cycle progression, induction of apoptosis and inhibition of cell migration. 

K^+^ ion channels play a crucial role in the proliferation and the resistance to apoptosis in GB. Specifically, certain voltage-gated K^+^ channels are overexpressed in GBs participating in signaling pathways that promote proliferation and inhibit apoptosis [143]. Some of these effects are due to the role of K^+^ channels in establishing the resting membrane potential, and therefore affecting the cell cycle. Different studies have shown that inhibition of K^+^ channels improves survival in GB patients, which emphasizes their role in GB development and progression [144]. Consequently, blocking of ion channels could represent an interesting therapeutic approach against GB progression.

### 2.4. Epigenetics of Glioblastoma

GB progression is associated with different types of epigenetic alterations, including histone modifications, DNA methylation, chromatin remodeling, and aberrant microRNA (miRNA) [145,146], a group of small non-coding RNA (19–22 nucleotide long) molecules that regulate the post-transcriptional degradation of mRNA [147]. Ciafré et al. performed the first experiment related to miRNAs in GB, investigating the expression of 245 miRNAs using microarrays [148]. The most interesting results came from miR-221 upregulation, and a set of brain-enriched miRNAs (miR-128, miR-181a, miR-181b, and miR-181c) that are down-regulated in GB. At the same time, miRNAs have been shown to be important regulators of gene expression and may also regulate cellular processes, including apoptosis, proliferation, invasion, angiogenesis, and chemoresistance [149,150]. Therefore, microRNAs can be classified according to their role in tumorigenesis (i.e., tumor suppressor or oncogenic). Table 1 summarizes what is known, at present, regarding miRNAs and their role in GB progression and resistance to therapies.

CircRNAs exert their biological effects through four different mechanisms: serving as sponges of RNA binding proteins, modulating parental gene transcription, encoding functional proteins and, most importantly, serving as sponges of miRNAs [189]. As is thoroughly reviewed in [189,190], circRNAs regulate GB proliferation and invasion and are also involved in angiogenesis activation. Good stability, broad distribution and high specificity make circRNAs promising biomarkers for GB prognosis and/or diagnosis, although their clinical implementation still has a long way to go.

Long-noncoding RNAs (lncRNAs) are a class of regulatory noncoding RNAs (>200 nt) that interact with DNA, RNA, and proteins to regulate various biological processes. As reviewed in [191,192,193,194], numerous studies have shown that lncRNA regulates the expression of genes involved in GB tumorigenesis (CHRM3-AS2, DLGAP1-AS1, DGCR10, LINC01057, LINC-PINT, MIR31HG, MIR210HG, NEAT1, NONHSAT079852.2, PVT1, SEMA3B, RBPMS-AS1), progression (ASLNC22381, ASLNC20819, CRNDE, DGCR10, HNF1A-AS1, HOXD-AS2, HRA1B, HOTAIRM1, LINC-PINT, PRADX, NEAT1, OXCT1-AS, TCONS-00004099) and therapeutic resistance (H19, MALAT1, MUF, DANCR, HOTAIR, HOTAIRM1, LINC00511, UCA1, OIP5-AS1, DANCR, FOXO3, HERC2P2) of GB cells. Moreover, lncRNAs exhibit stable secondary structure; thus, some of them (HOTAIR, GAS5, HOXA11-AS, HOTAIRM1, AGAP2-AS1, and AC002456.1) have been proposed as prognostic and diagnostic GB biomarkers [195,196,197,198]. More specifically, SBF2-AS1, MALAT1, CRNDE, TP73-AS1 and LINC00511 have been suggested as biomarkers of TMZ resistance in GB [198]. Lately, some evidence has indicated that lncRNAs also take part in GB cell metabolism. For instance, the lncRNA TP53TG1, under glucose deprivation, may promote cell proliferation and migration by influencing the expression of glucose-metabolism-related genes in glioma cells [199], and the lncRNA LEF1-AS1 facilitates the multiplication of GB cells and impedes apoptosis via the Akt/mTOR pathway [200]. Clinical trials involving the use of lncRNA as biomarkers for GB detection and prognosis are only in the recruitment phase but look promising.

Other epigenetic alterations, such as DNA methylation, histone modifications, and chromatin remodeling, are mechanisms involved in transcriptional activation of critical genes for GB development, lethality and resistance [145,201,202,203]. Thus, several epigenetic agents, including histone methyltransferase inhibitors, DNA methyltransferase inhibitors, histone deacetylase (HDAC) inhibitors, and other agents, are currently being tested for GB treatment in preclinical and clinical trials [146,203]. Protein arginine methyltransferase 5 (PRMT5) is a member of the PRMT family of proteins that plays a key role in the regulation of cellular signaling and gene expression by methylating histones as well as nonhistone proteins [204]. Nuclear expression of PRMT5 negatively correlates with glioma patient survival [205]. Engineered loss of PRMT5 or treatment with CMP5 (a PRMT5 inhibitor) results in apoptosis or loss of self-renewal for differentiated or undifferentiated GB cells, respectively, [206]. CMP5 derails the negative regulation of PTEN by PRMT5, which, in turn, decreases Akt activity in patient-derived GB neurospheres [207]. 

HDACs have been widely studied in GBM cells due to their relationship with therapeutic resistance, cell proliferation and invasion, angiogenesis and apoptosis [208,209,210,211]. In preclinical studies, HDAC inhibitors (HDACi) have proven to be effective anti-GB agents via multiple mechanisms, such as upregulating the expression of tumor suppressor genes, inhibiting oncogenes, inducing cell cycle arrest, promoting cell apoptosis and differentiation, inhibiting motility/migration, abolishing autophagy and tumor angiogenesis, and upregulating natural killer (NK)-cell-mediated tumor immunity [212,213,214,215]. Additionally, HDACis have demonstrated the capability to reduce cancer stem cell burden in GB tumors by modulating stemness, proliferation, differentiation, cell cycle arrest, apoptosis, autophagy and vasculogenic mimicry of GSCs [209,210,216]. Several HDACis (i.e., valproic acid, voristonat, panobinostat) have been assayed in clinical trials due to their capacity to act as chemo/radio-sensitizers and target GSCs [216]. Voristonat combination regimens with TMZ/radiotherapy and/or bevacizumab (BEV, recombinant humanized monoclonal antibody that blocks VEGFR-A) have proven to be tolerable (NCT01738646), but no statistical improvement in OS and/or PFS was noted [217]. Similar results were obtained with panobinostat in combination with BEV (NCT00859222) [218]. Valproic acid is a potent anticonvulsant that promotes apoptosis and impairs glioma cell proliferation and invasiveness and sensitizes GB cells to several anticancer drugs, such as TMZ, etoposide, gefitinib, nitrosoureas, and radiation therapy [219,220,221]. A meta-analysis [222] and a recent open-label phase II study [223] results seem to confirm that GB patients may experience prolonged survival due to valproic acid administration, providing further justification for a phase III trial of valproic acid/SOC. Levetiracetam (LEV), a relatively new antiepileptic drug, modulates HDAC levels ultimately silencing MGMT, thus increasing TMZ effectiveness in GCSs [224]. Retrospective analyses and an open-label phase II study (NCT02815410) seem to evidence that LEV improves GB patients’ PFS and OS [225,226]. So, it is perhaps time to reconsider the results performed in 2016, where a pooled analysis of a large series of cases treated with valproic acid or levetiracetam failed to find an association with patients’ survival [227]. A double-blind randomized clinical trial (ChiCTR2100049941) focusing on the clinical benefits of LEV + TMZ in the treatment of GB is ongoing in China. Nuclear imaging of HDAC expression in GB can be useful to improve the understanding and role of HDAC enzymes in gliomagenesis and identify patients likely to benefit from HDACi-targeted therapy [215,228].

### 2.5. The Angiogenetic Capacity of Glioblastoma

Aberrant vascular proliferation, necrosis, and infiltration of surrounding brain tissues are considered “hallmarks” of GB. Neo-vessels form from preexisting blood vessels due to VEGF expression by tumor and stromal cells under hypoxic conditions. The combination of VEGF with FGF (fibroblast growth factor)-2 or PDGF (platelet-derived growth factor) is known to synergistically enhance angiogenesis [229]. Vasculogenic mimicry (VM) is a new mechanism of tumor neovascularization in which highly invasive and genetically dysregulated tumor cells acquire vascular cell function, forming de novo vascular-like structures [230]. The involvement of GSCs in VM has been reported by several studies [231,232]. The disruption of GB vasculature through radiation or anti-angiogenic therapies induces a hypoxic microenvironment that promotes VM as an adaptative strategy to assist GB cells in surviving and progressing even when angiogenesis is blocked [233,234]. In keeping with this idea, the inhibition of vasculogenesis, but not sprouting angiogenesis, prevents the recurrence of GB after irradiation in mice [235].

SCs play a crucial role in VM, mainly due to their high plasticity and potential differentiation into endothelial-like cells [236]. The vascular niche is very important for the maintenance of GSCs as it promotes their survival and proliferation [230,237]. Additionally, communication between endothelial and tumor cells allows tumor vasculature formation and tumor cell dissemination [232,238]. Tumor vasculature has been considered a contributor to treatment resistance and relapse [239]. GSCs seem to be attached to the arterioles but not to the capillaries [240]. Arterioles transport, but do not exchange, gasses and nutrients, and promote a peri-hypoxic area. Integrin ligation causes an activation of the integrin-linked kinase leading to increased HIF-1α, as well as increased VEGF production [241]. HIF-1α acts as a potent activator of angiogenesis by stimulating the production of VEGF-A, PDGF and many other factors that initiate endothelial cell proliferation, invasion, and migration [242]. In GB, HIF-1α is not only influenced by oxygen but also by oncogenic signaling pathways, such as MAPK/ERK, p53, and PI3K/Akt/mTOR [243]. Although many approaches have been tried to inhibit HIF-1α, drugs that only target specific components of the hypoxia signaling pathway have generally failed to produce an enduring clinical response in GB. It is thought that the complete inhibition of HIF-1α is necessary to show potent antitumor activity and to promote the activation of the immune system [243]. Inhibition HIF-2α, which can also block the hypoxia pathway, is an alternative attractive strategy for GB treatment. HIF-2α is specifically overexpressed in GB cells and GSCs, but not in normal tissues [244]. Although the HIF-2α inhibitor PT2385 had limited activity in rGB (phase II, NCT03216499) [245], other HIF-2α inhibitors that are currently under research may help in blocking GB progression. It is also important to mention that recent findings suggest that GB hypoxia regulates gene expression in an HIF-independent way. In that sense, Srivastava et al. demonstrated that FAT1 (a FAT atypical cadherin) modulates the epithelial-mesenchymal transition and stemness gene expression in hypoxic GB [246], and hypoxia induces epigenetic regulation of the transmembrane protein odd Oz, altering DNA methylation status and activating the ODZ1-mediated migration of GB cells [202].

Importantly, Aderetti et al. demonstrated the existence of hypoxic peri-arteriolar GSC niches in GB tumor samples [247]. Apparently, GSCs remain attached to peri-arteriolar niches by the same receptor–ligand interactions as hematopoietic stem cells in the bone marrow. GSCs’ infiltration can be promoted via VEGF secreted by endothelial cells, which may induce the trans-differentiation of GSCs into endothelial cells, promoting angiogenesis and invasiveness [86]. Furthermore, a phenomenon of metabolic zonation has been described depending on the relative distance between the tumor cell and the blood vessel [248]. Proximity to the blood vessels promotes the mammalian target of rapamycin mTOR-derived anabolic metabolism and enhances tumor aggressiveness and therapy resistance [248]. Indeed, GB cells located in the perivascular tier exhibit robust anabolic metabolism and deviate from the Warburg principle by extensively engaging in oxidative phosphorylation. These perivascular cancer cells acquire specific functional characteristics, such as heightened tumorigenicity, enhanced migratory and invasive abilities, and surprisingly, remarkable resistance to chemotherapy and radiation; most of these traits are dependent on the mTOR pathway [248].

The BBB is a major obstacle to drug penetration within the brain parenchyma. Only 20% of small molecules/therapeutics agents cross the BBB and reach tumor cells at an effective concentration. GSC or GB cells protected against therapeutic agents by an intact BBB are the source of tumor recurrence [23]. P-glycoprotein, multidrug resistance proteins, organic anion transporters and breast cancer resistance proteins are especially important efflux pumps within the BBB that limit the accumulation of small-molecule-targeted therapies [249]. To make it more difficult, GSCs overexpress ABC transporters, further hindering drug delivery. ABC transporters promote therapy resistance by promoting the efflux of exogenous compounds, such as TMZ, at the cellular and BBB levels [22]. Infiltrating tumor cells are known to compromise the integrity of the BBB, resulting in a vasculature known as the blood–tumor barrier (BTB), which is highly heterogeneous and characterized by numerous distinct features, i.e., non-uniform permeability and active efflux of molecules [249]. Therefore, delivering therapeutic agents across the BBB and BTB, but avoiding their accumulation in the healthy parenchyma, is essential to making significant progress in GB treatment.

## 3. Present Therapy and Challenges

### 3.1. Standard of Care in Newly Diagnosed GB Patients

The Stupp protocol became the standard of care (SOC) for newly diagnosed GB (ndGB) patients since a randomized phase III trial (Table 2) evidenced an improved mOS from 12.1 to 14.6 months and an increase in the 2-year survival rate from 10% to 27% [7]. This SOC includes maximal safe resection, radiotherapy with concurrent (75 mg/m^2^/day × 6 weeks) and adjuvant TMZ (150–200 mg/m^2^/day × 5 days for six 28-day cycles). Since then, similar results (mOS 15–18 months) have been observed in other clinical studies [250,251,252]. Despite significant advances in the understanding of the molecular biology and pathophysiology of the GB, SOC has remained unchanged, excepting the possibility of adding or not tumor treating fields (TTFields) [13,253,254].

GB mostly recurs within 2–3 cm from the borders of the initial lesion and with multiple lesions, thus, maximal surgical resection improves survival irrespective of the age of the patient or the molecular status of the tumor [252,268]. Preoperative brain mapping techniques such as navigated transcranial magnetic stimulation (nTMS), magnetoencephalography, functional MRI, and diffusion tract imaging (DTI) are used to facilitate safe resections and minimize surgical complications [269]. Compared to non-nTMS techniques, nTMS has been associated in GB patients with smaller craniotomy size, less residual tumor tissue, shorter hospital stays, and improved survival at 3, 6, and 9 months, with no significant difference in surgery-induced neurological deficits [270].

During surgery, various tools are employed to optimize the extent of resection and minimize residual tumor volume. These include functional monitoring, fluorescence-based visualization of the tumor using 5-aminolevulinic acid (5-ALA), ultrasonography, and intraoperative MRI (ioMRI) [268,269,271]. Additionally, techniques like evoked potentials, electromyography, and brain mapping in awake patients, under local anesthesia, are used to monitor and preserve language and cognition during resections in critical brain areas [272]. The use of the amino acid 5-Ala helps to identify tumor volume and areas of neoplastic infiltration through fluorescent visualization, improves PFS, OS, and reduces postoperative neurological damages [269,273,274,275,276]. 5-Ala has also been effectively used in rGB resection, but the risk of false-positive fluorescence for reactive non-tumor tissue is more remarkable in relapse forms, likely due to an altered BBB [277]. Nevertheless, recently, an off-label fluorophore (sodium fluorescein) has become popular due to numerous benefits compared to 5-Ala, including lower cost, non-toxicity, easy administration and a wide indication for other brain tumors [278]. Microsurgical resection of GB using sodium fluorescein has been associated with an increased GTR rate and OS [279], although it is still considered inferior compared to 5-Ala [280].

Intraoperative ultrasound (ioUS) involves the use of sonography to locate tumor tissue during surgery and to delineate it from healthy brain tissue. As opposed to 5-Ala, which can only identify high-grade gliomas, ioUS is able to identify both low- and high-grade gliomas. In practice, 5-Ala and ioUS are considered complementary techniques [269]. Intraoperative magnetic resonance imaging (IoMRI) improves the accuracy and definition of the tumor and provides near real-time information about the dynamic changes occurring during surgery [281]. Analysis of residual GB volumes and neurological outcomes demonstrates that ioMRI is significantly superior to 5-Ala and white-light surgery at comparable peri- and post-operative morbidities [282]. The combination of ioMRI and 5-Ala facilitated achievement of the highest extent of resection (95%), followed by ioMRI alone (94%), 5-Ala alone (74%), and no imaging (73%), and this was associated with fewer post-chirurgic neurological deficits [283,284]. However, the lack of evidence regarding the cost-effectiveness compared to less advanced techniques raises uncertainty [268,285]. Regardless of the technique used, a postoperative contrast-enhanced MRI should be carried out within 48 h to assess the extent of resection and serve as a baseline for further treatments. Additionally, MRIs are performed every 2–3 cycles of TMZ treatment to monitor the tumor’s response [286].

After surgery, the smallest amount of residual tumor correlates with higher survivals [287,288]. However, radical surgical resection is limited by the highly invasive nature of GB cells [289]. Additionally, postoperative complications are a negative prognostic factor, and in this it is essential to prevent permanent neurologic deficits to safeguard the quality of life of the patients [14,47,289]. Carmustine (BCNU) wafers placed in the tumor resection cavity at the time of surgery provide a modest survival advantage (≈2 months) [290]. Wafer implants have been approved by the FDA and the EMA, but are not included in the SOC mainly due to their limited brain penetration, safety and tolerability, and because the treatment may preclude patients from enrolling into clinical trials [291]. Where surgical resection is not possible, stereotactic biopsy or open biopsy are alternative options for histological diagnosis and further molecular testing, which can determine an optional therapy [292,293]. However, this recommendation is not exempt from criticisms since in GB patients with low-performance status and/or advanced age, biopsies imply very little clinical gain [294].

Compared with surgery alone, postoperative radiotherapy is used to control microscopic unresectable disease, delay neurological deterioration and increase survival [7,295]. Radiotherapy is less efficacious in hypoxic TME due to a lower oxidative stress and because cancer cells develop mechanisms to repair DNA [115].

TMZ is a mono-alkylating agent that induces cytotoxic lesions including N7-methylguanine, N3-methyladenine and O^6^-methylguanine. N7-methylguanine and N3-methyladenine are repaired by base-excision repair (BER) and contribute minimally to the overall cytotoxicity of TMZ, while O6-methylguanine is repaired by O^6^-methylguanine-DNA-methyl transferase (MGMT) [296]. Methylation of the MGMT gene promoter (40% of GB patients) causes a reduction in MGMT protein expression and activity that results in persistent O^6^MeG lesions that trigger replicative stress and cytotoxicity via futile cycles of mismatch repair (MMR) [297]. Therefore, MGMT promoter methylation confers a better prognosis and overall survival (OS) associated with a positive response to alkylating agents in GB patients aged <70 years [7,13,298]. Radiotherapy has been shown to upregulate MGMT, whereas prolonged exposure to alkylating agents may suppress MGMT activity making the cells more susceptible to TMZ [257]. Nevertheless, several trials (Table 2) evidence that extending post-radiation TMZ from 6 to 12 months does not improve PFS6 and is associated with greater toxicity, functional deterioration, and poorer quality of life [257,258].

TMZ is a cornerstone of GB treatment, but its effectiveness is limited by the blood–brain and blood–tumor barriers, and the inherently or acquired GB resistance [19,299,300,301]. Upon TMZ treatment, GB and GSC cells induce DNA repair mechanisms, NF-kB signaling mediated antiapoptotic pathways, the expression of anti-apoptotic Bcl-2 family members, EGFR activity, drug efflux by ATP-binding cassette (ABC) transporters, autophagy-mediated resistance, expression of STAT3 and miRNAs, and overexpression of antioxidant proteins [83,299,301,302,303,304,305]. Nitrosoureas, i.e., lomustine (CCNU), carmustine and procarbazine, were widely used before the availability of TMZ, but their use is now limited to the treatment of rGB. At is shown in Table 2, patients in good physical condition with hypermethylated MTMG promoters (NCT01149109) slightly increase their OS survival by the addition of lomustine to SOC (48.1 vs. 31.4 months) [259,306]. Nevertheless, the benefit of this regimen remains unclear since the sample size was small and few patients were able to complete all six cycles of adjuvant treatment due to the greater hematologic toxicity in the lomustine-TMZ arm [307]. The addition of BEV to the SOC improved PFS but not OS in both AVAglio and RTOG 0825 trials (NCT00943826 and NCT00884741) [260,262].

Up to now, TMZ is still commonly used for GBs with unmethylated MGMT promoters, due to the lack of significant benefits of alternative options (BEV plus irinotecan, dose-dense TMZ, BEV+SOC) [257,260,262,298]. Several preclinical studies demonstrated that O^6^-benzylguanine (O^6^BG) or O^6^-bromothenylguanine inactivate MGMT, but the addition of O^6^BG to radiation and BCNU treatment (Table 2, NCT00017147) did not provide further benefit and instead increased toxicity [264].

### 3.2. Tumor-Treating Fields (TTFields)

TTFields is a non-invasive cancer treatment modality that applies low-intensity (0.7–3 V/cm), intermediate-frequency (100–500 kHz), and alternating electric fields over regions of the body where tumors are localized [308,309]. In growing GB cells, TTFields cause chromosome missegregation, disrupt DNA repair, inhibit mitosis and the cell cycle, and induce apoptosis and autophagy [310,311,312,313,314,315,316,317]. TTFields also interfere with the directionality of cancer migration by inducing changes in the organization and dynamics of microtubules and actin and ablate the primary cilia on GB cells that contribute to tumor growth and chemoresistance to TMZ [318]. TTFields also downregulate the expression levels of VEGF, HIF-1α, and matrix metalloproteinases (MMP2 and MMP9), which are necessary for tumor growth, invasion and metastasis [319]. TTFields also increase the membrane permeability of cancer cells and the BBB [320,321], which can help to increase the uptake and bioefficacy of chemotherapeutic drugs. Although this treatment modality reduces the viability of proliferating T cells [322], it also stimulates maturation and phagocytosis by dendritic cells (DCs) [323], increases CD8 T infiltration in TME [315], promotes the production of type I IFNs in GB cells in a cGAS/STING- and AIM2 inflammasome-dependent mechanism [324] and, thereby, facilitates the immune system response. Interestingly, the combination of hyperthermia and TTFields has shown synergistic effects in GB [325].

Over the past decade, TTFields have emerged as a complementary treatment strategy, which is now part of the SOC in GB treatment [289,326,327,328]. The FDA’s approval of rGB was based on the results from the EF-11 trial (NCT00379470) showing that TTFields monotherapy provided similar efficacy compared to the best physician’s choice chemotherapy in patients with rGB, albeit with better quality of life, less toxicity and a lower incidence of serious adverse events [329]. A randomized clinical trial in ndGB patients (NCT00916409) previously treated with chemoradiotherapy showed that patients treated with the TTFields and TMZ had a median free survival (mPFS) of 6.7 months compared to 4.0 months with TMZ alone. The addition of TTFields to the SOC therapy improved median OS (mOS) from 15.6 to 20.5 months without a negative influence on the health-related quality of life [13,330,331]. In ndGB, TTFields are applied within 6 weeks after the end of the radio-chemotherapy, ideally simultaneously with TMZ monotherapy [13,328]. Patients with compliance > 90% showed extended median and 5-year survival rates [256]. The most common adverse effect is skin irritation, occurring in 43% of patients (2% grade 3 or higher) [13,309,327,332], which is generally managed with array relocation and topical treatments including antibiotics and steroids. The frequency of systemic adverse events was 48% in the TTFields-TMZ group and 44% in the TMZ-alone group. Several limitations should be noted in the NCT00916409 trial: (a) only PF patients after the completion of chemoradiation were enrolled, which excluded those who were more likely to have a poor prognosis; (b) randomization in the EF-14 trial occurred over 2 months after diagnosis, which suggests a selection bias of patients who did not have progression after the initial treatment and would therefore likely have a better survival rate; (c) a “sham” device—to better discern a potential placebo-effect of wearing the device—was not used; (d) second-line therapies (chemotherapies, salvage radiation, radiosurgeries or craniotomies) after tumor progression in both groups were not reported while the TTFields plus TMZ group allowed patients to continue TTFields for up to 24 months or after the second GB progression; (e) molecular markers, such as the IDH1/2 status, were not performed [308,333]. Recently, recognized brain cancer experts concluded that TTFields plus TMZ represents a major advance in the field of GB therapy, though other experts maintain their skepticism regarding the use of the TTFields because of the lack of effect in some patients and because the time lengths required to reach (modest) benefits (at least 18 h per day) limit its utility [308,309,334].

Dexamethasone is administered to GB patients to alleviate cerebral edema and provide symptomatic relief. However, the corticoid-induced immunosuppressive effects may also increase infections and decrease survival [335,336]. In fact, a recent meta-analysis revealed that dexamethasone interferes with the therapeutic effects of TTFields [324]. The threshold dose at which dexamethasone can be used with minimal interactions with the TTFields was 4.1 mg per day or lower [337]. Several ongoing clinical trials are studying the optimal timing for TTFields administration (e.g., NCT04471844, NCT04492163, NCT03705351) and the safety and efficacy of the combination of TTFields with other cancer modalities [308,315,316,338,339]. For instance, PriCoTTFields is a phase I/II clinical trial that evaluates the safety and efficacy of TTFields initiated prior and concomitant to combined radiation and TMZ therapy in ndGB patients [340].

TTFields can reduce the DNA double-strand repair by downregulating the activity of the breast cancer type 1 susceptibility (BRCA1) signaling pathway, thereby increasing the sensitivity to the blockade of DNA repair caused by PARP inhibition [341]. Consequently, an ongoing phase II trial (NCT04221503) will try to determine whether niraparib (a PARP inhibitor) can enhance the effect of TTFields in GB patients (NCT04221503). In addition, the combination of TTFields, TMZ and lomustine has shown benefits in ndGB patients [342] and the triple combination of BEV, irinotecan, and TMZ plus TTFields improved the OS of patients with rGB [343]. Mechanisms involved in the acquisition of TTField resistance include activation of voltage-gated Ca^2+^ channels linked to cell migration [344]; CDK2NA deletion, mTOR (V2006I) mutations [345], and the upregulation of autophagy which can be reversed by combining TTFields with an autophagy inhibitor [312].

In clinical practice, TTFields are mainly used at a frequency of 200 kHz, but preclinical studies show that different GB cell lines respond to other optimal electric frequencies, as is the case of SF188 (400 kHz) or U87 (100 kHz) [346]. This phenomenon highlights the need for further investigation to individualize “TTFields prescription”. Despite the advances associated with the incorporation of TTFields to GB treatment, its clinical use is still quite restricted. EANO guidelines argue that the clinical benefit of TTFields has not been established yet, which contradicts the ASCO-NSO recommendations [253,347]. Certainly, price, regulation, the increase in the efficacy of combined treatments, and likely the development of novel intracranial electrodes, may assist in increasing the utilization and acceptance of TTFields [316].

### 3.3. Treatment in Special Patient Populations

Elderly patients (>65 years) or patients with a poor functional status have worse prognosis and are less tolerant to toxicities. Surgical resection is not associated with improved survival [348], but according to a recent retrospective single-center study, BCNU wafer implantation during the surgical resection is safe and improves mOS 39.0 months (≥12 implanted wafers) vs. 16.5 months (<12 implanted wafers) in patients in “extreme” neurosurgical conditions (>80 years and patients with preoperative Karnofsky Performance Status score < 50) [349]. Although just a few patients (6/49) reached that number of implants, these results are impressive, since mOS in the “extreme” conditions subgroup was 10.0 months, and there was a significant improvement in the postoperative KPS score compared to the preoperative KPS score.

The combination of TTFields with maintenance TMZ resulted in improved PFS and OS in ≥ 65-year-old patients with ndGB in the phase III EF-14 trial, without affecting patient quality of life [350]. Nevertheless, clinical trials have shown that standard radiotherapy is associated with poor outcomes, especially in patients older than 70 years (Table 1, ISRCTN81470623) [266]. Here, abbreviated courses of radiation therapy must be considered [266,267], although age alone should not represent the sole determining factor for the duration and intensity of the therapy [351]. Hypofractionated radiotherapy schedule (40 Gy delivered in 15) fractions and the addition of concurrent and adjuvant TMZ (NCT00482677) significantly increase survival (9.3 vs. 7.6 months, respectively) without impairing the quality of life [267]. Consequently, partial-brain fractionated radiotherapy with concurrent and adjuvant TMZ is the SOC for elderly patients with good performance status [352,353]. The addition of BEV to radiotherapy had no benefits in elderly patients [354].

A single modality therapy can be considered for patients with poor functional status. RT was more effective than TMZ for unmethylated MGMT-promoter tumors, whereas TMZ was more effective than RT for methylated MGMT-promoter tumors [266,267].

### 3.4. Options of Treatment in rGB patients

Regardless of the use of multimodality treatments, GB invariably returns after a median interval of less than 10 months, and typically even sooner (≈6 months) in older patients [267]. The genetic and biological changes induced by radiotherapy and/or cytotoxic chemotherapy differentiate rGB from primary tumors. These changes empower GB tumors to navigate the host microenvironment, evade the immune system, and foster intrinsic and acquired resistance to further administration of radiation and/or alkylating agents. Upon recurrence, patients typically exhibit a poor performance status and compromised overall health, with GB tumors often being unresectable, thus requiring substantial use of corticosteroids to manage cerebral edema [355]. This makes rGB prognosis much worse than that of the primary GB.

Actually, although there is no clear SOC salvage therapy for rGB [289], patients who received no salvage treatment had poorer survival than those who received radiation and/or chemotherapy [356]. Therefore, re-resection, re-irradiation and systemic chemotherapy with TMZ rechallenge, nitrosoureas, BEV, and TTFields or clinical trial enrolment to test experimental drugs are considered for all recurrent patients [295,357,358,359,360]. Unfortunately, fewer than 43% of rGB patients were fit enough to be included in clinical trials [361].

Consensus guidelines for selecting candidates for second surgery recommend that patients need to have a good performance status, particularly if more than 6 months have elapsed since the initial surgery [357,362]. According to a retrospective review of the brain tumor database (1997–2016), stereotactic radiosurgery is associated with longer OS and/or PFS in rGB patients with good performance status and small-volume tumor recurrences [363]. In practice, not more than 20–30% of relapsed patients are eligible and only complete resections have any survival benefit (11–17 months) [364,365]. Toxicity to normal brain parenchyma limits re-irradiation in rGB [366]. Radiosurgery or hypofractionated radiotherapy (30–35 Gy in 5–15 fractions) is considered a potentially effective option and is increasingly used for younger patients with good performance status [356,367]. Data from a few prospective studies in rGB suggest that re-irradiation modestly improves PFS compared with systemic treatment alone [356]. OS after re-irradiation (9.7 months) was sufficient to justify this treatment [367,368], but marginal recurrence is significantly more frequent in patients who had prior BEV exposure [369].

Lomustine has become the SOC at relapse in Europe, with thrombocytopenia being the most frequent limiting toxicity [358]. Lomustine is generally preferred to other nitrosoureas given its oral formulation, schedule of administration, and better safety profile. However, lomustine activity is largely restricted to patients with tumors with MGMT promoter methylation and its survival benefit has been found limited: objective response rate was around 10%, mPFS < 2 months, PFS6 was 20%, and OS was 6–9 months [358,370].

One of the most significant features of GB is its hypervascularization, mainly promoted by the hypoxia-facilitated VEGF overexpression in tumor and stromal cells [371]. BEV is an anti-VEGF humanized monoclonal antibody that inhibits tumor-driven angiogenesis and may help in reducing patients’ immune suppression [234,372,373]. rGB with a low apparent diffusion coefficient, large tumor burden, or IDH mutation is more likely to benefit from BEV treatment [374]. BEV gained approval in 2009 for rGB treatment in the US and later in other countries, but BEV is not approved by the EMA as an SOC for rGB [375,376]. BEV has shown promise in extending PFS treating GB, but there is no evidence for its ability to prolong OS [377,378,379,380,381]. The anti-angiogenic effect of BEV decreases contrast uptake during MRI, which can lead to false negatives in recurrences [370]. Despite this, BEV is almost used due to the lack of alternative treatment options, and because it also serves to control brain vasogenic edema [289,375] and to avoid the need for corticoid treatment [382,383]. BEV combined with re-irradiation was found to be safe and tolerable and showed a significant reduction in the incidence of radiation necrosis, patient dependence on corticosteroids and improvement in the *Karnofsky score* during disease progression-free periods. Survival benefits (10.1 months) have been reported following fractionated stereotactic radiotherapy (35 Gy/10 fractions) and concurrent BEV in a prospective randomized phase II trial [384,385]. A recent retrospective study showed that BEV combined with re-irradiation improved mPFS and mOS to 8 and 13.6 months, respectively [359]. Nevertheless, the validity of these results is constrained by the inclusion of a small number of patients, the heterogeneity of treatment options, and the absence of a control group. Despite these limitations, recent conclusions drawn from a meta-analysis endorse the benefits of this therapeutic option [386]. Lomustine plus BEV for rGB (phase II, NCT01290939) somewhat prolonged PFS but did not confer a survival advantage over treatment with lomustine alone [387]. Although earlier reports suggested that BEV had glucocorticoid-sparing effects, in this trial, the addition of BEV did not reduce the use of glucocorticoids [387].

TTFields did not increase OS (phase III, NCT00379470) but showed efficacy equivalent to chemotherapy commonly used for rGB, with lower toxicity and improved quality of life [329]. According to a recent phase II trial (NCT01894061), the combination of BEV and TTFields is safe and has clinical efficacy in rGB [388].

## 4. Targeted Therapies

Genetic changes that have been well recognized in GB cells include alterations in the Rb/p16 pathway (> 90%), loss of heterozygosity of 10q (70%), EGFR amplification or mutation (≈50%), TP53 mutations (31%), PDGF receptor gain/amplification (≈25%), mouse double minute homolog 2 (MDM2) gene mutations (10–15%) and the phosphatase and tensin homolog (PTEN) gene mutations (20–34%) [16,389]. Analysis of the large-scale molecular and genomic information present in the *Cancer Genome Atlas Program* (TCGA) database indicated that p53 pathway (TP53/MDM2/P14^arfç^), the PI3K/Akt/mTOR pathway, and the RB pathway (CDK4/RB1/P16^ink4^) are the main signaling pathways involved in GB tumorigenesis, pathophysiology and acquisition of resistance to treatment [16,390,391]. Intrinsically targeting these altered molecules and pathways was seen as a novel avenue in GB treatment. Unfortunately, despite research efforts and clinical trials, except for prolonged PFS afforded by the BEV, no pharmacological intervention has been demonstrated to alter the course of disease [392,393].

### 4.1. pRB/CDK4/RB1/P16^ink4^

The retinoblastoma (RB) gene encodes a tumor suppressor protein (pRB) that inhibits the progression of the cell cycle. This gene is inhibited by the cyclin-dependent kinase (CDK) complex, especially CDK4, CCND2, and CDK6. The pRB-controlled pathway is often disrupted due to CDK4/6 amplification or CDKN2A/B loss or mutation [16]. The G1-S phase cell cycle checkpoint is mainly controlled by the kinases pathway, and thus improper formation of the cyclin D-1 complexes with CDK 4/6 leads to the promotion of cell-proliferation-involved carcinogenesis. Homozygous deletion of the CDKN2A-p16^INK4α^ gene in chromosome 9p is one of the three genetic traits of primary GB; in fact, activation of CDK4 is commonly observed in GB cells leading to cell invasion and stemness [394,395,396,397]. Three CDK4/6 inhibitors, including palbociclib, ribociclib and abemaciclib, have been approved by the FDA as monotherapy for treating breast cancers and are under clinical investigation in GB patients (Table 3) [398]. Palbociclib enhanced radiotherapy cytotoxicity in GB xenografts with RB expression [395] but failed to provide a benefit (NCT01227434) [399]. Ribociclib monotherapy exhibited good CNS penetration but showed limited clinical efficacy in patients with rGB, a fact that was attributed to upregulation of the PI3K/mTOR pathway [400]. A recent pharmacokinetic study [398] shows that abemaciclib penetrated into the human brain to a larger extent and was retained longer, thus representing a better treatment option. In GB xenograft models, combined treatment with CDK4/6 inhibitor and oncolytic virus (VSVΔ51) induced severe DNA damage stress, amplified oncolysis, inhibited tumor growth, and prolonged survival [401]. In the phase II trial (NCT02977780), the addition of abemaciclib to the SOC treatment in ndGB patients was well tolerated and prolonged PFS, but there was no evidence of an OS improvement compared to standard radio-chemotherapy [402,403].

CDKN2A encodes two tumor suppressor proteins INK4a (p16^INK4a^) and ARF (p14^ARF^), which are crucial regulators of the pRB- and p53-dependent growth control, respectively, [470]. One of the most frequent mutations found in GBs is the homozygous deletion of the p16^INK4a^/p14^ARF^/p15^INK4b^ locus [471]. Loss of p14^ARF^ promotes the accumulation of the p53 repressor, reducing p53 levels, and thereby promoting tumorigenesis. In turn, P16^INK4a^ inhibits the association of CDK4 and CDK6 with cyclin D. When p16^INK4a^ is lost, CDK4 and CDK6 associate with cyclin D and participate in the phosphorylation (inhibition) of the RB protein, which in turn facilitates the release of E2F (E2 transcription factor), leading to aberrant cell proliferation. Furthermore, Labuhn et al. found that the co-deletion of ARF and INK4a increased, accordingly with tumor progression, from low- to high-grade gliomas, thus suggesting that deletions of this locus may be fundamental for GB development [471]. A novel piperazine-based benzamide derivative regulates the cell-cycle-related proteins and influences the p16^INK4a^-CDK4/6-pRb pathway. It significantly inhibited the growth, migration, and invasion of human GB cell lines in vitro. The most interesting aspect of this study is that it could penetrate the BBB with an exceptional brain-to-plasma ratio of 1.07 in vivo, which was accompanied by a superior anti-GB potency on the U87-MG-xenograft model without any apparent host toxicity [472].

### 4.2. TP53/MDM2/P14^arfç^

TP53 is a gene that encodes the p53 tumor suppressor protein, commonly referred to as the “Guardian of the Genome” due to its ability to respond to genotoxic stress and to protect the genome by inducing a variety of biological responses including DNA repair, cell cycle arrest, and apoptosis [473,474]. The principal regulators of p53 are the proto-oncogenes MDM2 and ARF (open reading frame). MDM2 is an E3 ubiquitin ligase that hampers p53 by promoting its ubiquitination and subsequent degradation through the proteasome [475]. Conversely, ARF triggers the activation of p53 through direct physical interaction with MDM2, effectively blocking MDM2’s ability to interact with p53. This intricate regulation has fueled intensive research efforts and the development of successful therapeutic approaches to modulate p53 in recent years [476,477,478]. The p53-ARF-MDM2 pathway is dysregulated in 84% of patients with GB and in 94% of the GB cultured cell lines [479]. Furthermore, there is a high occurrence of TP53 and ARF co-inactivation observed in GB [480]. p53 mutations play a particularly significant role in the development of secondary GBs [470], and the insurgence/progression of GB, as well as the related chemoresistance, have often been attributed to MDM2 overexpression [481,482].

Activation of p53 has become a pivotal therapeutic objective within this regulatory trio. In China, starting in 2003, *wild-type* p53-induced expression within tumor sites has been utilized as a therapeutic approach for several cancer types [480]. Several pharmacological agents have been assayed in order to restore normal p53 functions. One option is to inhibit the MDM2/p53 complex to prevent degradation of the p53, thus restoring the function of *wild-type* p53 in tumors with mutant p53, and inhibiting gain-of-function mutations in mutant p53 [474,478]. Combining an MDM2 inhibitor with chemotherapy or radiotherapy, higher levels of p53 can be achieved, leading to the activation of apoptosis [482], which is also a viable strategy to overcome resistance to therapy [483,484]. AMG232 (a p53-MDM2 inhibitor) exhibited the most remarkable efficacy against GB stem cells and spheroids [479,485]. The first molecules reported to interrupt the p53–MDM2 interaction were nutlins (1, 2, and 3) [486]. Preclinical studies show that nutlin-3 induces apoptosis and cellular senescence in human GB cancer cells and enhances the efficacy of radiotherapy [487]. TMZ/nutlin3a was synergistic in decreasing growth of *wild-type* p53 GB cells. The inhibition of cell growth following exposure to TMZ/nutlin3a correlated with (1) activation of the p53 pathway; (2) downregulation of DNA repair proteins; (3) persistence of DNA damage; and (4) decreased invasion. In a xenograft model of intracranial GB, TMZ/nutlin3a treatment resulted in a significant increase in survival compared with single-agent therapy [483]. Nutilin-3 was followed by the development of other MDM2 inhibitors such as idasanutlin (RG7388, NCT03158389), AMG232 (NCT01723020), navtemadlin (NCT03107780) and BI-907828 (NCT05376800), which are undergoing clinical trials for GB and other brain cancers [482].

### 4.3. PI3K/Akt/mTOR

Constitutive activation and mutation of this [488] signaling pathway play crucial roles in the development of GB and are present in GSCs [297,489]. As illustrated in Figure 3, engagement of an extracellular domain with its ligand activates the intracellular tyrosine kinase that translocates to the plasma membrane resulting in the formation of phosphatidylinositol 3,4,5-triphosphate (PIP3), which further stimulates the serine/threonine kinase phosphoinositide-dependent kinase 1 (PDK1) and protein serine-threonine kinase (Akt, also known as protein kinase B) activities [490]. The tumor suppressor phosphatase and tensin homolog deleted from chromosome 10 (PTEN) negatively regulate the PI3K/Akt pathway by removing the phosphate group from PIP3 to PIP2 [491,492]. PTEN also functions as a lipid phosphatase and in this manner; it is capable of regulating cell polarity, motility and senescence [493] and, in addition, is involved in modulating innate and adaptive immune responses [494]. Akt is a crucial regulator of cell proliferation and survival and is hyperactivated in many cancers. After Akt phosphorylation and induction, mTOR, as a downstream target, is activated. In the PI3K pathway, mTOR acts as a downstream effector and also as an upstream regulator of apoptosis and cell cycle progression. mTOR is a protein kinase localized in two structurally and functionally distinct multiprotein complexes known as mTORC1 and mTORC2 [495,496]. Via the PI3K/Akt pathway, growth factors, low energy status, low oxygen level, and DNA damage converge in mTORC1 activation which, in turn, influences cell growth and proliferation by promoting biosynthetic pathways, limiting catabolic processes, and inhibiting autophagy [497,498]. Differently, mTORC2 activation is triggered by growth factors but does not respond to nutrient availability. Once activated, mTORC2 drives reorganization of the cytoskeletal structure, motility, cell proliferation, and survival and is also involved in the induction of the Warburg effect [495,499]. Recently, it has been shown that acetyl coenzyme A is used by GB cells to induce RICTOR acetylation (a core component of the mTORC2 signaling complex) that results in mTORC2 activation. This mechanism creates an autoactivation loop whereby mTORC2 triggers cell proliferation and growth, bypassing growth factor-activated upstream signaling and rendering GB cells resistant to receptor tyrosine kinase and mTORC2 inhibitors [500].

The PI3K pathway is altered in about 70% of GBs, either by deletion of PTEN or amplification of EGFR and/or VEGFR and/or PDGF receptor (PDGFR) [470,501,502,503]. PTEN is deleted or mutated in 30–60% of GB cases and, consequently, the PI3K signaling pathway is hyperactivated, which in turn accelerates tumor growth, progression, and metastasis [504,505].

EGFR alteration, including overexpression or gene amplification, is present in 50–60% of GBs [9]. EGFR can activate a variety of signal transduction pathways (PI3K/AKT/mTOR, RAS/RAF/MEK/ERK, JAK/STAT, and PKC)[506,507,508]. EGFR gene amplification is detected in 57.4% of primary GBs, whereas the EGFR variant III (EGFRvIII) is a characteristic mutation that emerges later in tumor development [509,510]. EGFRvIII is characterized by a deletion of 267 amino acids in the extracellular domain, leading to a receptor that is unable to bind to a ligand yet is constitutively active [511,512]. Together with its impaired internalization and degradation, the EGFRvIII enhances the tumorigenic potential of GB by activating and sustaining mitogenic, anti-apoptotic and pro-invasive signaling pathways [513]. EGFRvIII is associated with a more aggressive tumor behavior and worse prognosis and contributes to therapy resistance [514,515,516,517]. However, GB samples taken from the first diagnosis and at the time of progression show that in 80–90% of cases, the EGFR amplification status is unchanged, whereas the expression of EGFRvIII often changes [429].

First-generation EGFR inhibitors (gefitinib, erlotinib and lapatinib) were designed to orthosterically block the ATP/substrate-binding pocket of EGFR (Figure 3) [510]. Although these inhibitors showed promising results in inhibiting growth and improving survival in preclinical models, no benefits were observed in clinical trials (Table 3) [435,444,445,518,519,520,521,522,523,524]. Erlotinib as monotherapy or as part of combined treatments has been assayed in multiple trials in rGB and ndGB (Table 3). It is worth mentioning the study carried out by D’Alessandris et al. in which, despite not giving positive results, the administration of BEV and erlotinib was tailored to the molecular profile of the patient’s tumor (VEGF overexpressing tumors or EGFRvIII tumors, respectively) [436]. A phase II trial showed that ndGB patients treated with erlotinib plus SOC had significantly improved OS (19.3 vs. 14.1 months in historical controls) [438], which is surprising because just one year later, another phase II trial reported that combined treatment not just was ineffective but also had an unacceptable toxicity [439]. The effectiveness of erlotinib was also assessed in combination with SOC and with BEV in patients with unmethylated MGMT promoter (Table 3), with a lack of survival benefits [437,439]. Lapatinib, which binds to EGFR better than erlotinib or gefitinib, showed a modest inhibition of EGFR achieved in biopsied post-treatment tumors, highlighting the important limitation of crossing the BBB [525,526].

Second-generation EGFR inhibitors (afatinib and dacomitinib) are designed to bind irreversibly to the tyrosine kinase domain of EGFR and other ERBB family members [527]. Both showed good safety but limited single-reagent activity in clinical trials in rGB patients (Table 3) [404,427,528]. Afatinib (with and without TMZ) showed limited activity in unselected patients, but an increase in the PFS was observed in patients overexpressing EGFR or expressing EGFRvIII [404]. Dacomitinib (PF-00299804) had limited activity in rGB with EGFR amplification, although the molecular characterization of four patients with positive response can be useful to select patients who could benefit from EGFR inhibition [427,528,529]. The third generation of EGFR inhibitors (rociletinib and osimertinib) was designed to target the T790M resistance mutation, which is responsible for about 50% of the acquired resistance to the earlier generation of tyrosine kinase inhibitors [530,531]. Osimertinib (AZD9291) induced GB cell cycle arrest and significantly inhibited colony formation, migration, and invasion to a greater extent than other EGFR tyrosine kinase inhibitors, and prolonged survival of orthotopic GB-bearing mice [532]. Compared with afatinib, AZD9291 demonstrated lower potency in inhibiting EGFRvIII, but it showed excellent BBB penetration making it an attractive candidate for inhibiting EGFR in GB. ZD9291 has been approved for the treatment of lung cancer with good safety and tolerability. ZD9291/BEV combination was marginally effective in most rGB patients with simultaneous EGFR amplification and EGFRvIII mutation, but it is interesting to mention that OS was superior to regorafenib (7.4 months) and a subgroup experienced a long-lasting meaningful benefit [533]. Several anti-EGFR monoclonal antibodies, such as cetuximab, GC1118, ABT-806, ABT-414 and nimotuzumab (Table 3 and Figure 3) failed in clinical trials to increase OS in GB patients [423,424,429,430,451,534,535]. Depatuxizumab mafodotin (depatux-M, ABT-414) is a tumor-specific antibody–drug conjugate comprising ABT-806 and the toxin monomethyl auristatin-F. It was evaluated in a randomized controlled phase II trial (NCT02343406) for EGFR-amplified rGB patients, either as a single agent or in combination with TMZ. The combination therapy demonstrated an improved OS of 19.8% compared to 5.2% in the control group and 10% in the monotherapy group. These findings suggest clinical benefits for the Depatux-M + TMZ combination, particularly in patients relapsing more than 16 weeks after the last TMZ cycle [429]. On the contrary, Depatux-M + TMZ treatment in ndGB patients had negative outcomes in a phase III clinical trial (NCT02573324) [430].

Nimotuzumab is another monoclonal antibody against EGFR that binds more specifically to EGFR overexpressing cells. It showed promising efficacy in a phase II trial for high-grade glioma, but failed to increase survival in a randomized phase III trial in ndGB patients [451]. The heterogeneity of GB tumors, in which EGFR deletion and EGFR amplificated mutations can coexist in different cells, leads to adverse effects arising from collateral inhibition of EGFR in normal tissues, as well as to redundant and alternative compensatory pathways, which represent the most important escape mechanisms that limit the anti-glioma effects of the different EGFR-targeting drugs [536,537,538,539]. 

Similar to EGFR, the PDGFRs (α and β) are involved in the activation of the PI3K (Figure 3) [540]. PDGFRα is the second most frequently amplified tyrosine kinase receptor in GB behind EGFR, and its expression associates to poor prognosis [541]. Once activated, PDGFR triggers intracellular signaling cascades that regulate cancer cell survival, growth, and progression. Consequently, dysregulation of PDGF signaling stimulates malignant transformation of normal neural stem cells into GB cells and enhances GB cell growth and motility through autocrine signaling [542,543]. No specific PDGFR inhibitor exists; thus, PDGFR is either targeted by specific anti-PDGFR antibodies (IMC-3G3 and MEDI-575) or multi-kinase inhibitors (alone or in combination with other chemotherapeutics). MEDI-575 was well tolerated but showed limited activity in rGB [450]. Pazopanib, sorafenib and sunitinib inhibit both VEGFR and PDGFR but clinical evaluations (Table 3) did not provide any significant benefit in GB [441,452,456,457,458,459,460,522]. Similar results were obtained in other clinical trials with dasatinib [428] and imatinib plus hydroxyurea [544]. ERBB3 (Erb-B2 Receptor Tyrosine Kinase 3), IGF1R, and TGFβR2 seem to be responsible for the resistance to PDGFR inhibitors. Consistent with this notion, a combination of PDGFR inhibitors with inhibitors targeting either ERBB3 or IGF1R more potently suppressed the growth of GB cells than each inhibitor alone. ERBB3 has been suggested as potential prognostic marker and therapeutic target for GB with high PDGFR-α expression [545]. Recently, the PDGFRα/β inhibitor CP-673451 induced differentiation of GSCs and improved the anti-tumor effects of TMZ in vivo using a subcutaneous xenograft mouse model [546].

Targeting angiogenesis has been and continues to be an attractive therapeutic modality in GB. The best-known angiogenesis regulators in GB progression include VEGF, basic fibroblast growth factor (bFGF), PDGF, EGF, TGFβ, MMPs, and angiopoietins [547]. These angiogenic factors are upregulated in GB by a variety of mechanisms including oncogene activation, loss of tumor suppressor gene function, and/or hypoxic microenvironments [243,548]. Loss of the PTEN signaling leads to VEGFR2 expression in GB cells, which may contribute to resistance to anti-angiogenic treatments [549], and is also involved in resistance to TMZ [550]. As was previously mentioned, trials evaluating BEV (Table 1 and Table 3) as monotherapy or in combination with lomustine or TMZ have not improved OS in GB [260,262,387,409,411,412], and lack of discernible benefits was evident in clinical trials where BEV was combined with erlotinib, temsirolimus, sorafenib, or tandutinib in rGB [415,455,464,519]. Despite these negative results, BEV is widely used for the treatment of rGB and, in fact, it is often the control arm in many clinical trials. Regorafenib is an oral multikinase receptor inhibitor that blocks tyrosine kinases active in angiogenesis (VEGFR1–3 and TIE2) and cancer growth (KIT, RET, RAF1, BRAF and BFRAF^V600E^), growth factors (FGFR and PDGFR), and tumor immunity promoters as the colony-stimulating factor 1 receptor (CSF1R) [370]. Regorafenib has shown superior activity to lomustine in rGB patients with tumors bearing the methylated MGMT promoter. The REGOMA phase II randomized trial (NCT02926222) showed a small mOS benefit of regorafenib (7.4 months) versus lomustine treatment (5.6 months) [454,551]. Age and MGMT methylation appeared to influence OS, thus suggesting that regorafenib treatment could be an alternative in older patients [552]. Regorafenib (phase II/III trial, NCT03970447) has not received the EMA approval, although it is the first-choice treatment for rGB according to Italian Association of Medical Oncology guidelines [553,554]. Most probably, this is due to some questioned results of the clinical trial [555]: (1) low OS in the lomustine group (other clinical trials achieved mOS rates of 8–10 months); (2) the absence of a molecular tumor characterization in 31% of 119 recruited patients; (3) high rate of adverse effects in the regorafenib arm (grade 3–4 adverse events occurred in 56% of patients) in 40% of patients treated with lomustine; and (4) the difference in the two groups of study, i.e., two patients with more favorable prognostics (IDH-mutated GB) were included in the regorafenib group, whereas no IDH mutation was reported in the lomustine group [454]. Multivariate analysis confirmed that MAPK pathway mutations predicted a shorter PFS after regorafenib treatment, while EGFR-altered cases had a better response to regorafenib [556]. Further data from retrospective studies support the efficacy of regorafenib in GB [557], and retrospective analysis on 54 rGB patients treated with regorafenib reports an even longer mOS of 10.2 months [558]. Five trials of regorafenib are ongoing (NCT04810182, NCT04051606) or are being recruited (NCT06095375, NCT03970447, NCT06047379) for the treatment of GB.

Despite the importance of angiogenesis in the growth and acquisition of resistance of GB, none of the anti-angiogenic therapies tested in clinical trials (Table 3), i.e., vandetanib (NCT00821080) [465], cediranib (NCT01062425, NCT00777153, NCT01310855) [421,446,559], dovitinib (NCT01753713) [431], and axitinib (NCT01562197, NCT01290939) [407,560], have managed to improve the benefits obtained of historical controls. Preclinical studies showed that cediranib sensitizes cancer cells to PARP inhibitors by downregulating homology-directed DNA repair [561], but cediranib combined with olaparib (NCT02974621) failed to increase PFS and OS in rGB patients [426]. These results demonstrate that the use of anti-angiogenic agents (either as first- or second-line treatment, and either as a single agent or in combination with chemotherapy), while possibly improving quality of life, does not extend survival in unselected GB patients [230,393,562]. Mechanisms involved in disappointing results of anti-angiogenic chemotherapy in GB include the compensatory switch to alternative angiogenic pathways (i.e., the vasculogenesis mentioned before), GSC transdifferentiation, temporary and dependent treatment dosage and duration until the vasculature returns to normal, the impact of angiogenesis inhibition on tumor distribution of other chemotherapeutic agents [563,564], and upregulation of the hypoxia-inducible protein 2 [244].

Currently, more than 50 PI3K inhibitors have been designed and produced for cancer treatment, but just a few, such as enzastaurin, BKM120, XL147 and XL765, have entered into clinical trials for GB treatment [565,566]. Enzastaurin was well tolerated and had an acceptable hematologic toxicity profile (phase III, NCT00295815), but did not show superior efficacy than lomustine [433]. Currently, buparlisib (BKM120) is the most frequently tested panPI3K inhibitor in clinical trials (Table 3) since it is well tolerated and permeable to the BBB [126]. BKM120 induces G2/M cell cycle arrest and apoptosis in GB cells through microtubule misalignment and mitotic dysfunction in a p53-dependent manner [567], and has shown antiproliferative and proapoptotic properties in various GB cell lines, as well as in xenograft models [568,569,570,571]. However, single-agent BKM120 had minimal response (NCT01339052) in patients with PI3K-activated GB at first or second recurrence. BKM120 had significant brain penetration, and the lack of clinical response was attributed to the incomplete blockade of the PI3K pathway in tumor tissues [417]. Paradoxically, even though BKM120 inhibits Akt phosphorylation, patients harboring PTEN loss and/or PIK3CA mutations were not sensitive to BKM120 treatment [572]. The results of clinical trials (Table 3) in which BKM120 was part of combined treatments with BEV [418], carboplatin, lomustine [419], capmatinib [420] did not show benefits vs. monotherapies.

Akt inhibitors or induction of PTEN expression can reverse the resistance and sensitize cells to chemo- and radiotherapy by impairing DNA repair [573,574]. Akt inhibition induced by perifosine (Figure 3) enhanced the growth inhibition effects of low-dose heavy-ion radiation on GB C6 cells, via proliferation inhibition, apoptosis and oxidative stress [575]. However, perifosine faces certain constraints, such as the restricted ability to penetrate the BBB, which justifies its lack of effectiveness in the clinical trial for rGB (NCT01051557) [453]. The drugs that target Akt, combined with TMZ and fractional radiation, gradually enter the field of vision because chemoradiotherapy should increase the level of phosphorylated Akt [574]. To counteract the impact of Akt, which remains persistently active in GB, an alternative therapeutic strategy involves targeting mTOR, which is hyperactivated in 90% of GBs and is associated with poorer prognosis **[505].**

The pharmacological inhibition of mTOR inhibits GB growth, induces autophagy and reduces the invasive potential of GSCs [576]. Rapamycin (sirolimus) and its analogs, including everolimus (RAD001), temsirolimus (CCL-779), and ridaforolimus (AP23573), exhibit inhibitory effects on mTORC1 in in vitro and in vivo models and have been extensively clinically studied (Table 3) for GB treatment [442,462,463,492,577,578]. The effectiveness of rapamycin was noticed in terms of antitumor activity during a phase I trial involving patients with rGB [579], but in combination with erlotinib, did not render satisfactory results due to persistent mTORC2 signaling [440,580]. Combining everolimus with conventional SOC leads to increased toxicities and has no survival benefit in ndGB [442]. Surprisingly, these adverse events were not shown when BEV and everolimus were assayed as part of first-line combined modality therapy for GB (NCT00805961). Here, the PFS compared favorably to previous reports with SOC and was similar to results achieved in other trials in which BEV was added to first-line treatment [443]. Third-generation inhibitors of mTOR, (torin1 and 2, vistusertib (AZD2014) and rapalink-1) bivalently target both mTORC1 and mTORC2 [581]. In preclinical models, torin2 effectively inhibited both mTOR pathways in GB, leading to significant suppression of proliferation and migration of GB cells and GSCs [582]. Torin2 displayed a longer half-life, improved water solubility, and better oral bioavailability than torin1. Rapalink-1 targets GSCs and acts synergistically with TTFields to reduce resistance against TMZ [583]. Similarly, vistusertib enhances the radiosensitivity of GSCs in both in vitro and in vivo conditions [584] and, combined with TMZ, shows good safety at the tested dose levels in patients with rGB (phase I, NCT02619864) [585]. The clinical lack of efficacy of mTOR inhibitors can be attributed to the activation of parallel signaling pathways (MAP/ERK) or adaptive and acquired resistance after an initial response [586]. Moreover, mTOR inhibitors act both on tumors and on immune cells; thus, one can hypothesize that the putative anti-tumor efficacy of mTOR inhibitors might be counterbalanced by their suppressive effects on immune cells, thereby building an immunosuppressive environment that facilitates tumor progression [497].

Despite strong preclinical evidence for the therapeutic potential of tyrosine kinase inhibitors targeting the PTEN/PI3K/Akt/mTOR pathway, clinical trials conducted over the past 20 years have not resulted in the desired therapeutic breakthrough for GB [497,503,514,565,587,588]. Brar et al.’s (2022) review focuses on the potential of tyrosine kinase inhibitors in GB therapy and provides a clear insight into the reasons behind the unsuccessful clinical trials in GB, despite the success in treating other cancer types [589].

### 4.4. RAS/RAF/MEK/ERK

The Ras/RAF/MEK/ERK pathway (or MAPK signaling pathway) is hyperactive in virtually all GBs owing to the overexpression of key regulators like EGFR and PDGFR [546,590]. Activation of the Ras protein occurs through the replacement of GDP by GTP, initiating the activation of MAP kinases. These kinases phosphorylate and activate downstream ERK proteins, which translocate to the nucleus and induce transcriptional pathways, leading to cellular proliferation, survival, and dedifferentiation (Figure 3).

The BRAF^V600E^ missense mutation leads to constitutive activation of the Ras/Raf/MEK/Erk pathway, promoting tumor cell proliferation, survival and inhibition of apoptosis [591,592]. GBs with BRAF mutation differ significantly in location, age of diagnostic, survival rates, and global gene-expression profiles from the rest of GBs [593,594]. Several cases of impressive response to BRAF^V600E^ inhibitors (vemurafenib and dabrafenib) in GB have been reported, including cases of prolonged disease control [595,596,597,598]. The clinical trial (NCT01524978) showed that vemurafenib treatment had a durable antitumor activity in some patients with BRAF^V600^-mutant gliomas, although efficacy seemed to vary qualitatively by histologic subtype [599].

MEK1/2 are involved in tumor development and apoptosis inhibition, enhancing DNA damage repair capacity and migration/invasion of GB cells. MEK inhibitors (MEK162, trametinib, PD0325901) in GSCs showed antiproliferative and apoptotic cell death and induced neuronal differentiation in sensitive GSCs. Trametinib decreased tumor growth and improved survival in GSC xenografts [600]. Targeting multiple RAS effector pathways with a combination of MEK and mTOR kinase inhibitors is a strategy that has been tested in melanoma and has provided survival benefits in preclinical models of glioma with BRAF mutations or KRAS mutations [601]. At present, concurrent use of trametinib and dabrafenib exhibits notable clinical importance in addressing low-grade gliomas that carry the BRAF^V600E^ mutation [602]. A recent case report presented the case of a young female with BRAF^V600E^ GB who had a prolonged response to targeted therapy with dabrafenib and trametinib [603]. Although the specific BRAF^V600E^ mutation is uncommon in GB patients, dabrafenib in combination with trametinib showed clinically meaningful activity in patients with recurrent or refractory BRAF^V600E^ mutated high/low-grade glioma (phase II, NCT02034110) [432]. The significance of the results is limited because just 31 GB patients were included in the study, but we completely agree with the conclusions of researchers that “genetic testing tumor profiling should be introduced early in the management plan to promptly identify those patients who may be eligible for BRAF^V600E^-targeted treatment”, which should be extended to all GB studies.

### 4.5. PARP Inhibitors

Poly(ADP-ribose) polymerase-1 (PARP-1) facilitates the repair of DNA strand breaks. PARP-1 mRNA expression is associated with poor survival in GB patients [604]. Since TMZ induces replicative stress via futile attempts of DNA repair at O6MeG:T mismatches, and PARP inhibitors (PARPis) enhance stress by compromising the stability of stalled replication forks, it seems logical to think that PARPis may restore TMZ sensitivity in GB and GSCs [605,606,607]. In addition, PARP inhibitors synergize with radiation therapy because PARP-1 activity increases 500-fold in the presence of DNA damage [608].

Although several promising PARPis have limited distribution across BBB, some of them reached GB margins in vivo. The brain-to-plasma concentration ratio of veliparib (ABT-888) is substantially higher than other PARPis such as olaparib, rucaparib, or talazoparib. Veliparib restores sensitivity in TMZ-resistant glioma cells and xenografts [609] and extends survival in the MGMT-hypermethylated GB model, but is ineffective in MGMT-unmethylated lines [610]. Unfortunately, veliparib did not increase the efficacy of TMZ in rGB patients (NCT01026493) [467] and had no clinical benefit in ndGB patients with hypermethylated (NCT02152982) [469] or unmethylated-MGMT promoter status (NCT02152982) [468]. The authors support the clinical interest of the last trial, based on the radiosensitizing effects of veliparib [611], and mention “encouraging responses when applied to MGMT-unmethylated cell lines” without citing their sources. Unfortunately, they did not take into consideration the contradictory results previously shown in [610] that are in agreement with the lack of efficacy of the treatment in this trial. More recently, Wu et al.’s (2021) results demonstrated that inhibition of PARylation by PARP inhibitor (talazoparib) reduces MGMT function rendering sensitization to TMZ, providing a rationale for combining PARP inhibitors to sensitize TMZ in MGMT-unmethylated GB [612]. The potential of veliparib in GB can best be demonstrated in patients with PTEN null tumors; therefore, clinical trials with veliparib should evaluate these patients as a separate group [612]. Olaparib combines DNA repair inhibition and impairment of cancer cell respiration as anticancer activities [613]. In vivo, mice treated with TMZ alone or in combination with olaparib showed greater survival than those untreated or with the olaparib monotherapy, as well as a significant decrease in tumor volume; however, there was no significant difference in survival between both groups [614]. Encouragingly, olaparib was found to penetrate brain tumors at radiosensitizing concentrations (NCT01390571). A combination of olaparib and low-dose TMZ was safe and well tolerated (NCT01390571), yielding PFS6 rates [615,616].

PARPi meriting further clinical study include pamiparib and talazoparib (NCT04740190). Compared to olaparib, pamiparib demonstrated improved penetration across the BBB in mice and the strongest capacity to inhibit brain tumor growth in the xenograft model. EGFR-amplified GSCs showed remarkable sensitivity to talazoparib treatment, which significantly suppressed tumor growth in EGFR-amplified subcutaneous models [617]. In the SCLC-derived TMZ-resistant H209 intracranial xenograft model, the combination of pamiparib with TMZ overcomes TMZ resistance and shows significant tumor inhibitory effects and a prolonged life span [618]. These results support the phase I trial (NCT03150862) in which pamiparib is being assayed in combination with SOC in nd/rGB patients [619].

Bisht et al. have recently reviewed several strategies to increase penetration through BBB and reverse the resistance to PARPi [620].

### 4.6. Other Assayed Strategies

In preclinical studies, statins (well known as hypocholesterolemic drugs) exerted potent anti-GB effects through different mechanisms stemming from the inhibition of the mevalonate cascade, and resulting in the inhibition of proliferation, migration, invasion, and in the induction of apoptosis and autophagy [621,622,623,624,625]. Blockade of the Ras/MEK/ERK and Ras/PI3K/Akt pathways by statins reduces the expression of TGFβ as an angiogenic factor in GB [626]. Synergistic action with other anticancer drugs has also been described [627,628,629]. For instance, atorvastatin augments TMZ’s efficacy in vitro as well as in GB xenografts via prenylation-dependent inhibition of Ras signaling [630]. However, atorvastatin did not improve PFS when was evaluated in combination with radiotherapy and TMZ in GB patients (phase II, NCT02029573) [406].

As tyrosine kinase receptors usually activate similar downstream pathways, in GB upregulation and/or activation of IGF receptors (IGF-1R and IGF-2R), c-Met, and PDGFRβ contribute to resistance to EGFR/EGFRvIII inhibition [505]. Approximately 37% of patients with GB have c-Met overexpression, which plays an important role in promoting invasion and tumor recurrence [631]. Moreover, concomitant c-Met/VEGFR2 overexpression was associated with worse overall survival in GB and is linked to resistance to anti-angiogenic drugs [632]. A phase II study investigated the effect of the monovalent MET inhibitor onartuzumab plus BEV in rGB, but found no evidence of further clinical benefit [413]. Crizotinib in combination with SOC was safe and resulted in a highly promising efficacy for ndGB (phase Ib/ NCT02270034), warranting further investigation [633].

IGF-R1 is overexpressed and is necessary for neoplastic transformation in GB. IGF-1R blockade can inhibit GB growth by different mechanisms, including direct effects on the tumor cells as well as indirect anti-angiogenic effects [634]. High IGF1R expression is associated with chemoresistance to TMZ as well as reduced survival, thus suggesting its possible use as a biomarker [635]. Tyrosine kinase inhibitors, anti-sense oligonucleotides, or monoclonal antibodies (cixutumumab) targeting the IGF-IR have been clinically tested to inhibit IGF signaling in GB, with lack of benefits or inconsistent results [636]. On the contrary, autologous glioma cells treated ex vivo with an antisense oligodeoxynucleotide targeting the IGF-IR, and re-implanted in patients (phase 1b, NCT02507583), seem to increase the OS in ndGB-treated patients [637].

Some natural polyphenols (resveratrol, curcumin, silibinin, quercetin, etc.) have shown antiGB potential and synergize with radio/chemotherapy in vitro. Inhibition of proliferation, migration, cell invasion and angiogenesis are mechanisms proposed to explain the potential effect of polyphenols on reducing GB progression [638,639,640,641,642,643]. Specifically, metformin and resveratrol have been suggested to inhibit GB cell proliferation, invasion and migration by downregulating the PI3K/Akt pathway, activating mTOR, and increasing AMPK phosphorylation [644]. Rutin, epigallocatechin-3-gallate, quercetin and curcumin have shown synergistic effects with TMZ, whereas curcumin enhances the action of etoposide, paclitaxel, cisplatin, camptothecin, and doxorubicin [641,645,646,647]. Nevertheless, most of these proposed mechanisms only have in vitro support. It should be pointed out that, under in vivo conditions, the antitumoral activity of polyphenols is limited due to their short half-life. Even when polyphenols are administered at high doses, their rapid metabolism precludes reaching efficacious anti-tumor concentrations in a growing cancer. In practice, after an oral dose as high as 100 mg/kg, the peak plasma level of resveratrol was 11 ± 4 μM at 10 min, pterostilbene was 25 ± 6 μM at 10 min, EGCG was 5 ± 2 μM at 30 min, curcumin was 27 ± 6 μM at 15 min, quercetine was 8 ± 2 μM at 15 min, and genistein was 17 ± 5 μM at 10 min. Plasma levels of all these PFs were <1 μM at 60 min [648]. Exposure of many different cancer cells (i.e., U87 and LN229 GB cells) to these concentrations for just 1 h does not affect their growth and viability. Our results are supported by Beylerli et al. who recently pointed out that, although potential beneficial effects exerted by polyphenols are promising, their efficacy in vivo is strongly limited by their bioavailability and BBB permeability [646]. Zanotto-Filho et al. showed that resveratrol, a BBB-permeable drug, improved TMZ/curcumin efficacy in brain-implanted tumors in rats [647]. In these experiments, 50 mg curcumin/kg × day and 10 mg resveratrol/kg × day were administered i.p., but it is uncertain at which concentration each polyphenol reached the tumor and for how long. Based on the above discussion, it is easy to deduce that the tumor levels of in vivo administered polyphenols will be very low and acting for a short period of time. Facts that raise many doubts about the real mechanisms involved in the anti-tumor effects. The effect of curcumin has also been assayed using its intratumoral injection [649]. In this case, curcumin (100 mg/kg) inhibited U87 xenografts growth by approx. 50%. However, all the mechanisms proposed for curcumin were studied under in vitro conditions and using unreliable conditions compared to the in vivo setting. Methods to improve polyphenol absorption, pharmacokinetics, and efficacy [650,651,652] are key in order to deliver therapeutic concentrations at specific target sites and to increase the antitumoral bioefficacy of polyphenols in vivo.

## 5. Immunotherapies

Recent studies show the presence of a variety of immune cell types within the GB TME with a dominance of immunosuppressive cells, i.e., MDSCs, microglia, M2 macrophages, FoxP3+ regulatory T cells (Tregs), and antigen-presenting cells (APCs) (including DCs and bone-marrow-derived macrophages). The presence of M2 macrophages is linked to an increased GB aggressiveness and plays a pivotal role in the acquisition of chemo and radioresistance of GB cells [653,654]. In addition, frequently, CD4+ and CD8+ T cells are functionally deficient, inactivated, or exhausted, often co-expressing immune checkpoint molecules, i.e., programmed cell death receptor 1 (PD-1), lymphocyte activation gene 3 (LAG3) and T cell immunoglobulin mucin 3 (TIM-3) [655]. GB cells secrete immunosuppressive factors such as TGFβ2, PGE-2, IL-1, IL-10 (check Section 2.1) and indoleamine 2,3-dioxygenase (IDO), which work cooperatively to suppress the activity of effector cells and to evading the anti-tumor immune response [516,656]. As described here below, a plethora of novel immunotherapies, i.e., checkpoint inhibitors (ICIs), vaccines, T-cell-based immunotherapies, NK-cell-based therapies, viral therapies, and combined treatments, have been attempted in order to control GB expansion and/or recurrence.

### 5.1. Checkpoint Inhibitors (ICIs)

Cytotoxic T lymphocyte antigen 4 (CTLA-4) and programmed cell death 1 (PD-1), also known as checkpoints, are co-inhibitory receptors expressed on the T cells surface to promote immunotolerance to self-antigens. GBs (and other cancer cells) overexpress programmed cell death ligand 1 (PD-L1) which, upon interaction with PD-1, inhibits T cell proliferation and the T cell receptor (TCR)-dependent IL-2 production and suppresses the CD4+ and CD8+ response [657]. Thus, PD-L1 overexpression triggers T cell exhaustion and leads to the immunosuppressive TME that promotes GB progression and correlates with patients’ worse outcomes [658]. Pro-inflammatory molecules (i.e., IFNs, IL-12 or TGFβ) and several pro-oncogenic transcription factors (HIF-1α, STAT3, EGFR or PTEN loss) have been identified as direct regulators of PD-L1 transcription [391,659].

Monoclonal antibodies targeting CTLA-4 (ipilimumab), PD-1 (pembrolizumab, nivolumab, and cemiplimab) or its associated programmed cell death ligand 1 (PD-L1, atezolimumab, durvalumab and avelumab), known as checkpoint inhibitors (ICIs), have been approved by the FDA for the treatment of various types of cancer [660]. Blockage of the interaction between PD-1 and PD-L1 may re-establish proper immunity against GB. Preclinical studies have confirmed this hypothesis and demonstrated a survival benefit in immunocompetent murine models of GB following dual radiation and anti-PD-1 treatment [661]. The results of the phase II/III trials testing ICIs on GB are presented in Table 4. None of them showed significant increases in mOS. Pembrolizumab was ineffective as monotherapy and combined with BEV for rGB [662]. CheckMate 143 (NCT02017717) compared nivolumab with BEV therapy in rGB patients. Although the follow-up of 12 months did not find significant differences between the two arms (42%), the objective response rate was significant higher with BEV (23.1%) vs. nivolumab (7.8%), disfavoring the experimental drug [663]. The addition of nivolumab to the SOC has been assayed in two clinical trials for ndGB patients with or without hypermethylated MGMT promoters (NCT02617589 and NCT02667587) without evidence of clinical benefits [664,665].

Two clinical studies have demonstrated enhanced T cell responses and a clinical benefit with neoadjuvant ICI treatment versus anti-PD-1 adjuvant administration in patients with rGB [693]. Patients in the neoadjuvant arm received pembrolizumab 14 ± 5 days prior to surgical resection and patients in the adjuvant arm did not. After surgical resection, both groups received pembrolizumab every 3 weeks. Compared to only adjuvant PD-1 blockade, neoadjuvant pembrolizumab confers significant improvement in OS (228.5 days versus 417 days) and PFS (2.4 months versus 3.3 months) [693]. A single-arm phase II clinical trial, utilizing neoadjuvant nivolumab in GB patients, demonstrated similar intratumoral and systemic immune changes (NCT02550249), even though survival advantages could not be demonstrated [683]. Although the pros and cons of starting neoadjuvant therapy are hotly contested in clinical trials [694], preclinical studies indicate the remarkable benefit of starting immunotherapy before surgery [695], which is also in agreement with other clinical assays evaluating the efficacy of nivolumab and neoadjuvant PD-1 blockade in melanoma, breast and lung cancers [696,697]. No improvement in PFS or OS (ISRCTN84434175) was observed with the addition of ipilimumab to SOC in ndGB patients

Co-suppression of CTLA-4 and PD-1 inhibited GB progression in mice models [698,699]. However, the anti-CTLA-4/PD-L1 treatment combination produced grade III-IV toxicities in GB patients, thus drawing into question its safety [700]. TTFields upregulate different immune checkpoints (e.g., PD-L1, CTLA-4, or TIGIT, T cell immunoglobulin and ITIM domain), which supports the possibility that combined treatment may increase the efficacy of immunotherapy [322,323,324]. In that sense, two phase II trials are evaluating possible synergistic effects combining SOC with TTFields plus pembrolizumab (NCT03405792), and nivolumab plus/minus ipilimumab (NCT03430791) in ndGB. The preliminary results of NCT03405792 seem promising because of evidence of an improved OS (24.8 vs. 14.7 months) in ndGB patients that received the complete cocktail treatment vs. controls [684].

A recent meta-analysis concludes that a postoperative combination of radiotherapy, chemotherapy and PD-1 inhibitors may increase antigen cross-presentation activity, promote tumor-lymphocyte infiltration, and increase the expansion of effector T cells. Furthermore, radiotherapy has a synergistic effect on immunotherapy by decreasing the number of tumor cells and upregulating the tumor expression of PD-L1, but evidence of survival benefits is scarce [701]. The limited efficacy of ICIs can be attributed to compensation of blocked PD-1 function by other checkpoint molecules (i.e., LAG-3) [702]; TAMs, which have been observed to remove anti-PD1 antibodies bound to CD8+ T cells [703]; and impaired epigenetics and memory cell formation, resulting in T cell exhaustion [704]. LAG-3 (also known as CD223) is expressed on activated human T (often co-expressed with PD-1), DCs and NK cells [705]. Its binding with the major histocompatibility complex (MHC)-II in APCs promotes apoptosis, decreases proliferation and increases T cell tolerance [705]. The combination of anti-LAG-3 and anti-PD-1 in a variety of tumor models has led to synergistic antitumor efficacy. Harris-Bookman et al. showed that LAG-3 is expressed in human and mouse GB samples, and either knockdown or anti-LAG-3 antibodies improved survival in a preclinical GB model and the efficacy of anti-PD-1 treatment [706]. Currently, a phase I trial (NCT02658981) is underway, testing anti-LAG-3 alone and in combination with anti-PD1 in rGB patients [707]. Ongoing research is exploring the potential of using ICIs in combination with other immunotherapies or targeted therapies to further enhance their effectiveness in treating GB.

### 5.2. Other Strategies to Avoid Immunosuppression in the GB TME

IDO1, IDO2 and tryptophan 2,3-dioxygenase (TDO2) are “rate-limiting” enzymes in the kynurenine pathway of Trp catabolism, [708,709,710]. The resulting Trp metabolites are involved in immune regulation, energy metabolism and the production of NAD^+^. Expression of IDO and TDO enzymes in mammals differs by tissue location and by their stimuli-dependent induction [711,712]. They are constitutively expressed in a restricted set of tissues, including placenta, mucosa, and lymphoid organs [713]. IDO1 expression is increased in many human tissues, and it is overexpressed in the majority of cancers (90% of glioma cells) by proinflammatory cytokines, such as IFN-γ and TNF-α, or in response to interaction with tumor-infiltrating T or NK cells [714,715]. IDO1 overexpression in glioma cells impedes an effective immune response through increased apoptosis of CD8+ T cells and by converting naïve T cells into inducible immunosuppressive Tregs [716]. TDO is primarily expressed in the liver and induced by Trp and corticosteroids [717]. Human IDO2 is a relatively inefficient Trp dioxygenase, and little is known about its link to cancer, although different pro-inflammatory stimuli induce its expression in melanoma, pancreatic, gastric, and brain cancers [712]. TDO and IDO2 show higher tissue specificity and much lower enzyme activity than IDO1 [718]. A multitude of cancer cells constitutively express or upregulate IDO1, TDO2, or both, and coerce stromal and tumor-infiltrating immune cells to express IDO1, thus supporting evasion of immunosurveillance [710,719,720]. Increased expression and a high kynurenine/Trp ratio in the peripheral blood have been associated with cancer progression, poor prognosis and lower survival in GB and other cancer patients [721].

Although GB cells typically do not express IDO, it is activated when GB cells are identified by TILs or NK cells and exposed to important anti-cancer cytokines such as IFN-γ and TNF-α. Moreover, IDO1 expression is elevated in GSCs compared to GB cells [722,723]. IDO1 overexpression increases the intratumoral accumulation of immunosuppressive Tregs and decreases OS in experimental mice with brain tumors [708,716,724,725,726]. Compared to healthy controls, GB patients showed decreased serum Trp levels and, surprisingly, reduced Trp metabolite levels. High tumor volume is associated with low systemic metabolite levels, low systemic kynurenine levels and worse OS [727]. Nevertheless, real-time measurements of IDO1 activity in GB patients is still a follow-up strategy = pending development.

Cancer cells require Trp to produce energy, and when its availability is low, they acquire motility in order to search for alternative sources of energy. Trp deprivation inhibits proliferation and induces apoptosis in DCs and T cells through the activation of the serine/threonine-protein kinase GCN-2 (general control non-derepressible 2) [728,729]. GCN-2 activation also inhibits fatty acid production in naïve T cells, which is necessary for their proliferation and activity [730]. However, in vivo, Trp levels show very low fluctuations in the brain [731], and recent studies have questioned the immunosuppression attributed to local Trp starvation [732], thus suggesting that accumulation of kynurenines-associated aryl hydrocarbon receptor (AHR) activation could be the main mechanism involved in the IDO- or TDO2-induced immunosuppression [710,733,734]. In fact, AHR expression is associated with immunosuppression in human tumors and AHR blockade reverses the IDO/TDO-mediated immunosuppression [720]. AHR activation by Trp catabolites promotes TGFβ production in GB [735], generates immune-tolerant Treg cells and DCs, and suppresses anti-tumor immunity [736]. AHR promotes TGFβ production in GB [735], induces immune checkpoints such as PD-L1 [737], and induces immunosuppression via its effects on T cells [738], DCs [739], macrophages [739], B-cells [740], or GB-specific TAMs [741], thereby contributing to the generation of an immunosuppressive milieu [742]. Moreover, Trp metabolites like quinolinic acid, 3-hydoxyanthranilic acid, and 3-hydroxykynurenine have been shown to induce apoptosis in Th1 helper cells, CD8+ effector T cells, and B cells, while sparing immunosuppressive Th2 helper cells [743]. Dysregulation of kynurenine signaling promotes DNA damage, is associated with brain edema formation (which facilitates invasion and motility of cancer cells) and induces NAD^+^ generation in glioma cells (a mechanism involved in the development of resistance to therapy) [744]. Interestingly, cancer immune suppression was reversed by administration of PEGylated kynureninase that degrades kynurenine into nontoxic and immunologically inert metabolites [745].

Proteolysis targeting chimera (PROTAC) technology has been used in the development of molecules capable of targeting IDO1 [746], and to date, several IDO1 inhibitors (PF-06840003, epacadostat, indoximod, navoximod, BGB-5777, 1-methyl-tryptophan, BMS-986205, GDC-0919) are under clinical trials for GB therapy [724,746,747]. Although the blockade of IDO as monotherapy has not shown any effect on OS in GB patients, IDO inhibition can enhance the response to immunotherapy. In a murine preclinical model of GB, combined treatment with anti-IDO1, anti-CTLA-4, and anti-PD-1 showed synergistic effects, increasing the antitumor effect compared to monotherapies [748]. The combination of an IDO1 inhibitor (BGB-5777 or 1-methyl-tryptophan), PD-1 blockade and irradiation significantly increased OS and resulted in long-term tumor control in 30–40% of mice with advanced GB [749,750]. Furthermore, radiotherapy response can be enhanced by GDC-0919 by reducing radiotherapy-induced immunosuppression [129], whereas IDO inhibitors have been shown to synergize with TMZ and radiation therapy [751]. In mice, PF-06840003 reduced intra-tumoral kynurenic levels and inhibited tumor growth in both monotherapy and, with increased efficacy, in combination with anti-PDL-1 antibodies [752]. Supported by these preclinical data, a phase I multicenter clinical trial (NCT02764151) showed that PF-06840003 was generally well tolerated, had a pharmacodynamic effect and had a durable clinical benefit in a subset of glioma patients [753]. In mice, epacadostat generated a potent immune response by altering the abnormal signaling pathways of cancer cells and reducing kynurenine levels by 90% in both plasma and the tumor tissue [754], and also enhanced the efficacy of TMZ and ICIs improving the OS [755]. Epacadostat combined with anti-PD-1 antibodies (pembrolizumab or nivolumab) improved the response rate and PFS (>6 months) in patients with gastric cancer [756], but in GB patients, non-significant improvements in OS were observed (NCT02327078). Preliminary results of a phase II study (NCT03532295) in rGB with retifanlimab (PD-1 inhibitor) plus or minus epacadostat in combination with BEV and hypofractionated radiotherapy suggests that treatment is well tolerated and had encouraging OS and PFS at the time of data cutoff [685].

Interestingly, advanced age is associated with an increase in brain IDO1 expression, and this is not reversed by treatment with IDO1 inhibitors [757], which can contribute to explaining the decreased survival of older GB patients during treatment with immune checkpoint blockade therapy.

### 5.3. Vaccines

Vaccines targeting cancer have two types of antigens: tumor-associated antigens present on health tissues and overexpressed in cancer cells, and tumor-specific antigens (TSA, earlier described as “neo-antigens”) present only on cancer cells. Obviously, vaccines based on TSA antigens are more selective and effective, but in highly heterogeneous tumors such as GB, it is difficult to find them. The lack of specificity and high expression of epitopes in GB lead to autoimmunity and side effects such as brain inflammation, which limits the tolerability of vaccine-based strategies [758,759]. Nevertheless, here, we discuss the results obtained with the development of cellular (tumor cell and DC) and non-cellular (peptide) vaccines.

#### 5.3.1. Tumor Cell Vaccines

Early vaccines utilized dead or inactivated tumor cells, with very limited success. To improve the efficacy, gene editing of tumor cells began in the late 1980s, involving the expression of certain immune-stimulating cytokines, granulocyte macrophage colony-stimulating factor (GM-CSF) being one of the most studied [760]. Phase I trials were completed using the latest generation of autologous and allogeneic tumor cell lines secreting GM-CSF (e.g., K-562, lymphoblast cells isolated from the bone marrow of a 53-year-old chronic myelogenous leukemia patient). The effectiveness of vaccination depends on T cell activation and anti-tumor immunity [761]. Additionally, research into the direct injection of formalin-fixed GB as an antigen for treating GB has been studied. A phase I/IIa trial of fractionated radiotherapy, TMZ and autologous formalin-fixed tumor vaccine (AFT) showed a favorable PFS and OS in ndGB patients [675]. IGV-001 consists of autologous GB cells that are incubated with an antisense oligodeoxynucleotide (IMV-001) targeting IGF-1R that is implanted in patients’ abdominal for approximately 48 h, as opposed to other autologous cancer vaccine modalities, which require multiple dosages over weeks. The results are promising because ndGB patients had a PFS of 9.8 months compared to 6.5 months in historical controls receiving SOC [762]. There is a phase IIb trial (NCT04485949) opened for accrual for ndGB patients.

#### 5.3.2. Dendritic Cell Vaccines (DCV)

DC vaccination has gathered considerable attention after some encouraging reports showing acceptable efficacy and safety levels. DCs are powerful APCs capable of inducing acquired and innate immunity responses. DC vaccine preparation involves the isolation of CD-14 positive monocytes from the patient, loading of tumor antigens into the immature DCs, treatment of the DCs with the cytokines (GM-CSF and IL-4) to induce maturity, and finally the preparation of human DC vaccines for re-injection into GB patients [763,764,765].

In a phase I/II prospective non-controlled clinical trial, 37 patients harboring GB or grade 4 astrocytoma received monthly intradermal injections of allogenic DC vaccine, starting at the time of first recurrence after surgery. Compared with patients in the Genomic Data Commons data bank, OS for vaccinated GB patients was 27.6 ± 2.4 months (vs. 16.3 ± 0.7), and it was 59.5 ± 15.9 for vaccinated astrocytoma grade 4 patients (vs. 19.8 ± 2.5). Seven vaccinated patients (two IDH-1-mutated and five wild-type) remained alive at the time of the Lepski et al. report [666].

Results of phase II/III trials on DCV in GB are presented in Table 4. ICT-107 is an autologous DC vaccine targeting six tumor antigens (MAGE-1, HER-2, AIM-2, TRP-2, gp100, IL-13Rα2) specifically overexpressed in glioma stem cells. Patients receiving ICT-107 (NCT01280552, Table 4) reported no significant increases in mOS (17 vs. 15 months) [681]. Yao et al. assayed a DCV loaded with GSC antigens, and results seem to evidence a slight increase in mOS (13.7 vs. 10.7 months) when compared with the placebo control [672]. Northwest Biotherapeutics developed the lysate-loaded dendritic cell vaccine (DCVax^®^-L) as an adjunct for the treatment of GB [766]. Recently, Liau et al. reported impressive results of nonrandomized phase III clinical trial (NCT00045968) ndGB and rGB patients treated intradermally with DCVax^®^-L plus SOC vs. contemporaneous matched external control patients treated with SOC [671]. The mOS of the 232 ndGB patients treated with DCVax^®^-L was 19.3 vs. 16.5 months in controls. Survival at 48 months was 15.5% vs. 9.9%, and at 60 months 13.0% vs. 5.7% in the treated and control arms, respectively. For 64 patients with rGB, mOS was 13.2 months from relapse in the DCVax^®^-L group vs. 7.8 months in the external control cohort [671]. Despite these promising results, the study design received criticisms [767,768], e.g., the criteria adopted for recruiting GB patients did not consider the latest WHO classifications [253] and there was a lack of evaluation of IDH mutations. Consequently, it is possible that treated patients with longer survivals might have less aggressive GBs [767,768]. In fact, other randomized phase II clinical trials (Table 4, NCT01280552, NCT01213407, NCT03400917) have not shown similar survival benefits [667,669,681]. Compared to the peptide vaccine, the DC vaccine ensures high antigen-presenting efficiency with sufficient exogenous costimulatory signals, whereas DC maturation may be impeded by insufficient costimulatory signals in patients with a weak primary immune status. This can be one explanation for such different results in clinical trials.

Cytomegalovirus is a herpes virus with a high detection rate in many primary or metastatic brain tumors, including GB, whereas it is not detected in surrounding normal brain tissue [769]. Cytomegalovirus has oncogenic features, and its presence in gliomas is linked to the increased production of PGE2, IL-10 and B7-H1. PGE2 activates cell proliferation and angiogenesis, inhibits apoptosis, and activates invasion, and promotes the formation of the TME [770]. IL-10 recruits M2 macrophages that contribute to TME immunosuppression and B7-H1 enhances tumor stem cell migration [771]. Many studies have suggested that anti-cytomegalovirus therapy can restrain glioma progression, and for this purpose, valganciclovir (inhibits viral DNA duplication) [772,773], dendritic cell vaccine and adoptive cytomegalovirus-specific T cell therapy (NCT02661282) [774] have been attempted in GB treatment [770], with promising results. Valganciclovir treatment is well tolerated and, in combination with SOC, significantly increased ndGB patient survival (24.1 months vs. 13.1 months in patients with shorter treatment duration and 13.7 months in the control group) [775], and survival of rGB patients with unmethylated and methylated MGMT promoter genes [776]. Cytomegalovirus-DC vaccines usually target the pp65 antigen. After the radio/chemotherapeutic treatment for ndGB, this vaccine was combined with dose-intensive TMZ and GM-CSF, resulting in a significant increase in survival (Table 4) [670]. A second trial (NCT00639639) compared the effects of tetanus toxoid preconditioning and a DC vaccine with standard TMZ, and resulted in a 36% survival rate at 5 years from diagnosis [777,778]. According to the results of these trials, nearly one-third of the GB patient population receiving cytomegalovirus-specific DC vaccines showed exceptional long-term survival. However, these studies have some limitations: a) consolidative results for the presence of cytomegalovirus within GB are needed; b) the onco-modulatory role of cytomegalovirus for gliomagenesis is not confirmed; and c) it is necessary to confirm positive clinical outcomes in larger and randomized studies [779].

The addition of tumor lysate-pulsed autologous DCs vaccination to tumor resection combined radio-chemotherapy was feasible and safe according to a recent phase II trial (NCT01006044). An increase in tumor-specific immune response was shown in 11/27 evaluated patients, but no correlation between immune response and survival was found [668]. No increase in OS was shown. A new phase II randomized clinical trial (NCT03395587) is now ongoing for ndGB patients [780].

A recent systematic meta-analysis review concludes that adjuvant DC vaccine is feasible, overall toxicity is limited and anti-tumoral cytotoxic responses have been demonstrated. Regimens including DC vaccines led to a significantly longer 1-year and 2-year OS in GB patients after the induction of an immune response (which is not always observed) [781], thus suggesting the need for more time to achieve an anti-GB immune response. The results of these trials robustly support the continued investigation and development of DCV as a treatment for GB.

#### 5.3.3. Peptide Vaccines

Protein/peptide variations derived from mutated genes are distinct to tumor cells and absent in normal cells. As a result, they can serve as TSA to induce immune responses against tumor cells. While only a few mutations are transformed into new epitopes, their presentation by APCs on the human leukocyte antigen (HLA) can lead to T-cell-based immunity. However, many potential tumor antigens do not originate from mutations but arise due to errors or overexpression of normal proteins also present in other tissues. Targeting such antigens may lead to autoimmunity, resulting in unintended effects like brain inflammation [782]. The lack of specificity and the high expression of epitopes in GB pose challenges in the development of peptide vaccine-based strategies. Vaccine adjuvants refer to immunostimulatory components, given in addition to the antigen, to enhance the durability and magnitude of the immune response.

IMA950 is an example of this type of vaccine, containing 11 tumor-associated peptides designed to activate specific T cells. It consists of nine HLA-A2-restricted peptides derived from the surface of GB tumor samples and two HLA-isotype DR-binding peptides that facilitate the CD4+ T cell response. One advantage of using pre-set antigens in the vaccine is that it opens up the possibility of neoadjuvant usage, allowing treatment before tumor samples (through biopsy or surgical resection) become available. To induce the CD8+ T cell response, the vaccine is combined with polyinosinic-polycytidylic acid stabilized with polylysine and carboxymethylcellulose (poly-ICLC), an immunogenic molecule (NCT01920191). In this trial, a multipeptide CD8+ and sustained T helper-1 CD4+ T cell response was observed. For the entire cohort, CD8+ T cell responses to single or multiple peptides were found in 63.2% and 36.8% of patients, respectively. OS was 19 months for GB patients [783]. A randomized phase I/II trial called IMA950-106 (NCT03665545) for rGB is assessing IMA950/Poly-ICLC in combination with pembrolizumab.

Liu et al. characterized the immunogenomic landscape of the aggressive orthotopic GB CT2A model and tested the efficacy of a CT2A-specific neoantigen vaccine combined with an anti-PD-L1 blockade combination [784]. In mice bearing CT2A tumors, the neoantigen vaccine alone did not affect median survival, while immune checkpoint blockade alone only slightly prolonged median survival from 17.5 to 25 days. However, the combination resulted in a 60% long-term survival, thus suggesting that the immune checkpoint blockade facilitates the clonal expansion and activity of neoantigen-specific T cells [784].

In the GAPVAC trial, 16 ndGB patients received two synthesized vaccines: one targeting neoantigens (APVAC2) and the other unmutated peptides (APVAC1), both applied independently by intradermal injections during TMZ therapy. Both vaccines elicited CD8+ and CD4+ T cell responses, inducing 84.7% (APVAC2) and 50% (APVAC1) immunogenicity, and mPFS and OS were encouraging [785]. SurVaxM is a peptide vaccine that is designed to activate the immune system against survivin, which is highly expressed by GB cells. The addition of SurVaxM to the SOC (NCT02455557) appeared to be safe and well tolerated and seemed to have clinical benefits in both methylated and unmethylated ndGB patients [689].

Heat shock protein (HSP) binds TSAs. HSP–peptide complexes can be taken up by antigen-presenting cells and then trigger both innate and adaptive antitumor responses [786]. Therefore, after a simple purification of HSP–peptide complexes from a patient’s tumor, these complexes can be directly administered as a personalized polyvalent antitumor vaccine. HSP peptide complex-96 (HSPPC-96) is the most widely used to treat gliomas [787]. A trial in rGB patients tested this hypothesis and showed an increase in the mOS, a specific peripheral immune response to the peptides bound to HSPPC-96, a focal cellular infiltrate of CD4+ and CD8+ and, in addition, brain biopsies were consistent with specific immune responses at the tumor site [678,679]. A following phase II trial (NCT01814813) failed to demonstrate a survival benefit for patients treated with HSPPC-96 alone or in combination with BEV compared to BEV alone [680]. HSPPC-96 may improve survival for ndGB patients when combined with SOC (NCT02122822) and warrants further study [788].

Up to now, therapeutic vaccines have generally yielded limited efficacy, despite their theoretical basis [360,789]. Combining diverse antigens, including tumor-associated antigens, neoantigens, and pathogen-derived antigens, and optimizing vaccine design or vaccination strategy may help with clinical efficacy improvement [790,791].

### 5.4. CAR T-Cell Therapy

CAR-T cells are genetically modified T cells (derived from T cells of a patient’s own blood) that express an extracellular TSA recognition domain (CAR, chimeric antigen receptor) and an intracellular activation domain to keep T cells constitutively active. The union between a CAR-T cell and a tumor cell triggers a strong cytotoxic response with the additional release of different proinflammatory cytokines serving as a defense against the immunosuppressive TME [792,793]. Recent studies show that multiple intratumoral (or intraventricular) infusions enhance the persistence of CAR-T cells, their cytotoxic efficacy and the durability of tumor control [794]. Moreover, CAR-T cells can cross the BBB, providing the additional advantage that can be parenterally administered. In a preclinical model of orthotopic GB, a complete antitumor response was achieved when an i.v. dose of anti-GD2 CAR-T cells was administered after focal irradiation. Intravital microscopy images successfully visualized the localization of CAR-T cells and confirmed apoptosis of the tumor cells. According to the authors, radiotherapy allows rapid extravasation and intratumorally expansion of CAR-T cells, leading to a more aggressive and durable immune response [795]. Interestingly, the cytotoxic activity of CAR-T cells is unaffected by TTFields, which provides possible compatibility between both treatments [322].

Six CAR-T cell products have been approved by the FDA for 12 different indications, but none include the GB [796]. Several clinical trials are examining the efficacy of CAR-T cell treatment for GB, being the most tested targets HER2, EGFRvIII, IL-13Rα2, and CD70 [794,797]. The expression of human HER2 plays a crucial role in cell proliferation and motility in cancer cells, and its overexpression in GB is usually associated with an unfavorable prognosis [798]. One potential strategy to optimize the persistence of adoptively transferred T cells relies on the expression of CARs in virus-specific T cells (VSTs) [799]. HER2-CAR–VSTs cells persisted after infusion for up to 12 months in 17 HER2-expressing GB patients. HER2-CAR VST treatment was feasible and safe, and induced transient tumor reduction and/or tumor necrosis effects that resulted in a clinical benefit in 8 of 17 treated patients [800]. Despite this encouraging result, and considering the expression of HER2 in some important organs, the safety of HER2-directed drugs still needs to follow more strict experiments before it is widely used in clinics [801].

IL13Rα2 is rarely expressed in healthy brain tissue and is overexpressed in up to 50–80% of GBs, being associated with poor prognosis [802,803]. Based on preclinical promising results in murine models, a pilot study assayed the efficacy of IL-13Rα2-directed CD8+ cytotoxic T cells (NCT00730613) following tumor resection in three rGB patients [804]. The results of the study demonstrate the feasibility of repetitive intracranial delivery of autologous CART cells via an implanted reservoir/catheter system, showing no serious adverse events. Despite transient regression of the tumor, along with corresponding increases in cytokines and immune cell levels in the cerebrospinal fluid, patients died, most likely due to downregulation of the target antigen IL13Rα2 [804]. One of the patients initially showed a complete remission response after infusion, but had a recurrence 7.5 months later [805].

GD2 is a disialoganglioside overexpressed on tumors of neuroectodermal origin, and especially in GSCs [806]. Animal studies have shown that CAR-T cells against GD2 can effectively eliminate GD2-positive human GB tumors implanted orthotopically in mice without obvious neurotoxicity or off-target effects [807]. GD2-specific fourth-generation chimeric antigen receptor 4SCAR-T cells have been evaluated in a phase I clinical trial assay (NCT03170141) and both single and combined infusions of GD2-specific 4SCAR-T cells were safe and well tolerated in GB patients, with no severe adverse events [805]. More recently, Liu et al. developed a novel combined immunotherapeutic using genetically engineered PBMC-derived induced neural stem cells (iNSCs) expressing HSV-TK and second-generation CAR-NK cells against GB. PBMC-derived iNSCs^TK^ possessed tumor-tropism migration and exhibited considerable anti-tumor activity in the presence of ganciclovir. In addition, combined with GD2NK92, the therapeutic efficacy of iNSCs^TK^ improved the tumor-bearing animal model’s median survival [808].

CARs require higher antigen density for full T cell activation. Hence, the main mechanisms involved in the disease relapse after CAR-T therapy include downregulation of target antigen in response to therapy, inadequate T cell potency, intratumoral heterogeneity, incomplete antigen coverage, upregulation of suppressive ligands (Fas, PD-L1, and other checkpoint molecules), downregulation of anti-inflammatory cytokines (IL-10 and TGF-β), an immunosuppressive tumor microenvironment, and secondary effects of palliative treatment with corticosteroids [809,810,811,812]. CAR T-cell therapy is relatively safe but is not exempt from complications. Neurotoxicity is not an infrequent, and potentially fatal complication. The spectrum of manifestations ranges from delirium and language dysfunction to seizures, coma, and fulminant cerebral edema [813]. Toxicity is related to the location of the tumor and may be reversible with intensive supportive care [814].

The expression of inhibitory immune checkpoints, e.g., PD-1, on CAR-T cells has been associated with a remarkable decrease in their ability to target tumor cells. In this regard, two approaches have been proposed to repress PD-1 expression, i.e., co-administration of anti-PD-1 monoclonal antibodies and CAR-T cells, and PD-1 gene editing in the CAR-T cells [815,816,817]. Work is underway to combine CAR-T cell therapies with immunomodulators designed to maintain the activity of immune cells or to resist specific immunosuppressive mediators. Currently, therapies are being developed using CAR-T cells and multiple antigens (such as B7-H3, CD147, GD2 or MMP2) and a new approach is studying the efficacy of combining EGFR CAR-T with pembrolizumab in rGB [818].

### 5.5. CAR NK-Cell Therapy

CAR NK-cell therapy offers a different approach to targeting cancer. Firstly, NK cells can recognize tumors composed of heterogeneous cells by employing multiple activating and inhibitory receptors, even when MHC class I (MHC-I) molecules are diminished or absent [819]. Secondly, NK cells play a crucial role in attracting conventional type 1 DC and subsequently CD8+ T cells [820,821]. These functions facilitate the activation of the cancer immunity cycle, which offers the advantage of overcoming the immunosuppressive GB tumor microenvironment [822].

NK cells play a key role in eradicating tumor cells that reduce their surface expression of MHC-I. However, the interaction between MHC-I and killer cell immunoglobulin-like receptors can inhibit NK cell function, leading to reduced destruction of healthy cells [823,824]. Due to these advantageous properties, numerous preclinical and clinical trials have explored the effectiveness of CAR NK-cell-based immunotherapy against GB, utilizing NK cells as either autologous or allogeneic therapy [825]. Allogeneic NK cells offer potential benefits for immune-suppressed patients as they exhibit greater anti-tumor capabilities against GB when compared to exhausted NK cells derived from patients’ peripheral blood mononuclear cells [826]. Cord blood-derived mononuclear cells also hold promise as a valuable source for allogeneic NK cell therapy.

Different approaches utilizing CAR NK-cell therapy include adoptive NK cell therapy (autologous and allogeneic therapy which implies obtaining the patient’s cells and tissues, expanding them ex vivo, and re-infusing them back into the patient), CAR-NK cell therapy (using NK cells engineered to express activating CARs), checkpoint blockade therapy (e.g., combination therapy of NK cells activated by IL-2 and TDK derived from HSP70, and anti-PD1 antibody), and gene editing NK therapy (e.g., the CRISPR/Cas9 genetic-editing system widely utilized to genome-edit T cells to disrupt inhibitory genes such as PD1 and CTLA4) (see [825] for a recent review). CAR-NK cells have advantages in safety profile compared with CAR-T cells, including reduced risk of graft-versus-host disease, cytokine release syndrome, and neurotoxicity [827]. Based on the above considerations and taking into account the inter- and intratumoral heterogeneity of GB, along with its immunosuppressive microenvironment, CAR NK-cell-based immunotherapy appears a suitable approach for addressing GB. In this, it is remarkable the pivotal role of NK cells in inducing anti-tumor responses in CD8+ T cells and other immune cells.

### 5.6. Multiple Immunotherapeutic Approaches against EGFR

EGFR is also a prevalent molecular target in immunotherapy assays. Rindopepimut (also known as CDX-110) is a small peptide that mimics the mutated region of EGFRvIII and was designed to generate humoral and cytotoxic T cell responses against GB cells expressing this mutation [758]. The combination of rindopepimut with SOC achieved 24-month OS (phase II, NCT00458601), with 50% of patients developing a humoral immune response to the vaccine. Moreover, 82% of the treated patients showed a loss of EGFRvIII expression in tumor recurrence, indicating that the vaccine may be effective in targeting this variant [687]. However, the phase III study (NCT01480479) was interrupted because it did not confirm an increase in mOS [688]. Rindopepimut + BEV was clinically tested (phase II, NCT01498328) to treat EGFRvIII-positive rGB [686]. Rindopepimut-treated patients achieved robust anti-EGFRvIII titers, and the combined treatment significantly enhanced PFS6 (27% vs. 11%) and mOS (12 vs. 8.8 months) [686]. A recent meta-analysis explored the efficacy and safety of various BEV combination regimens in patients with rGB and concluded that BEV + rindopepimut therapy seems to be safer and more effective [828].

In a phase I trial with DC vaccines targeting EGFRvIII, an immune response was demonstrated with in vitro antigen-specific T cell proliferation and in vivo delayed-type hypersensitivity responses, without significant adverse effects. Subsequently, trials have been planned with polyvalent DC vaccines (directed to numerous GB antigen epitopes) that have demonstrated safety despite the higher risk of autoreactive T cell selection [829].

In C57BL/6 mice, 806-28Z CAR-T cells were able to lyse GL261/EGFRvIII cells in a dose-dependent manner, even eradicating xenografted tumors at high doses. Cell dose and granzyme B release were crucial determinants for CAR-T efficacy, even in heterogeneous GB tumors [830]. A phase I study (NCT02209376) administered CAR-T cells to 10 rGB patients via i.v. infusion. One of the patients remained alive for over 18 months without receiving additional therapy. Nonserious toxicities were observed and trafficking of CART-EGFRvIII cells to regions of active GB was shown in seven treated patients, but with a subsequent antigen decrease in five of them accompanied by an increased and robust expression of immunosuppressive markers such as FOXP3 and PD-L, and infiltration by Tregs, which pointed out the induced adaptive resistance to CART-EGFRvIII cells [831]. Mice treated with dose-intensified TMZ, before administration of CAR-T targeting the EGFRvIII antigen, survived for a median of 174.5 days, compared to 69.5 days in mice pretreated with standard doses of TMZ. On this basis, a phase I trial (NCT02664363) in patients with ndGB incorporating dose-intensified TMZ, as a preconditioning regimen prior to CAR immunotherapy, was initiated in 2018 [832].

In addition, strategies assessing co-administration of vaccines and CAR-T engineered with EGFRvIII and PD-1 are being assayed (NCT04003649, NCT04201873, NCT02529072, NCT02287428). Tandem CAR-T cell targeting simultaneously EGFRvIII and IL-13Rɑ2 is now in preclinical investigation, with an activity superior to their monospecific CAR counterparts in heterogeneous GB cell populations [833]. Bispecific T cell engagers (BiTEs) are bispecific antibodies that redirect T cells to target antigen-expressing tumors. BiTE-secreting T cells have been proposed as a valuable therapy in solid tumors, with distinct properties in mono- or multivalent strategies incorporating chimeric antigen receptor CAR-T cells [834]. Combining BiTEs with CAR-T cell therapy has been tested in preclinical models to avoid antigen-negative relapse. Choi et al. [34] used T cells transduced with an EGFRvIII-targeted CAR construct co-expressing BiTEs against EGFR. These BiTEs bound to CD3 on T cells and EGFR simultaneously, allowing the CAR-T cells to target tumor cells expressing EGFRvIII, EGFR, or both. Although EGFRvIII-targeted CAR-T cells were unable to thoroughly treat GB tumors heterogeneous for EGFRvIII expression, CAR-T cells secreting BiTE molecules were able to eliminate murine GB models after their intracranial administration [805].

### 5.7. Viral Therapies

GB is particularly suitable for viral therapy to overcome TME immunosuppression and growth being surrounded mainly by post-mitotic cells, which allows the use of viruses that require active cell cycles for replication [835]. Viruses used to treat tumors can be divided into two categories: (1) oncolytic viruses (OVs) with the natural capacity to replicate only in cancer cells (reoviruses, and Newcastle disease viruses) or that are vulnerable to genetic manipulation that increases their tumor selectivity (adenoviruses, herpes simplex viruses, vaccina viruses, polioviruses, and measles viruses); and (2) viral vectors with a low replication rate, which are used as vehicles for other therapeutic genes [836]. Any of the viral categories can stimulate an immune response without the expression of an immunomodulatory transgene. This is most notable for OVs, which cause lytic tumor cell death and induce the release of tumor-associated antigens and damage-related molecular patterns that lead to the activation of innate and adaptive immune responses [837]. Immunogenic cell death is characterized by an immune response that indirectly kills the cancer cells using different mechanisms, i.e., apoptosis, necrosis, and autophagy [838]. Moreover, to potentiate anti-cancer immunity, some OVs have been genetically engineered to express different cytokines (GM-CSF, IL12, and IL15), to enhance antigen presentation, or to elicit a more effective immune response against cancer cells [839,840]. The immunostimulatory ability of OVs allows the transformation of “cold tumors” (i.e., those with an immunosuppressive microenvironment) into immune-responsive “hot tumors” [841], and, as a consequence, OVs function not only as direct cancer-killing agents but also as active anticancer vaccines [842].

Genetic modification can have other purposes, such as to enhance the safety of OVs, to improve the tropism of OVs to cancer cells, and to enhance the anti-tumor effect of OVs [843]. For instance, the genetically attenuated and modified ZIKV strain selectively targets GSCs and exhibits good oncolytic activity [844]. Engineered OVs can also introduce specific therapeutic genes including tumor-suppressor genes (i.e., p53, [845]), anti-angiogenic genes [846], and immunostimulatory genes [847], or induce PI3K inhibition [848,849] in GB cells to execute their expression. Administration of the virus must be efficient and safe for successful therapy. The challenge of transporting viruses to the CNS, crossing the BBB, involves potential elimination by the immune system and also limitations on the dose of viruses [850]. The ability to administer a virus systemically could improve efficacy by reaching distant areas where the tumor has spread, although, in the case of GB metastases, these rarely appear [851]. To circumvent the BBB, other promising techniques for drug delivery have been designed such as convection-enhanced delivery (CED) that uses a pressure gradient in a catheter to transfer therapeutic compounds to the interstitial spaces of the CNS [852,853]. OVs can also be administered intra-arterially and increase BBB permeability with focused ultrasounds [854] or using an osmotic agent such as mannitol [855] (see also Figure 4).

Due to the high transfection efficiency and the development of vector engineering techniques, OVs have been widely used and showed promising results in preclinical and clinical studies in GB (see [843,856,857] for a recent review). In addition, OVs have many advantages over conventional immunotherapies, including precise targeting, effective killing rates, and minimal adverse reactions [858]. To date, several clinical trials have been carried out for oncolytic viruses to improve the treatment of GB [836,843,857]. Examples include adenovirus [859,860,861], herpes simplex virus [862,863,864], reovirus [865,866], parvovirus [867], measles virus [868,869,870], poliovirus [853,871], vaccinia virus [872] and Newcastle disease virus [873]. In addition, other ongoing or completed trials have used the following: (1) modified HSV constructs such as G207 (NCT00028158, NCT03911388 and NCT02457845), HSV-1716 (NCT02031965), MVR-C252 (NCT05095441), M032 (NCT02062827), C134 (NCT03657576); and (2) genetically engineered oncolytic adenovirus combined with SOC or immune checkpoint blockade (NCT02197169, NCT01956734, NCT03896568, NCT01582516, and NCT02798406) [874].

Impressive results have been obtained in a phase II trial (UMIN000015995) in which 19 patients with residual or rGB received G47∆ (a triple-mutated, third-generation oncolytic herpes simplex virus type 1) intratumorally for up to six doses. The 1-year OS rate was 84.2%, the mOS 20.2 months after G47∆ initiation and 28.8 months from the initial surgery. Biopsies revealed an increased number of tumor-infiltrating CD4+/CD8+ lymphocytes and a decrease in immunosuppressive Foxp3+ cells. This study led to the approval of G47∆ as the first oncolytic virus in Japan [676]. The immunostimulatory effects of OVs make them excellent immune adjuvants to enhance chemo- and immunotherapy for GB [875]. Intratumoral DNX-2401 (a tumor-selective oncolytic adenovirus) followed by pembrolizumab was safe and had a notable survival benefit in rGB patients (NCT02798406) [673,674]. There is also an additional ongoing phase I study testing DNX-2401 and TMZ combination for rGB patients in Spain (NCT01956734). A live attenuated poliovirus type 1 vaccine (PVSRIPO) has demonstrated promising outcomes in a phase I study (NCT01491893) involving patients with rGB (20% of long-term survivors) [853]. Currently, PVSRIPO is undergoing further investigation in phase II trials for grade IV malignant glioma (NCT02986178) and in combination with pembrolizumab (NCT04479241) for GB [876].

Despite the promising results of some of the trials presented in Table 4 [673,674,676,689,691], others failed to demonstrate any benefit in ndGB [677] or rGB patients [690]. The discrepancy in results can be attributed to a different replication capacity in different hosts, the concomitant or previously administered treatments, or the corticoid coadministration that can limit the immune response. A recent comprehensive analysis conclude that virotherapy has a satisfactory safety profile and can improve the 2- and 3-year survival rates compared with non-virotherapy (2-year survival: 15% vs. 12%; 3-year survival rate: 9% vs. 6%) [877]. Current viral strategies focus on oral treatments and combination therapies [878,879].

### 5.8. Immunity-Related Adverse Events

ICIs to elicit anti-tumor response can be accompanied by activation of non-specific immune reactions against antigens expressed by normal tissues. The main immune-related adverse events include diarrhea, colitis, hepatitis, skin toxicities (pruritus, mucositis, and maculopapular rash) and endocrinopathies such as hypophysitis and thyroid dysfunction [880]. To increase the beneficial effects of immunotherapy, it is crucial to detect them as soon as possible and, at the same time, develop interventional strategies to control their severity [881].

The side effects of CAR-T treatment for rGB are still acceptable and most patients only suffer from transient discomfort [801]. One of the most important risks associated with CAR T cell therapy is on-target off-tumor toxicity. With the exception of EGFRvIII, all of the GB antigens that are currently being evaluated clinically may be expressed at low levels in normal tissues, which still can result in substantial toxicity [882,883].

The application of immunotherapy to GB implies a substantial concern for immune-associated CNS neurologic complications. Acute neurologic complications associated with oncolytic viruses and other immunotherapy treatments can often be more directly linked to their intratumoral administration than to a secondary immune-response-related adverse event [813]. While serious immune-mediated adverse events are rare overall, sometimes it is difficult to distinguish immune-mediated adverse events from disease progression. iRANO provides guidelines that can be applied to provide consistent metrics in clinical trials as well as daily practice [54].

## 6. Nanotherapies

Invasive (local administration) and noninvasive tools to deliver drugs are continuously evolving to overcome the BBB [23]. As reviewed in previous works [884,885,886,887], several nanostructures, including polymeric nanoparticles (NPs) (e.g., dendrimers, polymer micelles or nanospheres), inorganic NPs (e.g., silica, iron, gold or graphene NPs), lipid-based NPs (e.g., liposomes, emulsions), nanogels, carbon dots and nano-implants, have been developed as drug delivery systems and potential diagnostic agents for GB over the past decades. These elements can contain active anti-GB agents, such as chemotherapeutic/anti-angiogenic drugs, radio or chemosensitizers, or immune cells along with moieties that specifically target GB cellular receptors/angiogenic blood vessels or facilitate opening of the BBB [888,889].

Nanosystems (under 200 nm) may readily cross the BBB and fenestrated arteries, formed during the angiogenesis process, and accumulate within the tumor. This accumulation may be facilitated due to the weak lymphatic drainage system surrounding the tumor [887] or actively through the addition of targeting moieties to the surface of the NP [890]. The surface charge also plays a significant role since the electrostatic interaction between positively charged NPs and the negative surface charge of the BBB endothelial cells facilitates NP internalization through adsorptive-mediated endocytosis [891]. However, positively charged NPs can induce the generation of reactive oxygen species (ROS), which elevates their toxicity and restricts the in vivo efficacy [892]. Moreover, nanocarriers can also undergo dispersal throughout the brain and cause damage [893]. To overcome these limitations, NP surfaces can include targeting ligands that selectively recognize specific or overexpressed receptors on tumoral cells (folate, transferrin, neurokinin-1 or v3 integrin receptors) [887]. Many of these targeting ligands can also interact with receptors present on the BBB, which enhances the ability of these systems to cross the BBB through receptor-mediated transcytosis. Different targeting ligands (i.e., proteins, peptides and aptamers) have been utilized to promote active targeting of nanocarriers specifically to glioma cells [894,895,896].

Other advantages of NPs include the following: (1) nanoencapsulation increases their half-life activity, for instance in the case of TMZ-loaded chitosan NPs from 1.8 to 13.4 h [897]; (2) they can incorporate additional fluorescent/MRI/radioactive compounds that allow the non-invasive monitoring of its biodistribution [898]; (3) they increase hydrophobic drug solubility while favoring a proper biodistribution and evading the mononuclear phagocyte system catabolism; (4) they can combine different additional therapeutic approaches, such as (although not exclusively) radiotherapy sensitization, immune cells stimulation, or induction of heat/ROS [899].

### 6.1. In Vivo Imaging, Chemotherapy and Radiotherapy

Iron oxide nanoparticles (IONPs) have demonstrated promising applications as MRI contrast agents to improve the visualization of intracranial neoplasms [900]. The T2* images of Ferumoxtran-10 (first-generation ultra-small superparamagnetic IONPs) also provide complementary information by enhancing the visualization of glioma vascularity, inflammatory components, and differentiation from radiation necrotic areas better than gadolinium-based contrast agents [901]. Other merits of IONPs include their excellent biocompatibility, biodegradability, and low toxicity. Although they have received regulatory approval for clinical use, some IONPs (i.e., abdoscan, sinerem, endorem and clariscan) were recently discontinued by the pharmaceutical industry because of an inconclusive impact on patient management and for marketing reasons [902].

There is recent interest in *magnetic particle imaging*, a novel real-time and three-dimensional whole-body imaging modality with high tracer sensitivity and temporal–spatial resolution [903]. Iron–platinum particles enhanced the resolution of T2-weighted MRI images and have magnetic properties that help to guide the nanocomposite to the tumor location [904]. In vivo fingerprint Raman Spectroscopy can distinguish between normal brain, tumor tissue and necrosis with accuracies upwards of 90% in patients for grade 2–4 gliomas [905]. Gold NPs can enhance the Raman signal by several orders of magnitude because of their plasmonic effect or “surface-enhanced Raman spectroscopy” phenomenon [906].

A number of anti-cancer drugs have been successfully delivered to the brain using nanocarriers, i.e., TMZ, paclitaxel, docetaxel, cisplatin, doxorubicin (DOX), vincristine and others [887]. Several nanoformulations are being developed to overcome the limitations of TMZ penetration to improve GB treatment [907,908,909]. Sharma et al. developed a polyamidoamine (PAMAM) dendrimer-chitosan conjugate that increased TMZ brain concentration by almost two-fold [910], thus proving that polymer-based nanosystems can increase the stability of TMZ and control its release. Most of these attempts have been carried out by administering NPs directly into the tumor site, which is largely inconvenient in practice. Systemic administration of the dendrimer–rapamycin conjugate (D-Rapa) allows the release of rapamycin into the TME, improving tumor reduction without significant systemic toxicity [911].

PEGylated liposomes increased the plasma concentration of TMZ and were also more concentrated in the brain, suggesting that PEGylation is a good strategy for overcoming the liver and spleen reticuloendothelial system clearance [912]. Unfortunately, the repeated intravenous injections of PEGylated liposomes led to “accelerated blood clearance” due to the induction of anti-PEG antibodies [913], although solid lipid NPs have demonstrated similar properties to liposomes [914].

Polymer-based nanosystems have been used to co-encapsulate TMZ with other chemotherapies including DOX [915], 5-fluoracil [916] and paclitaxel [917]. Behrooz et al. demonstrated that the designed dendrimer-based pharmaceutical system was able to increase the early apoptosis from 28.2% in the case of the administration of the free drugs by up to 73.3% with the encapsulated co-delivery [917]. In orthotopic human GB xenograft models, loco-regional treatment with TMZ-loaded thermogels caused a significant reduction in the growth in tumor recurrences with no systemic toxicity, compared to untreated controls [918]. Transferrin is overexpressed in proliferating cells and cells that have undergone malignant transformation [919]. Indeed, targeting the transferrin receptor has been demonstrated to be a good strategy to increase the crossing of TMZ or other drugs through the BBB and increase their accumulation in brain tissues [920]. Recently, Helal et al. showed that TMZ-loaded albumin NPs had good uptake by GL261 and BL6 GB cells and increased cytotoxicity [921]. Furthermore, when administered i.v., the NP (labeled with a fluorescent molecule) displayed co-localization with the bioluminescent syngeneic BL6 intracranial tumor murine model used, thus highlighting the potential efficacy of this approach to targeting and treating GBs.

Several NP-based strategies have been developed to interfere with VM formation in GBs, such as (Asn-Gly-Arg) peptide (NGR)-modified liposomes containing epirubicin and celecoxib. These NPs are able to cross the BBB, accumulating in the tumor areas of the brain and destroying VM structures in orthotopic xenograft mouse models [922]. Despite a number of treatment options to target GSCs, most of them have failed in recent clinical trials [923]. Recent examples of GSC targeting include CD133-directed gold NPs, which can be utilized as imaging agents for accurate diagnosis of GB [924] or to sensitize GSCs to radiotherapy [925]. Administration of two chemotherapeutics that act through multiple non-overlapping and synergistic mechanisms is expected to improve the efficacy of treatment and prevent cancer cell drug resistance. Dual functionalized liposomes demonstrated ~12 and 3.3 fold increase in DOX and erlotinib accumulation in mouse brains, respectively, compared to free drugs, and showed excellent antitumor efficacy by inducing a ~90% GB regression and significantly increasing the median survival along with minimum toxicity [926].

Inorganic, polymeric, lipidic, and miscellaneous nanoparticles have been developed for nose-to-brain drug delivery against GB [927]. Intranasal administration of BEV-loaded poly(lactic-co-glycolic acid) (PLGA) nanoparticles (size < 200 nm) improved BEV bioavailability in the brain, and a reduction in tumor growth was accompanied by a higher anti-angiogenic effect compared to the free BEV [928]. Intranasal administration of TMZ (NCT04091503) is being studied in the randomized phase I clinical trial.

Silver (AgNPs), gold (AuNPs) and iron oxide NPs have improved the radiotherapy efficacy in vitro and in GB xenografts [929,930,931]. AgNPs showed more powerful radiosensitizing ability than AuNPs leading to a higher rate of apoptotic cell death [930]. More recently, AgNPs exhibited a higher capacity to radiosensitize hypoxic cells than normoxic cells, which represents an important advantage in GB treatment [932]. To this end, several radiosensitizing techniques are currently under investigation. These include the use of PI3K pathway inhibitors [497,933,934], DNA repair inhibitors [935], hyperthermia [936,937], aldehyde dehydrogenase inhibitors [938], or high atomic number metal NPs [939,940].

### 6.2. Gene Therapy

Aptamers are short three-dimensional structures of single-stranded nucleic acids (RNA or DNA) that bind molecular targets with high affinity. By adopting an unbiased cell-based variant of the original combinatorial systematic evolution of ligands via the exponential (SELEX) enrichment procedure, RNA aptamers that selectively bind GB cells and GSCs have been generated [941,942]. The results prove that they were able to inhibit cell proliferation, migration and stemness, and were able to strongly reduce tumor growth in vivo, thus proving that this approach is a promising innovative diagnostic and therapeutic tool for GB [943,944]. To date, just one clinical trial has assayed aptamers for GB treatment (NCT04121455). The first and only clinical trial to date using aptamer NOX-A12 in GB patients showed promising results and was well tolerated with no dose-limiting toxicities or treatment-related deaths, which is specifically interesting because it has been assayed in GB patients with unmethylated promoters [945]. The addition of the CXCL12 inhibitor NOX-A12 (olaptesed pegol) to SOC and BEV led to an overall OS rate of 83% (n = 5/6) at a median follow-up of 15 months in ndGB patients, according to preliminary data from an expansion arm of the phase 1/2 GLORIA trial [945].

RNA interference (RNAi)-based therapies use small RNA oligonucleotides (21–45 base pairs) of single- or double-stranded RNA molecules to inhibit protein synthesis. A lipopolymeric nanoparticle (LPNP) formulation has shown remarkable affinity for GSCs and can encapsulate multiple siRNAs for targeted anti-GSCs therapy. When directly infused into rat brain tumors, LPNP siRNAs efficiently inhibit tumor growth and offer promising survival benefits. This multiplexed nanomedicine platform holds great potential as a customized anti-GSCs therapy approach [946]. Despite the potential of RNAi-based therapy in cancer treatment, clinical limitations include short circulatory stability, rapid clearance from the body, and inadequate delivery to the brain tumor tissue. To solve these problems, RNAi-aptamers such as RNAi attached to cell-penetrating peptides and RNAi CNS-ligand conjugates have been explored for GB therapy [947,948,949]. Recently, Wang et al. developed an iron oxide NP system for targeted delivery of siRNAs to suppress the TMZ-resistance gene MGMT. The sequential intravenous administration of these NPs and TMZ resulted in increased apoptosis of GSC and GB cells, reduced tumor growth, and significantly prolonged survival as compared to mice treated with TMZ alone, and this without significant tissue toxicities [950].

The Clustered Regularly Interspaced Short Palindromic Repeat associated (Cas) nuclease 9 (CRISPR-Cas9) system is the latest gene editing technology, which stands out as the fastest, highly versatile, and most reliable gene editing tool for discovering genetic alterations, oncogenic targets, and for epigenetic regulation in various cancers, including GB [951]. It uses an engineered single guide RNA (sgRNA) to direct the Cas9 nuclease to specific DNA sections, leading to double-stranded breaks (DSBs). Induced DSBs are repaired through non-homologous end joining to generate insertions or deletions or homology-directed repair of gene modifications at targeted genomic locations [952,953]. CRISPR-Cas 9 gene editing technology offers novel insights into the roles of various genes in regulating proliferation, stemness, angiogenesis, and invasion of GB cell lines [951]. CRISPR-Cas9 screens are utilized to identify new biomarkers, oncogenic drivers, and causes of chemotherapy resistance in cancer. In that sense, the CRISPR-Cas9 methodology was used to identify growth-related subtypes of GB with therapeutical significance [954]. CRISPR-Cas9 has also been used to generate animals carrying genetic mutations to model human diseases. Based on this, the inactivation of multiple tumor suppressor genes such as Nf1, P53, Ptch1, and PTEN in mouse brains has facilitated the development of medulloblastoma and GB disease models [955,956].

The greatest challenge for CRISPR/Cas9 therapy is how to safely and efficiently deliver it to target sites in vivo, which is especially difficult in the CNS. To overcome this problem, CRISPR/Cas9 therapeutic delivery using viral and nonviral-based delivery vehicles is rapidly expanding [957]. Invasive intra-brain injection of CRISPR-Cas9 complexes contained in viral vectors or NPs often leads to serious side effects, i.e., infection, inflammation, swelling, and tissue injury [958]. Thus, noninvasive delivery of encapsulated CRISPR-Cas9 complexes is urgently needed. Rosenblum et al. describe a safe and efficient lipid nanoparticle (LNP) for the delivery of Cas9 mRNA and sgRNAs. A single intracerebral injection of CRISPR-LNPs against PLK1 in aggressive orthotopic GB caused tumor cell apoptosis, inhibited tumor growth by 50%, and improved survival by 30% but showed diverse adverse effects [959].

BBB-penetrating single CRISPR-Cas9 nanocapsules (~30 nm) have been utilized to target GB cells [960]. Mutagenesis and immunogenicity are the disadvantages of using virus-based vectors due to the long-term expression of CRISPR/Cas9 nuclease and sgRNA, so a transient delivery system is preferred to editing the genome for therapeutic purposes. In that sense, nonviral nanoparticle-based vectors have the advantage that they are often nonimmunogenic and permit bulk production [961].

The i.v. administration of Cas9/sgRNA ribonucleoprotein (RNP) complexes for GB treatment is limited by poor in vivo stability of Cas9 protein and sgRNA, low BBB permeability, and non-specific cell targeting [962]. Ruan et al. developed an angiopep-2 decorated, guanidinium and fluorine functionalized polymeric nanoparticle uploaded with Cas9/gRNA. The guanidinium and fluorine domains were both capable of interacting with and stabilizing Cas9/gRNA RNP in blood circulation without impairing its activity. This NP efficiently crossed the BBB, accumulated in brain tumors, suppressed their growth, and significantly improved the median survival time of mice bearing orthotopic GB, while inducing negligible side or off-target effects [960,962]. Liposome-templated hydrogel nanoparticles (LHNPs) with a unique core-shell nanostructure have been developed for the efficient delivery of Cas9 protein and nucleic acids. When combined with minicircle DNA technology, LHNPs show high effectiveness in cell culture. LHNPs can be tailored for targeted gene suppression in malignancies, especially brain tumors [963]. Additionally, LHNPs can be produced using an autocatalytic brain-tumor-targeting mechanism, which allows targeted delivery of CRISPR/Cas9 to brain tumors. Furthermore, LPHNs nanocarriers loaded with CRISPR/Cas9 plasmids targeting the MGMT gene and modified with the cyclic arginine-glycine-aspartic-conjugated (cRGD) peptide (which targets the overexpressed integrin αvβ3 receptors in tumor cells) were successfully synthesized and demonstrated the ability to protect pCas9/MGMT from enzymatic degradation. In vivo, microbubble–LPHN–cRGD complexes safely and locally enhanced the permeability of the BBB under the effect of focused ultrasound radiation. This facilitated the accumulation of NPs in GB tumors, enhanced the therapeutic effect of TMZ and prolonged survival in mice with orthotopic GB tumors [964]. Systemic delivery of CRISPR/Cas9 engulfed into extracellular vesicles showed improved CRISPR-Cas9 loading efficiency, excellent glioma targeting and penetration potentials, and sensitization of GB cells to radiotherapy, in preclinical animal models, without off-target side effects [965].

Recently, gene editing techniques targeting the PD-1/PD-L1 pathway have attracted considerable interest. However, efficient distribution without causing side effects in clinical trials remains a challenge. To address this, a nanoparticle delivery method, using a low-molecular-weight polyethylenimine (PEI) lipid shell and a PLGA core to package the PD-L1 gRNA-CRISPR/Cas9 plasmid, has been developed. This system effectively transfected human U87 glioma cells with PD-L1. The results represent a promising immunotherapy platform for treating GB [966]. In addition, PEI-coated Fe_3_O_4_ NPs have been utilized as a vehicle for delivering therapeutic siRNA to GB cells. These NPs effectively reduced cell proliferation and migration [967]. Similar results have been obtained by a dual-sgRNA CRISPR/Cas9 knockout of PD-L1, thus indicating that intracellular PD-L1 is necessary for tumor progression [968]. CRISPR/Cas9 technology has also allowed an increase in the efficacy of CAR immunotherapy in preclinical GB models. Human neutrophils effectively cross the BBB and display effector immunity against cancer cells, but the short lifespan and resistance to genome editing of primary neutrophils have limited their broad application in immunotherapy. Human pluripotent stem cells were engineered with CARs and differentiated into CAR-neutrophils that are loaded with rough silica (SiO_2_) NPs containing hypoxia-targeting tirapazamine or other drugs, as a dual immunochemotherapy. Systemically administered CAR-neutrophil@R-SiO_2_-TPZ NPs first attack external normoxic tumor cells by forming immunological synapses and killing tumor cells through phagocytosis. Thereafter, CAR-neutrophils release R-SiO_2_-TPZ NPs, which are taken up by hypoxic tumor cells. These CAR-neutrophils loaded with drug-containing SiO_2_ NPs display superior anti-tumor activities against GB, prolong lifespan in mice bearing orthotopic xenografted GBs and reduce off-target drug delivery [969].

Existing approaches to deliver CRISPR-Cas9 to the brain include viral vector delivery (lentivirus and adeno-associated viruses) [970] and nonviral synthetic delivery (gold, lipid, and polymers) [971], which have provided a valuable in vivo proof of principle. Nevertheless, they are not ideal systems for human clinical applications due to their limited BBB penetration and lack of specific brain disease site targeting [960,972]. Moreover, viral vector delivery can generate highly risky immune responses due to off-targeting effects. Current nonviral delivery systems are limited by their loading efficiency, non-disease targeting, and problematic clearance out of the brain [973]. These problems are being solved one by one [960,962,974] and, although most of the aforementioned studies are in preclinical phases, these novel CRISPR-Cas9–brain delivery systems seem a versatile and potent platform for treating GB and other brain diseases.

### 6.3. Nanosystems Used for Immunotherapeutic Purposes

NP platforms can be designed to efficiently deliver tumor antigens, cytokines, lymphocytes, and therapeutic agents, stimulating the host’s immune system to effectively target tumor cells in a safe manner. CCL21 is a cytokine predominantly expressed in the lymphoid tissue that functions as a chemoattractant for DCs, lymphocytes, and NK cells. Intratumor injection of CCL21-coupled vault NPs enhanced immune response and significantly inhibited GB growth in mice models [975]. T cells can be encapsulated in a poly(ethyleneglycol)-g-chitosan-based degradable hydrogel, which releases T cells to treat GB effectively [911]. Targeted nanoscale immunoconjugates with poly(β-L-malic acid) (a natural biopolymer), covalently attaching a-CTLA-4 or a-PD-1, crossed the BBB and activated a local brain anti-GB immune response. This resulted in an increase in CD8+ T cells, NK cells and macrophages with a decrease in Tregs in the GB tumor area, and in a significantly increased survival compared to animals treated with single free anti-CTLA-4 and anti-PD-1 monoclonal antibodies [976]. Other similar studies also support a greater availability and effectiveness of ICIs when they are incorporated into NPs [977,978]. An immunosuppressive TME in GB was efficiently altered by nanodiamonds containing doxorubicin (Nano-DOX) [979] and biomimetic nanoparticles (CS-I/J@CM NPs) [980], which stimulated the GB cell immunogenicity and activated a potent anti-GB immune response.

Porous biomaterial scaffolds loaded with GM-CSF have proven to be a promising strategy to increase the chemotactic signal and replenish DCs against GB [760]. Based on this idea, Qui et al. designed an injectable nanocomposite/hydrogel system (DOX/RAcNPs@GM gel) consisting of DOX, GM-CSF and imiquimod-loaded antigen-capturing NPs to actively recruit DCs, aiming to elicit a durable anti-tumor response. To confirm this hypothesis, the authors established an orthotopic GB tumor model that was resected on day 14 after tumor engraftment, while the remaining cavity was injected with hydrogel. DOX/RAcNPs synergized with GM-CSF to enhance immune responses, and rats treated with DOX/RAcNPs@gel displayed reduced tumor volume compared with controls or TMZ-treated rats, although all animals died within 32 days (a fact attributed to the insufficient immune infiltrates in the tumor site) [981].

NPs can protect vaccines from degradation and enhance antigen-presenting cell uptake. Various immunostimulants, such as Toll-like receptor (TLR) agonists and stimulator of interferon genes (STING) agonists, have been preclinically or clinically tested for in situ tumor vaccination, but render suboptimal therapeutic efficacy due to poor tumor retention and rapid systemic dissemination [982]. NvIH is a thermo-responsive hydrogel co-encapsulated with ICI antibodies and novel polymeric NPs loaded with three immunostimulatory agonists for Toll-like receptors 7/8/9 (TLR7/8/9) and STING. In situ vaccination with a single dose of NvIH reduced TME immunosuppression, enhanced TME antitumor immune milieu, and elicited systemic antitumor immunity in an orthotopic murine GB model [983].

### 6.4. Clinical Trials Using NPs for the Treatment of GB

Although numerous in vivo and in vitro studies have been conducted to prove the therapeutic efficacy of nanocarrier-based treatments against GB, few clinical trials using nanotherapies have been completed. NanoTherm^®^ (aminosilane-coated superparamagnetic iron oxide NPs) generates heat upon application of an external alternating magnetic field, inducing thermal ablation of the tumor tissue [178]. Mechanisms involved in hyperthermia-induced chemo/radiosensitisation include the generation of a transient disruption of the BBB, increased blood flow, interference with DNA repair mechanisms, heat-induced damage to ATP-binding cassette (ABC) transporters, changes in tumor cell drug metabolism, and an impaired ability to withstand apoptotic pathways [984]. Accordingly, in a larger phase II study, NanoTherm^®^ was directly injected intratumorally under stereotactic guidance and then heated using an alternating magnetic field. Treatment was combined with fractionated stereotactic radiotherapy. This approach doubled the mOS of rGB patients (up to 13.4 months) and no serious complications were observed [985]. In Europe, these findings led to the approval of NanoTherm^®^ for magnetic hyperthermia therapy application in combination with RT in patients with rGB.

Most clinical trials are based on liposomal drug-delivery systems. Doxorubicin liposomal formulations have been tested following the Stupp regimen (NCT00944801, NCT009448019) without any survival benefit [986,987]. An ongoing phase II trial (NCT01386580) is evaluating the efficacy of glutathione pegylated liposomal doxorubicin (2B3-101) as monotherapy and in combination with trastuzumab in patients with solid tumors and brain metastases or recurrent malignant glioma. SGT-53 is a complex of cationic liposomes encapsulating a normal human wild-type p53 DNA sequence in a plasmid backbone. This complex has been shown to cross the BBB and to deliver the p53 cDNA efficiently and specifically to the tumor cells. SGT-53 significantly chemo-sensitized human GB cell lines (U87 and U251) to TMZ in vitro and in vivo. Furthermore, in an intracranial GB tumor model, two cycles of concurrent treatment with systemically administered SGT-53 and TMZ inhibited tumor growth, increased apoptosis and, most importantly, significantly prolonged median survival [988]. Interestingly, combining anti-PD-1 with SGT-53 was very effective in inhibiting GB growth, inducing tumor cell apoptosis and increasing intratumoral T cell infiltration, resulting in a significant survival benefit in mice bearing intracranial GB [989]. Unfortunately, the phase II trial (NCT02340156) studying the efficacy of SGT-53 and TMZ combined treatment in rGB patients was terminated due to an insufficient number of participants, making it impossible to perform statistical analysis. Ongoing clinical trials include polysiloxane Gd-Chelates-based NP in combination with radiotherapy and TMZ in ndGB patients (NCT04881032), and NP albumin-bound rapamycin (ABI-009) alone or in combination with BEV, TMZ, or other drugs and radiation therapy in rGB and high-stage ndGB patients (NCT03463265); however, the results are not available yet.

siRNA oligonucleotides arranged on the surface of small spherical AuNPs (NU-0129) were designed to target the oncogene BCL2L12. After intravenous administration, NU-0129 uptake into glioma cells correlated with a reduction in tumor-associated Bcl2L12 protein expression, demonstrating the potential of nanoconjugates as a potential brain-penetrant precision medicine [990].

## 7. Non-Ionizing Energies in GB Therapy

In addition to the use of TTFields, which are part of the SOC for GBs, other types of non-ionizing energies (described here below) are also being developed for therapeutic purposes.

### 7.1. Laser Interstitial Thermal Therapy

Laser interstitial thermal therapy (LITT) represents a cutting-edge approach for treating brain tumors that are difficult to access through conventional surgery. By inserting a laser catheter into the tumor, LITT may eradicate the tumor by raising its temperature to lethal levels. The catheter implantation process utilizes state-of-the-art computer imaging techniques, ensuring precision and accuracy. Real-time MRI guides the laser through the catheter, enabling neurosurgeons to target the thermal energy solely at the tumor site to minimize damage to surrounding healthy brain tissue. One of the most notable advantages of LITT is its minimally invasive nature and that most patients can return home the day after treatment and quickly resume their normal activities. LITT also offers hope to patients who have not responded to stereotactic radiosurgery or are suffering from radiation necrosis [991,992]. A single-center study [993] and a prospective multicenter registry [994] conclude that LITT can safely reduce intracranial tumor burden in GB patients who have exhausted other adjuvant therapies or are poor candidates for conventional resection techniques. A statistically significant OS advantage was observed in ndGB patients receiving both radiation and chemotherapy within 12 weeks of LITT (16.14 months) versus those who received only one treatment modality or no treatment following LITT (5.36 months) [995]. New neurologic deficits and postprocedural edema (normally resolved with steroid treatment) are the most frequently reported adverse events after LITT [994].

### 7.2. Focused Ultrasound (LIFU and HIFU)

Focused ultrasound is an early-stage, therapeutic technology that offers possible adjuvant or alternative treatment strategies for GB [996]. This groundbreaking approach involves precisely targeting deep areas of the brain with beams of ultrasonic energy without the need for incisions. This being said, it is important to differentiate between low- and high-intensity (100–10,000 W/cm^2^) focused ultrasounds (LIFUs and HIFUs). LIFU disrupt the BBB or blood tumor barriers and enhance the uptake of therapeutic agents in the CNS. HIFU can cause thermoablation and mechanical destruction of the tumor. Both can be combined with radiotherapy [997,998].

The advantages of focused ultrasounds over current brain tumor treatments are considerable: (1) eliminating concerns related to surgical wound healing and the risk of infection, making it a safer option for patients; (2) precise targeting; (3) avoiding ionizing radiation exposure; (4) enhancing chemotherapy delivery by temporarily opening the BBB; and (5) their non-invasive nature allows for repeat treatments [854,997].

HIFUs produce frictional heat by causing the vibration of molecules within the tissue. The absorbed energy can quickly elevate the temperature to over 55 °C, which causes protein denaturation, DNA fragmentation and coagulative necrosis when maintained for just a few seconds [999,1000]. This thermoablative process further increases tumor sensitivity to radiation by damaging DNA repair enzymes [997]. To date, clinical data are limited to case reports such as those first reported by Coluccia et al. using MRI-guided HIFUs to achieve tumor ablation without inducing neurological deficits or other adverse effects in a patient with rGB [1001]. Two phase I clinical trials (NCT01473485 and NCT00147056) have evaluated the safety and efficacy of transcranial MRgFUS (magnetic resonance-guided focused ultrasounds) thermoablation for the treatment of either brain metastasis or recurrent glioma, but the results have not been published. MacDonell et al. have recently proposed an interstitial HIFU device that employs an intraparenchymal catheter to induce hyperthermia directly at the tumor tissue, assisted by an MRI-guided robotic system. The advantages of this interstitial device over external MRI-guided HIFU include avoiding attenuation from the skull, improving treatment margins, and enabling concurrent tissue sampling [1002]. Animal studies have demonstrated the feasibility of this technique, but its clinical success has not yet been validated. HIFU technology is approved by the FDA for treatment of several cancers (i.e., prostate, uterine leiomyoma and bone metastasis) and is under investigation for other neoplasms [1003]. The primary limitation of its application in GB is the absence of well-circumscribed lesions [1000].

In contrast, LIFU uses relatively lower energy pulsed waves (around 500 kHz) relying on mechanical perturbation and acoustic cavitation. Cavitation refers to the oscillation and collapse of gas bubbles in response to the compression and refraction of the ultrasonic pressure wave [1003]. LIFU is therefore generally used in conjunction with microbubbles, which can be delivered intravenously and travel to the site targeted by the transducer [1004]. These particles oscillate in the presence of the ultrasound wave, expanding and contracting to produce a stable cavitation effect that disrupts the tight junctions of endothelial cells. Thus, LIFU has been explored as a method to transiently increase the permeability of the BBB to enhance therapeutic delivery, limiting the side effects by ensuring that the impermeable state of the BBB is quickly restored [998]. The precision of LIFU can be enhanced using MRI (MRgLIFU), thus minimizing the effects on healthy tissue [1005]. Furthermore, the opening of the BBB can be confirmed with contrast-enhanced MRI, allowing real-time monitoring of the biological effects of LIFU [1003]. In animal studies, BBB opening is immediate, repeatable, resolves within 6 to 8 h, and does not cause axonal or neuronal injury. The improved delivery of BCNU, TMZ, carboplatin and others, traditionally rendered ineffective by the impermeability of the BBB, has been verified. As a consequence, this treatment not only delayed tumor progression but also enhanced survival in GB animal models [1006,1007,1008]. LIFU has also been used to deliver viruses [1009], cells [1010] and NPs (loaded with imaging agents, therapeutic agents, or both) [1011,1012,1013]. In addition, the microbubbles can be loaded with tumor antigens, allowing a more focused and effective immune response [1014,1015]. It is also noteworthy that preclinical studies suggest that LIFU reduces the TME-induced immunosuppression by increasing infiltration of NK and CD8^+^ T cells, thus facilitating DCs maturation and diminishing the number of Tregs and MDSCs [1016,1017,1018,1019]. MRgLIFU-mediated BBB disruption has been utilized in the clinical setting to deliver carboplatin, TMZ, doxorubicin, fluorescein or paclitaxel in GB patients [1020,1021,1022]. The treatment was well tolerated, and disruption resulted in a 15–20% increase in contrast enhancement almost instantaneously, resolving after 20–24 h. A single-center trial (NCT02253212) did not show serious adverse events or carboplatin-related neurotoxicity associated with the implantation of a LIFU device with microbubble injection in rGB patients. Patients with documented BBB disruption relative to patients without or with poor BBB disruption had longer PFS (4.11 vs. 2.73 months) and OS (12.94 vs. 8.64 months) [1020]. A small trial (NCT03712293) of six patients with GB treated with multiple cycles of MRgLIFU had an improved penetration of TMZ without immediate or delayed BBB-disruption-related complications. All subjects survived over 1 year, while tumor recurrence was noted in two patients at 11 and 16 months [1021]. These recent results evidence that LIFU-mediated BBB opening increases drug delivery in GB, thus improving tumor control and survival, although larger sample sizes are needed to confirm efficacy and the lack of hemorrhagic complications associated with the procedure [1023].

At present, a number of clinical trials (NCT03551249, NCT04440358, NCT04417088, NCT05370508, NCT06039709) are ongoing or recruiting patients with GB for focused ultrasound treatment (ClinicalTrials.gov, 4 February 2024).

### 7.3. Photodynamic and Sonodynamic Therapies

Photodynamic therapy (PDT) and sonodynamic therapy (SDT) are emerging modalities for non-invasive cancer treatment, based on the tumor-selective accumulation of non-toxic molecules [photosensitizers (PS) or sonosensitizers (SS)], which are activated by laser light or ultrasound radiation to produce a localized cytotoxic effect via ROS generation [1024,1025,1026,1027]. Apoptotic and necrotic cells elicit the proliferation of effector T cells in the lymph nodes, resulting in further GB eradication. Both techniques can also induce autophagy, endothelial damage, angiogenesis inhibition (associated with ischemia and necrosis), and immune responses [1027,1028,1029,1030].

Protoporphyrin IX (PpIX) and fluorescein are most widely used as PS and as SS due to their safety profile and selective accumulation in tumor cells, and most studies in glioma use 5-aminolevulinic acid (5-ALA) as a precursor of PpIX [1031,1032,1033]. After surgery and adjuvant treatment, the residual tumor is predominantly comprised of resistant GSC clones; thus, several strategies have attempted to enhance 5-ALA uptake by this type of cells in order to increase the efficacy of PDT and SDT [1032,1033]. Moreover, iron chelators, such as deferoxamine and CP94, and ABC transporter inhibitors have been shown to increase PpIX levels in GB cells when utilized as adjuvants [1034,1035]. GSCs exhibit less accumulation of PpIX than non-GSCs, and deferoxamine-induced iron chelation significantly enhances the 5-ALA-mediated PpIX accumulation in GSCs [1034].

Interstitial PDT (iPDT) is a minimally invasive procedure performed in patients whose tumors are present in areas of the brain with readily identifiable neurological function, or in fragile patients who cannot undergo a craniotomy. iPDT is applied via the stereotactic insertion of fiber optic cable(s) into the tumor to deliver photostimulation to the tumor mass after administering a PS to the patient. Intracavitary PDT is applied in the resection cavity at the end of the surgical procedure [1025]. The most frequent PDT-associated adverse events are retinal and cutaneous photosensitivity that can last from several days to a few weeks depending on the PS used and time of exposure. Less frequent adverse events include post-operative hemorrhage, neurological deficits (particularly in the case of large tumors), infection, uncontrolled cerebral edema, and even death [1025]. The use of iPDT against GB over the last 3 decades demonstrates that, overall, the technique is safe and effective. The mPFS was 14.5 months for ndGB and 14 months for rGB patients, which means there is an improvement compared to historical controls [1036]. Different single-center retrospective studies have reported prolonged long-term survivals when using iPDT [1037,1038,1039,1040]. However, the lack of a standard control group, restricted sample sizes, and the lack of information about their characteristics, makes it difficult to draw conclusions. Ongoing clinical trials (NCT04469699, NCT03897491, NCT04391062) are actively investigating the potential of PDT in GB treatment [1027]. The major disadvantage of PDT is the limited penetration of laser light into deep tissues. This could be mitigated using near-infrared radiation (NIR) that can penetrate 3 cm through skin and bone structures [1041]. In fact, NIR photons also diminish both phototoxicity and background autofluorescence, which then leads to improved bio-imaging when compared to traditional fluorescence with visible light [1042].

Recent advances in PDT-GB research are as follows: (1) coupled NIR and photo immunotherapy (NIR-PIT), where a photosensitizer is conjugated to a highly specific monoclonal antibody [1043,1044,1045]; (2) NP-based PDT to augment systemic therapies and avoid skin photosensitization [1046,1047]; (3) NP-based PDT linked to miRNA or other chemotherapeutic agents [1048]; (4) strategies to increase PDT efficacy in hypoxic TME conditions [1049,1050,1051,1052]. Although some studies have demonstrated the enhanced penetration of light by simultaneous application of ultrasounds [1053], combined effects of PDT and SDT did not show any benefit in glioma rat models [1054].

SDT requires the interaction of an SS, ultrasounds, and oxygen. The generation of ROS through the stimulated SS and the ultrasound-activated cavitation effects can together induce apoptosis, necrosis, and autophagy, ultimately causing tumor destruction [1026]. The major advantage of SDT for GB treatment is the ability of ultrasound energy to penetrate into soft tissues more than 10 cm, and the possibility of delivering a tightly focused ultrasound beam for focal treatment [1031]. SDT inhibits tumor growth and increases animal survival in preclinical studies [1028,1055,1056,1057]. A significant step forward was made by Raspagliesi et al., who reported, in a porcine animal model, the first intracranial MRI-guided SDT with fluorescein and 5-ALA using the ExAblate system. The porcine model allows a more precise target definition, and better approaches human conditions [1058]. The efficacy of STD is limited in larger tumors because ultrasound waves cannot penetrate deep enough into the tumor [1059]. Three clinical trials are currently underway to evaluate the safety and feasibility of SDT with 5-ALA in patients with high-grade glioma (NCT04559685), in ndGBs using the ExAblate Model 4000 Type-2 Neuro System (NCT04845919), and the efficacy of SDT using SONALA-001 and Exablate Type-2 devices in subjects with rGB (NCT04988750).

### 7.4. Microwaves and Pulsed Electric Fields

Microwaves are a form of electromagnetic radiation with wavelengths ranging from about 1 mm to 30 cm, corresponding to frequencies between 1 GHz and 300 GHz, respectively [1060]. Microwaves have been introduced in medicine for cancer diagnosis [1061] and treatment [1062,1063]. However, microwaves have well-known adverse effects on the CNS and can affect neurotransmitter release and, thereby, cause a delay in the signaling process [1064]. Despite these limitations, the research of Rana et al. is an excellent example of how microwave radiation can therapeutically be used to treat GBs [1065]. These authors have shown that a strong electric field (~23 kV/cm) of pulsed high-power microwave (HPM) irradiation causes ROS generation, DNA damage, p53 activation and death in exposed GB U87 cells. Importantly, these authors show that pulse dosage causing damage to GB cells and brain normal cells is different, thus representing a new therapeutic approach that deserves to be tested rapidly in in vivo models. In fact, pulsed microwave-induced thermoacoustic therapy has been recently proposed as a potential alternative modality to precisely and effectively eradicate orthotopic GB. Interestingly, an NP composed of polar amino acids and adenosine-based agonists has been developed having a high microwave absorbance and selective penetration of the BBB. The NP activates the adenosine receptor on the BBB to allow self-passage, and once it is accumulated in the TME, the NP converts absorbed microwaves into ultrasonic shockwaves, which can mechanically destroy tumor cells with minimal damage to adjacent normal brain tissue due to the rapid decay of the ultrasonic shockwave intensity [1066].

Pulsed electrical fields [PEF, high voltage/short-duration (nanoseconds-milliseconds) electrical pulses] have emerged as a non-thermal tissue ablation treatment for malignant neoplasms. This technique, where rod/needle-like electrodes are strategically placed directly in or surrounding the tumor, is associated with the terms electroporation and electro-permeabilization [1067]. Importantly, reversible permeabilization of the cell membrane can also increase the uptake of chemotherapeutics or facilitate transfection approaches [1068]. In the sub-microsecond regime, intracellular effects like nuclear membrane disruption have been observed [1069]. The strength and duration of the nanosecond pulse can create nano-sized pores and electroporation effects [1070,1071,1072]. Depending on these parameters, cell electroporation and permeabilization can be reversible, with membrane recovery typically taking minutes, although intracellular repair may require hours [1073]. Sub-microsecond pulses have also been found to trigger apoptotic cell death in different cell lines and tumor tissues. Apoptosis induction is associated with caspase activation [1074], intracellular Ca^2+^ release [1075], loss of mitochondrial membrane potential [1076], and DNA damage [1077]. These findings indicate that sub-microsecond PEF protocols, below electroporation thresholds, may offer potential therapeutic benefits for GB treatment and should be explored further [1078].

The first demonstration using irreversible electroporation (IRE) against a dog malignant intracranial glioma was published in 2011 by Garcia et al. [1079]. Canine malignant gliomas share similarities with GB in various clinical, biological, pathologic, molecular, and genetic aspects, making them a valuable model [1080]. In the referenced study, IRE delivery resulted in an approximately 74% reduction in tumor volume. IRE treatment was well tolerated and achieved safe tumor ablation when combined with radiotherapy, anti-edema treatment, and anticonvulsants, with minimal hemorrhage [1079]. More recently, further investigations using the Nanoknife procedure in seven dog glioma models revealed that IRE treatment was successful in six out of seven dogs without inducing or exacerbating edema or causing a significant hemorrhage [1081]. While most adverse effects were minimal or typical of post-operative surgery, one dog experienced severe cerebral edema (due to the tumor location being close to periventricular regions, a common site of occurrence in human glioma). This highlights the importance of considering tumor location and potential effects during the pre-treatment and planning of IRE in the brain. IRE procedures have shown preservation of critical structures and major blood vessels in humans which is an advantage over microwave or radiofrequency ablation methods [1082].

High-frequency irreversible electroporation (H-FIRE, or 2nd generation of IRE) works by delivering short, 1–10 μs pulses in a series of bursts, equivalent to a single monopolar 100 μs pulse used in traditional IRE. However, this approach requires much greater field strength to achieve the same lesion size [1083,1084]. Latouche et al. conducted an experiment using H-FIRE treatment to selectively ablate intracranial meningioma in three dogs. One of the dogs was alive after 6 months without evidence of the presence of a tumor. Another dog was alive but required increased anticonvulsants to control seizure activity, and there were suspicions of residual or recurrent tumors on an MRI 5 months after treatment. Unfortunately, the third died after 76 days due to a recurrent status epilepticus. This study showed no post-operative adverse effects attributed to H-FIRE [1085]. Recently, Campelo et al. reported further evidence that H-FIRE improves survival and immune cell infiltration in rodents with malignant gliomas [1086]. Although the application of these techniques is still in its infancy, available results indicate the potential of H-FIRE’s for brain tumor ablation, thus representing an exciting opportunity for clinical applications.

### 7.5. Targeted Radionuclide Therapy

Targeted radionuclide therapy (TRT) is based on the use of a molecule labeled with a radionuclide to deliver (through systemic or local administration) a toxic level of radiation to cancer cells. This is achieved by employing a biochemical vector, which is linked to a radionuclide and, in most cases, allows both diagnostic and also therapeutic applications. The energy, range of radiation, and type of emission are critical in targeted radionuclide therapy. Unlike molecular imaging, which involves the use of highly penetrating γ- and positron (β^+^)-emitting radionuclides, TRT employs β^−^, α, or auger electron emitters with lower penetration capacity but higher ionizing energy [1087]. β^−^ particle-emitting radionuclides (e.g., ^131^I, ^90^Y, ^186/188^Re and ^177^Lu) can irradiate tissue volumes with multicellular dimensions and induce radical formation leading to DNA single-strand breaks. For small tumors, micrometastatic lesions, or residual disease, α-particles (emitted by ^213^Bi, ^225^Ac, or ^211^At) are considered a better option, owing to their short travel distance in tissue (only a few diameters) and high linear energy transfer (LET) (50–230 keV/μm). α-particles induce DNA DSBs that directly trigger cell death, independently of the cell cycle phase, the cell oxygenation level and the MGMT gene promoter methylation status [1088].

Radiolabeled small molecules, radioimmunotherapy (RIT), peptide radionuclide therapy (PRT) and radioNPs are four different modalities of TRT [1089,1090,1091]. RIT uses a monoclonal antibody to achieve targeted vectorization of a radionuclide. Clinical trials for certain antigen targets, like EGFR [1092,1093], tenascin [1094,1095,1096,1097], or DNA histone H1 complex [1098], have shown positive outcomes. Tenascin targeting appears to be one of the most promising RITs for GB. mOS of GB patients treated by fractionated intracavitary radioimmunotherapy with ^131^I- or ^90^Y-labeled anti-tenascin monoclonal antibody reached 25.3 months, thus markedly exceeding that of historical controls, being adverse events well controllable [1099]. In the radiopeptide approach, an agonist or antagonist peptide is used to vectorize the radionuclide to a specific receptor overexpressed in cancer cells. In GB clinical trials, radiolabeled somatostatin analogs have been used to target the somatostatin receptor [1100], radiolabeled substance P has been used to target the neurokinin receptor type 1 [1101,1102,1103], and TM-601 (a recombinant version of chlorotoxin) has been used to target the matrix metalloproteinase [1104]. In most of these assays, partial remissions and an improved OS have been observed. TAM and microglia involved in TME immunosuppression are characterized by the upregulation of somatostatin receptor 2; therefore, targeting this receptor has the additional advantage of increasing the immune response against GB [1088]. RadioNP can passively accumulate in the tumor or can have a biologically active peptide or antibody for specific targeting in the same way as the molecular radiopharmaceuticals used in RIT and PRRT.

RadioNPs can be delivered passively or actively using liposomes, metallofullerenes, or lipid nanocapsules. Recently, Georgiou et al. administered ^177^Lu-AuNPs by CED to treat orthotopic U251-Luc human GB tumors in NRG mice. A high proportion of ^177^Lu-AuNPs was retained in the U251-Luc tumor for up to 21 days with minimal redistribution to the brain or healthy tissues. The radiation dose in the tumor was 599 Gy, whereas in the surrounding brain, it was 93-fold lower (6.4 Gy), and 2000–3000-fold lower doses were calculated for the contralateral left cerebral hemisphere (0.3 Gy). MRI at 28 days post-treatment showed no visible tumor in mice treated with ^177^Lu-AuNPs, and 5/8 of them survived up to 150 days, whereas controls had large tumors and required sacrifice within 45 days post-treatment. The results of this study are promising

TRT holds the potential to serve as a potent and supplementary treatment following SOC therapy for primary GBs. It can also be employed as an auxiliary treatment option in cases where the tumor tissue shows resistance to radiation and/or chemotherapy [1088,1105]. Radiopharmaceuticals can be administered systemically or intratumorally/ intracavitarily to circumvent the BBB [1089,1090]. The lack of serious adverse effects and promising results of the previously mentioned phase I/II trials makes TRT an attractive treatment modality that should be considered for phase III trials assays integrated into combined-modality regimens [1106,1107]. As an added advantage, TRT allows interesting theragnostic approaches, thus permitting personalized therapy [1089,1108].

## 8. Conclusions

No other cancer has witnessed as many therapeutic attempts as GB. Unfortunately, although every treatment option in the Oncotherapy arsenal has been tried, GB almost always recurs. In fact, if you read Dr. Siddhartha Mukherjee’s book *The Emperor of All Maladies: A Biography of Cancer*, it is easy to think about GB as the emperor of all cancers.

Despite improvements in shorter-term survival rates, the 5-year survival rate after diagnosis remains relatively constant (only 5.8% of all patients). Key advancements, limitations, and perspectives can be summarized as follows:GB location, aggressivity and infiltrative biological behavior represent exceptional challenges. Surgeons face great difficulties in removing all cancerous cells without causing damage to critical brain regions, thus preserving brain function while treating the tumor becomes a delicate balance. Aggressive treatments might easily lead to cognitive impairments and/or neurological deficits.Standard treatment and targeted therapies have only modestly improved patient outcomes, a fact likely due to redundant compensatory mechanisms, insufficient target coverage (in part related to the BBB), and GSC’s ability to remain quiescent or upregulate their molecular defenses until cytotoxic drug concentrations decrease. Mechanisms favor treatment failure and tumor recurrence.Targeting of key contributors to GSC’s self-renewal, survival, and plasticity will contribute to avoiding recurrences. ChemoID test (NCT03632135) uses CSCs isolated from tumor biopsies to select the most effective chemotherapeutic treatment for each patient [1109]. Strategies like this can assist in identifying the best treatment option and in properly designing recruitment for clinical trials.The BBB restricts the entry of many therapeutic agents into the brain, limiting the effectiveness of systemic treatments. Although the BBB and BTB can be disrupted in a variety of ways depending on the stage of the illness, their heterogeneous breakdown makes it quite difficult to achieve uniform drug concentrations inside the tumor. Strategies to improve drug delivery at bioefficacious concentrations include intranasal or intratumoral administrations (with limited acceptance in clinical practice), opening of the BBB with LIFU, and the use of nanocarriers. Especially interesting are multifunctional nanocarriers that allow the delivery of diverse molecules (chemotherapeutic and/or immunotherapeutic and/or imaging agents) and can also be used for theranostic purposes. In addition, receptor-based therapies have the potential to improve clinical outcomes. Nevertheless, most of these technologies are, at this moment, in the development stages.While immunotherapies have revolutionized treatment for other cancers, in GB, patients have not achieved similar positive responses, a fact associated with the unique challenges associated with immunosuppressive TME. GB exhibits an immune-privileged nature characterized by limited lymphocytic infiltration, cytotoxic T cell exhaustion, recruitment of pro-tumorigenic TAMs, downregulation of cancer cells’ MHC I complexes, and abundance of immune-inhibitory molecules such as IL-10 and TGFβ. In our opinion, DC vaccination, oncolytic virotherapy, CAR-cell therapy or combined strategies (with ICIs) seem promising, and outcomes could likely be improved by targeting antigens present in GCS. In any case, immunotherapy should be considered as part of the combined treatment needed to increase patient survival. The integration of immunotherapy with SOC for GB presents opportunities and challenges, since radio/chemotherapy can substantially influence the immune response in two different ways. Radiotherapy can induce immunogenic cell death, thus facilitating the release of tumor antigens that could potentially amplify immunotherapy effectiveness. On the contrary, radio and chemotherapy can also induce lymphopenia and augment the expression of immunosuppressive molecules (PD-L1) within the TME, hence compromising the efficacy of immunotherapeutic treatments. The timing (adjuvant or neoadjuvant) for the best synergistic effect remains a problem to resolve. The implementation of multi-modal combined strategies is essential to increase survival and the quality of life of GB patients. The design of personalized treatments should be based on the genetic profiling and characteristics of this heterogeneous tumor, its specific location, the selection of the most promising targets, including the GSC, and the stimulation of the immune system. Personalized combined treatments offer advantages in terms of efficacy and reduced collateral damage, but the time needed to make decisions and cost represent significant challenges. The use of mathematical models of synergism/antagonism of drugs and pathways will help in predicting drug combinations for this multifactorial disease.We should reconsider what has not been done properly. To enroll patients in a trial, the genetic characteristics of their tumors should be considered, including only those that present the corresponding therapeutic targets. Only in this way will we ensure that results become objective and significant. Administering immunotherapies concurrently with corticosteroids or other immunosuppressive treatments is nonsensical. Such combinations must be considered when analyzing clinical study results to prevent misleading conclusions.TRT, HIFU, PDT and SDT can be alternatives (particularly in patients with poor physical condition), complements to radiotherapy, or become part of the SOC.

## Figures and Tables

**Figure 1 ijms-25-02529-f001:**
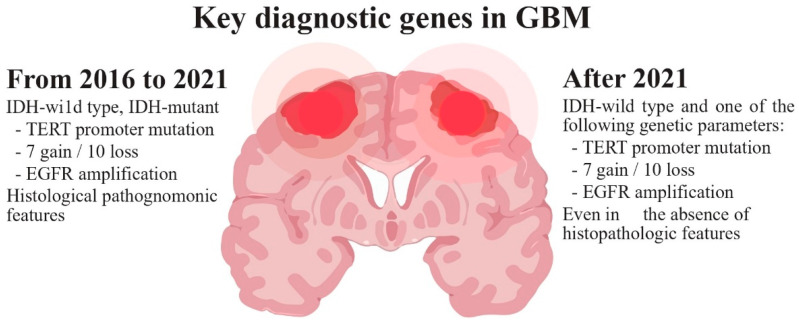
WHO diagnostic criteria for glioblastoma.

**Figure 2 ijms-25-02529-f002:**
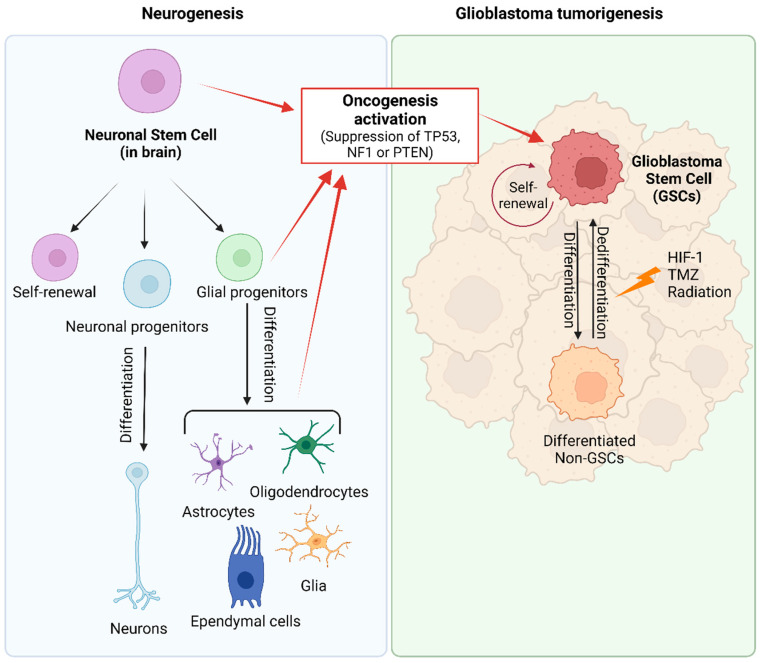
Origin of glioblastoma. During normal embryonic development and in the adult brain, neural stem cells (NSCs) generate glial and neuronal cells. Glioma stem cells (GSCs) arise from NSCs, astrocytes, oligodendrocytes, or glial precursor cells through the activation of oncogenic pathways (inactivation of TP53, NF1 or PTEN). GSCs are described as slow-dividing or quiescent cells, with multilineage differentiation capacity that allows them to differentiate into GB cells and cells with astrocytic, neuronal, and endothelial features and even trans-differentiation abilities. In GB tumors, there exists a dynamic equilibrium between quiescent and proliferative GSCs, and between GSC populations and their lineage-committed counterparts (differentiated non-GSC) that can also dedifferentiate into stem-lineage GSCs. Created with BioRender.com, accessed on 24 January 2024.

**Figure 3 ijms-25-02529-f003:**
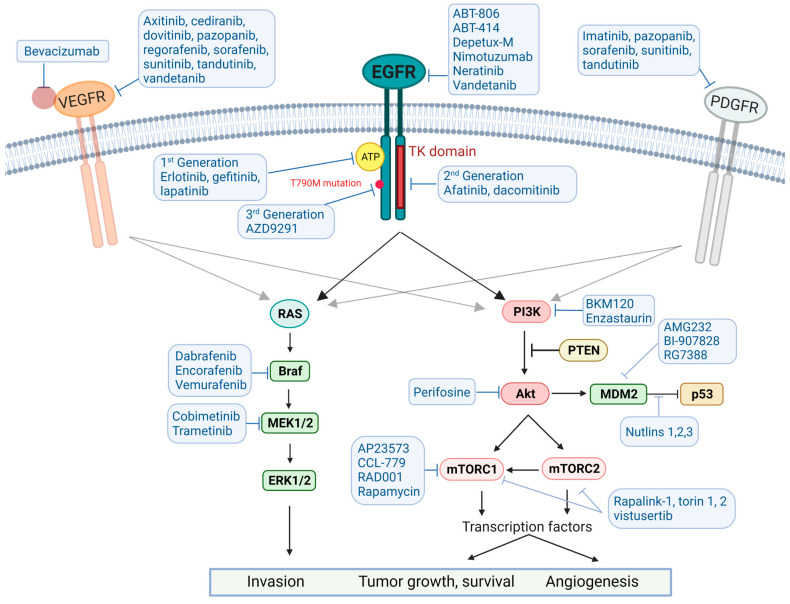
PTEN/PI3K/Akt/mTOR and RAS/RAF/MEK/ERK axis, and list of related chemotherapeutics clinically assayed against glioblastoma. Created with BioRender.com.

**Figure 4 ijms-25-02529-f004:**
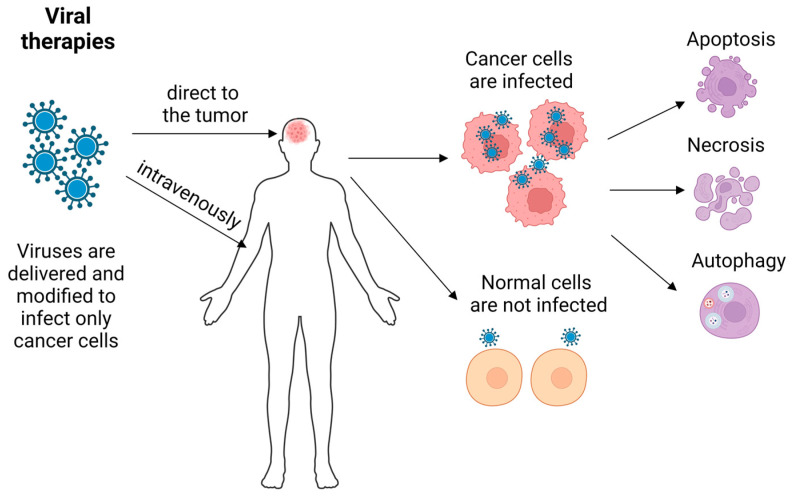
Viruses modified to infect only GB cells or to specifically induce cancer cell elimination can be delivered either directly into brain tumors or intravenously. Created with BioRender.com. Accessed on 15 January 2024.

**Table 1 ijms-25-02529-t001:** microRNAs: role in GB, functions, and targets.

Tumor-Suppressing			Oncogenic		
Name	Regulation	Targets	Name	Regulation	Targets
miRNA-7 [151,152,153,154]	Survival, proliferation, apoptosis, invasion, angiogenesis	FAK, EGFR, Akt, c-KIT, TGFβ2, CDK6, Akt2, LRRC4, YBX1, CD24, MTDH	miRNA-10b [148,155,156,157]	Survival, proliferation, apoptosis, invasion, stemness	HOXD10, uPAR, RhoC, PTEN, TFAP2C, BCL2L11, CDKN1A
miRNA-34 [148,158,159,160,161,162,163,164,165,166,167]	Survival, proliferation, apoptosis, migration, invasion, stemness	SIRT1, c-Met, Notch1/2, PDGFRA, Msi1, Akt and Wnt	miRNA-21 [168,169,170,171,172,173,174]	Survival, proliferation, apoptosis, migration, invasion, chemoresistance	HNRPK, TAp63, PDCD4, P53, TGFβ, MMPs, Ras/Raf, SPRY2, ANP32A, PTEN, SMARCA4, ERK, LRRFIP1
miRNA-128 [175,176,177,178,179,180,181,182,183,184,185,186]	Stemness, radioresistance, apoptosis, proliferation, angiogenesis	P70S6K1, SUZ12, BMI1, PDGFRA, EGFR, E2F3a, WEE1 and Msi1	miRNA-93 [179,187,188]	Survival, proliferation, angiogenesis, stemness	Integrin b8

**Abbreviations:** Akt (Akt serine/threonine kinase), Akt2 (Akt serine/threonine kinase 2), BCL2L11 (BCL2 like 11), BMI1 (BMI1 proto-oncogene, polycomb ring finger), CDKN1A (cyclin dependent kinase inhibitor 1A), CDK6 (cyclin-dependent kinases), CD24 (CD24 molecule), c-KIT (receptor tyrosine kinase), c-Met (MET proto-oncogene, receptor tyrosine kinase), EGFR (Epidermal growth factor receptor), E2F3a (E2F transcription factor 3), FAK (Focal adhesion kinase), HOXD10 (homeobox D10), Integrin b8 (Integrin b8), LRRC4 (leucine rich repeat containing 4), MMPs (matrix metalloproteinases) miR (miRNAs), MTDH (metadherin), Msi1 (musashi RNA binding protein 1), Notch1/2 (notch ½), P70S6K1 (ribosomal protein S6 kinase B1), PDGFRA (platelet derived growth factor receptor alpha), PTEN (phosphatase and tensin homolog), RhoC (ras homolog family member C), SIRT1 (sirtuin 1), SUZ12 (SUZ12 polycomb repressive complex 2 subunit), YBX1 (Y-box binding protein 1), TGFβ2 (transforming growth factor beta 2), uPAR (urokinase-type-plasminogen-activator receptor, WEE1 (WEE1 G2 checkpoint kinase), Wnt (Wingless/Integrated).

**Table 2 ijms-25-02529-t002:** Phase III clinical trials that support the SOC in ndGB patients.

Name/Trial Number	Treatment	No./Type of Patients	Outcome	Ref.
**EORTC-NCIC** NCT00006353	**RT** **RT+TMZ (SOC)**	573 ndGB patients	SOC had survival benefits in all groups, including patients ≥ 60 years mOS: 12.1 m (RT) vs. 14.6 m (RT+TMZ) OS (2,3,4,5 years): 10.9%, 4.4%, 3%, 1.9% (RT) vs. 27.2%, 16%, 12.1%, 9.8% (SOC)	[7,255]
**EF-14** NCT00916409	**SOC** **SOC+TTFields**	695 ndGB patients	The addition of TTFields to the SOC extended mOS (16 vs. 20.9 m) and mPFS (4.0 vs. 6.7 m)	[13,256]
NCT00304031	**RT+TMZ (SOC)** **RT+TMZ (DD)**	833 ndGB patients	No difference was found in mPFS and mOS (16.6 vs. 14.9 m) between both arms. GEINO14-01 (Spain) and EX-TEM (Australia) studies confirm the results of this trial	[257,258]
**CeTeG/NOA-09** NCT01149109	**SOC+Lomustine** **SOC**	129 ndGB or gliosarcoma patients with +MGMT	Combined treatment might improve mOS (48.1 vs. 31.4 m), but these findings should be interpreted with caution, owing to the small size of the trial	[259]
**AVAglio** NCT00943826	**SOC+BEV** **SOC**	921 supratentorial ndGB patients	The addition of BEV to SOC increased in 4.4 mo. the mPFS but not mOS (16.8 vs. 16.7 m). Maintenance of baseline QoL and performance status were observed with BEV, but with higher rate of AE	[260,261]
**RTOG 0825** NCT00884741	**SOC+BEV** (adjuvant) **SOC**	637 ndGB patients	The addition of BEV to SOC increased the PFS in 7.3 m, without differences in mOS. Higher rates of neurocognitive decline, symptom severity, and decline in health related QoL affected patients treated with BEV + SOC.	[262,263]
NCT00017147	**O^6^BG**+BCNU+**RT** BCNU+**RT**	179 ndGB or gliosarcoma patients	The addition of O^6^BG did not provide benefit and caused additional toxicity. Significantly more grade 4/5 AE in the experimental arm. mOS: 11 m (O^6^BG + BCNU + RT) vs. 10 m.	[264]
**NOA-08-Trial** NCT01502241	**TMZ** (100 mg/m^2^) **RT** (60 Gy, 2 Gy per fr.).	412 patients with anaplastic astrocytoma or ndGB ≥65 years	TMZ or RT alone render similar results (mOS: 8.6 vs. 9.6 m). +MGMT patients had better responses with TMZ, vs. −MGMT patients that better responded to RT.	[265]
ISRCTN81470623	**TMZ** **Hypofractionated RT** (34 Gy, 3–4 Gy, fr. over 2 weeks) **Standard RT** (60 Gy, 2 Gy fr. over 6 weeks)	291 patients with ndGB > 60 years	Standard RT was associated with poor outcomes, especially in patients > 70 years. Similar OS (8.4 vs. and 7.4 m) in TMZ vs. hypofractionated RT. Both strategies should be considered as standard treatment options in the elderly. +MGMT status is a predictive marker for TMZ-derived benefits.	[266]
NCT00482677	**RT**(40 Gy/15 fr.)+ **TMZ** **RT alone** (40 Gy/15 fr.)	562 ndGB patients ≥ 65 years	PFS (5.3 vs. 3.9 m) and (OS 9.3 vs. 7.6 m) were longer if TMZ was added to RT vs. RT alone. QoL was similar in both groups.	[267]

**Abbreviations:** AE, adverse events; BCNU, carmustine; DD, dose-dense; fr., fraction; +MGMT, methylated MGMT promoter; -MGMT, unmethylated MGMT promoter; m, months; ndGB, new diagnosed glioblastoma; PFS, progression-free survival; O^6^-BG, O^6^-benzylguanine; OS, overall survival; QoL, quality of live; SOC, standard of care treatment; TMZ, Temozolomide; TTFields, tumor treating fields; RT, radiation therapy.

**Table 3 ijms-25-02529-t003:** Selected phase II/II trials for target therapies in GB.

Trial Number	Treatment Assayed	Trial Information	Outcomes	Ref.
NCT02977780	**Abemaciclib+SOC** **Neratinib+SOC** **CC-115+SOC**	Phase II in ndGB with -MGMT	Addition of abemaciclib or neratinib to the SOC was well-tolerated, but CC-115 associated with ≥grade 3 related toxicities in 58% of patients. An increase in mPFS was observed in the first two treatments and lack of efficacy in CC-115. None of the therapies demonstrated a benefit in OS.	[402,403]
NCT00977431	**Afatinib** **TMZ** **Afatinib+TMZ**	Phase II in rGB	Afatinib monotherapy had limited activity, but it increased mPFS in patients overexpressing EGFR or expressing EGFRvIII. Addition of afatinib to TMZ did not improve PFS6 rate (3% vs. 23% vs. 10%) or mPFS. There were no significant differences in OS across the study arms.	[404]
NCT03158389	**Alectinib, Idasanutlin, Palbociclib, Vismodegib, Temsirolimus**	Phase I/IIa in ndGB with -MGMT	The N^2^M^2^ trial is investigating a number of different targeted compounds in combination with RT. No results have been published yet.	[405]
NCT02029573	**Atorvastatin+SOC**	Phase II in ndGB	Atorvastatin did not improve PFS6. High LDL levels seem to be associated with poor outcomes.	[406]
NCT03291314	**Axitinib**+**avelumab**	Phase II in rGB	Efficacy of the combination is similar to axitinib monotherapy. Addition of avelumab does not show synergistic efficacy. Axitinib confirms its role as a potent corticoid-sparing option to control tumor-related edema.	[407]
NCT01562197	**Axitinib** **Axitinib**+**lomustine**	Phase II for rGB	Axitinib’s tolerability and clinical outcomes match those of BEV. However, axitinib offers advantages like oral dosing and a short half-life for quick reversibility in urgent interventions. Combined treatment had no benefits. mPFS6 was 26% vs. 17%.	[408]
NCT00967330	**BEV** during **RT** followed by **BEV+IRI**	Phase II in ndGB with -MGMT	BEV+ Irinotecan did not alter QoL and resulted in a superior PFS6 rate and mPFS compared with TMZ.	[409]
NCT00345163	**BEV+Irinotecan** **BEV**	Randomized phase II in rGB	BEV alone or in combination with irinotecan, was well tolerated and active in rGB. Estimated PFS6 rates were 42.6% and 50.3%, respectively; the mOS were 9.2 months and 8.7 months, respectively.	[410]
NCT00817284	**BEV+irinotecan+SOC** **BEV+SOC**	Randomized phase II in ndGB	Results did not indicate any benefit from BEV-irinotecan-SOC in first-line therapy as opposed to BEV-SOC in terms of response and PFS.	[411]
NCT00979017	**BEV+irinotecan+TMZ** in combination with **RT**	Phase II for unresectable GB	Combined treatment was tolerable leading to radiographic response in unresectable and/or subtotally resected GB.	[412]
NCT01290939	**BEV+lomustine** **BEV** **Lomustine**	Phase III in rGB	BEV+lomustine improved mOS (12 vs. 8 vs. 8 months) compared to either monotherapy in the prior phase II trial. In this study, despite a somewhat prolonged PFS, there was no mOS advantage over lomustine alone (9.1 vs. 8.1 months).	[377,387]
NCT01632228	**BEV+onartuzumab**	Phase II in rGB	No survival benefits were evidenced with the combined treatment. mOS was 8.8 months (BEV+onartuzumab) vs. 12.6 months (BEV+placebo).	[413]
NCT01730950	**BEV+RT**	Randomized phase II in rGB	Re-irradiation was shown to be safe and well tolerated. BEV + RT improved PFS6 rate but no difference in OS was found.	[414]
NCT00667394	**BEV+tandutinib**	Phase II in rGB.	The efficacy of combination therapy was comparable to that of BEV monotherapy, but more toxic.	[415]
NCT00883298	**BEV+TMZ**	Phase II in rGB	Combined treatment showed efficacy and was well tolerated. The PFS6 and 12-month OS rates were 62.5% and 31.3%, respectively. mOS was 37.1 weeks. The benefits were similar to BEV monotherapy.	[416]
NCT01738646	**BEV+vorinostat**	Phase II in recurrent grade 4 glioma	Combined treatment was well tolerated, but not improved PFS6 or mOS vs. BEV monotherapy.	[217]
NCT01339052	**Buparlisib**	Phase II in rGB with PI3K pathway-activated	Although buparlisib achieved significant brain penetration, it had minimal efficacy in patients with PI3K-activated rGB.	[417]
NCT01349660	**Buparlisib+BEV**	Phase II in relapsed/refractory GB	The combination was poorly tolerated, with frequent dose interruptions, and showed no greater efficacy than BEV alone	[418]
NCT01934361	**Buparlisib+carboplatin** **Buparlisib+lomustine**	Randomized phase Ib/II in rGB	None of the combinations significantly improved the antitumor activity compared with historical data on single-agent carboplatin or lomustine.	[419]
NCT01870726	**Buparlisib+INC280**	Phase Ib/II in PTEN-deficient GB	No improvement was evidenced in the combined treatment.	[420]
NCT01062425	**Cediranib** **Placebo**	Randomized phase II in ndGB	PFS6 was 46.6% vs. 24.5% (placebo). No difference in OS between the 2 arms. Cediranib had more AEs (≥ 3) than placebo.	[421]
NCT00777153	**Cediranib** **Cediranib+lomustine** **Lomustine**	Phase III in rGB	Cediranib monotherapy or in combination did not improve PFS or mOS (8.0, 9.4, and 9.8 months, for each group). Cediranib had corticosteroid-sparing effects and clinical activity on deterioration of neurologic function.	[422]
NCT00311857	**Cetuximab+SOC**	Phase I/II in ndGB	Early data from trimodal therapy indicated feasibility without an increased toxicity profile. OS was 87% at 12 months. +MGMT was not associated with longer OS.	[423]
NCT00463073	**Cetuximab+BEV+** **irinotecan**	Phase II for rGB	Patients with an EGFR amplification lacking EGFRvIII expression had a significantly superior PFS (3.03 vs. 1.63 months) and OS (5.57 vs. 3.97 months). However, the efficacy was not superior to the combination BEV+irinotecan.	[424,425]
NCT02974621	**Cediranib+olaparib** **BEV**	Randomized phase II for rGB.	Treatment with cediranib + olaparib failed to increase PFS and OS in rGB patients.	[426]
NCT01520870	**Dacomitinib**	Phase II in rGB with EGFR amplification or EGFRvIII	Dacomitinib had minimal activity but, a subset of patients (4.1%) experienced a durable, clinically meaningful benefit.	[427]
NCT00423735	**Dasatinib**	Phase II in target-selected patients with rGB	Dasatinib was ineffective in rGB	[428]
NCT02343406	**Depatux-M+TMZ** **Depatux-M** **Lomustine or TMZ**	Randomized phase II in rGB with EGFR amplification	OS: 19.8% vs. 5.2% (control group) vs. 10% (depatux-M). Depatux-M had no impact on QoL, except for more visual disorders.	[429]
NCT02573324	**Depatux-M+SOC**	Phase III in ndGB with EGFR amplification	Combined treatment enhanced PFS (8.0 vs. 6.3 months), particularly among patients with EGFRvIII-mutant (8.3 vs. 5.9 months) or -MGM tumors. However, no significant OS benefits were evidenced. Most of depatux-M-treated patients had corneal epitheliopathy (61% grade 3–4).	[430]
NCT01753713	**Dovitinib**	Phase II in rGB	Dovitinib was not efficacious in prolonging the PFS irrespective of prior treatment with anti-angiogenic therapy.	[431]
NCT02034110	**Dabrafenib+trametinib**	Phase II in BRAF^V600E^-rare cancers	Dabrafenib plus trametinib showed clinically meaningful activity (mOS 13.7 months) in high-grade glioma patients with BRAF^V600E^ mutation. Limited significance of result due to the small sample.	[432]
NCT00295815	**Enzastaurin** **Lomustine**	Phase III in rGB	Enzastaurin was well tolerated and had a better hematologic toxicity profile than Lomustine but did not show superior efficacy.	[433]
NCT00337883	**Erlotinib**	Phase II in rGB assessing if response was related to concomitant use of EIAEDs	Erlotinib had limited activity. However, in the EIAED subgroup, the PFS6 reached or exceeded historical survival values.	[434]
NCT00086879	**Erlotinib** Control: **TMZ** or **BCNU**	Phase II in rGB	Erlotinib had insufficient single-agent activity. PFS6: 11.4% vs. 24.1% in the control arm. Both, PFS and OS were worse in the patients with EGFRvIII in the erlotinib arm.	[435]
NCT00671970	**Erlotinib+BEV**	Phase II in rGB based on VEGF/EGFRvIII expression	Combined treatment improved mPFS (6.75 vs. 5.5 months) and mOS (17.0 vs. 6.75 months). The significance of the results is limited due to the small size of the cohort.	[436]
NCT00720356	**Erlotinib+BEV** (after completion of SOC)	Phase II in -MGMT ndGB or gliosarcoma	The combination of erlotinib and BEV was tolerable but did not meet the primary endpoint of increased survival.	[437]
NCT00187486	**Erlotinib+SOC**	Phase II in ndGB and gliosarcoma	Erlotinib plus SOC significantly improved OS (19.3 vs. 14.1 months) with respect to historical controls. There was a strong positive correlation between +MGMT and survival.	[438]
NCT00274833	**Erlotinib+SOC**	Phase II in ndGB	The addition of erlotinib to SOC was not efficacious and had an unacceptable toxicity.	[439]
NCT0062243	**Erlotinib+sirolimus**	Phase II in rGB	Erlotinib plus sirolimus was well tolerated but had negligible activity in rGB.	[440]
NCT00445588	**Erlotinib+sorafenib**	Phase II in progressive and rGB	Erlotinib and sorafenib have significant pharmacokinetic interactions that may negatively impact the efficacy of the combination regimen.	[441]
NCT01062399	**Everolimus+SOC**	Phase II in ndGB	Combining everolimus with conventional SOC leads to increased toxicities and had no survival benefit	[442]
NCT00805961	**Everolimus+BEV+SOC**	Phase II in ndGB	The addition of everolimus and BEV to SOC was feasible, deleting the maintenance with TMZ. The PFS compared favorably to previous reports with SOC but was similar to results achieved in other trials in which BEV was added to first-line treatment.	[443]
NCT00250887	**Gefitinib**	Phase II in rGB	Gefitinib reaches the tumor in high concentrations, efficiently dephosphorylates the target, but does not affect GB growth.	[444]
NCT00016991	**Gefitinib**	Phase II in GB at first relapse	Tolerable and modest activity. 56.6% of the patients suffered therapy failure within the initial 8-week assessment period.	[445]
NCT01310855	**Gefitinib+ cediranib** **Cediranib**	Randomized phase II in rGB	Combined treatment was tolerated and showed a tendency to improve PFS and response rates. mOS = 7.2 vs. 5.5 months in cediranib arm. Incomplete recruitment led to the study being underpowered.	[446]
NCT00014170	**Gefitinib+RT**	Phase II in ndGB	The addition of gefitinib to RT was well tolerated, but there were no significant differences in PSF as compared to historical control cohorts treated with RT alone.	[447]
NCT00171938	**Imatinib and/or Hypofractionated RT**	Phase II for inoperable/incompletely resected GB expressing PDGFR	Imatinib showed no measurable efficacy.	[448]
NCT00154375	**Imatinib+hydroxyurea** **Hydroxyurea**	Phase III in TMZ- resistant progressive GB	No clinically meaningful differences were found between the two treatment arms (PFS6 5% vs. 7% in the monotherapy) and the primary study end point was not met.	[449]
NCT01268566	**MEDI-575**	Phase II in rGB	MEDI-575 was well tolerated but showed limited clinical activity	[450]
NCT00753246	**Nimotuzumab+SOC**	Randomized phase III in ndG	Nimotuzumab was well tolerated, but its combination with the SOC had no significant survival benefits (22.3 vs. 19.6 months in SOC)	[451]
NCT01227434	**Palbociclib** **Palbociclib w/o resection**	Phase II in RB1-positive rGB	Palbociclib was administered 7 days prior to resection and the analysis of the surgical samples demonstrated tumor biologically effective concentrations, but there was no reduction in RB1 expression or decrease in cell proliferation, thus lacking efficacy.	[399]
NCT00459381	**Pazopanib**	Phase II in rGB	Pazopanib did not prolong mPFS (12 weeks) but showed in situ biological activity as demonstrated by the radiographic responses.	[452]
NCT01051557	**Perifosine**	Phase II in rGB	Perifosine is tolerable but ineffective as monotherapy for GB.	[453]
NCT02926222	**Regorafenib** **Lomustine**	Randomized phase II in rGB	mOS was significantly improved in the regorafenib group (7.4 months) compared with the lomustine group (5.5 months). Despite the criticisms of the study and the high rate of AEs, the encouraging results led regorafenib to become the SOC in Italy.	[454]
NCT00621686	**Sorafenib+BEV**	Phase II in rGB	No improvement in the outcome of patients.	[455]
NCT00597493	**Sorafenib+TMZ** (LD)	Phase II in rGB	Combined treatment was feasible and safe, showing some efficacy. Patients: 12% partial response, 43% stable disease, 48% progression. PFS6 was 26% and mOS was 7.4 months.	[456]
NCT00544817	**Sorafenib+SOC**	Phase II in ndGB	The addition of sorafenib did not improve the efficacy of SOC	[457]
NCT00535379	**Sunitinib**	Phase II in rGB	Minimal anti-GB activity and substantial toxicity when sunitinib was given at higher doses. c-KIT expression in vascular endothelial cells was associated with improved PFS.	[458]
NCT01100177	**Sunitinib**	Phase II in non-resectable ndGB	No efficacy as monotherapy.	[459]
NCT01100177	**Sunitinib+SOC**	Phase II in ndGB with -MGMT	The addition of sunitinib modestly increased OS. Sunitinib potentially sensitizes -MGMT GB to adjuvant TMZ.	[460]
NCT00022724	**Temsirolimus**	Phase II in rGB	Temsirolimus was well tolerated, but there was no evidence of efficacy.	[461]
NCT00016328	**Temsirolimus**	Phase II in rGB	Temsirolimus was well tolerated. Radiographic improvement was observed in 36% of patients and was associated with significantly longer median time to progression.	[462]
NCT01019434	**Temsirolimus**	Phase II in ndGB with -MGMT	Temsirolimus showed no clinical benefit compared to TMZ.	[463]
NCT00800917	**Temsirolimus+BEV**	Phase II in rGB	Temsirolimus + BEV treatment was well tolerated but did not improve BEV monotherapy.	[464]
NCT00821080	**Vandetanib**	Phase I/II in recurrent malignant glioma	Vandetanib did not have significant activity in unselected patients with recurrent malignant glioma. mOS was 6.3 months in the GB arm. Seizures were an unexpected toxicity of therapy.	[465]
NCT00272350	**Vandetanib+SOC**	Phase II in ndGB or gliosarcoma	The addition of vandetanib was reasonably well tolerated. However, the regimen did not prolong OS (16.6 months) compared with SOC (15.9 months).	[466]
NCT01026493	**Veliparib**	Randomized phase I/II in TMZ resistant rGB	Combined treatment did not significantly improve PFS6 in the BEV-naïve or BEV-failure patients who were previously treated with TMZ.	[467]
ACTRN12615000407594	**Veliparib+SOC**	Randomized phase II in ndGB with -MGMT	Combined treatment showed no clinical benefit. PFS6: 46% (experimental arm) vs. 31% in SOC. mOS = 12.7 months (experimental arm) vs. 12.8 months (SOC).	[468]
NCT02152982	**Veliparib+SOC**	Phase II/III in +MGMT ndGB	There was no significant difference in mOS (28.1 vs. 24.8 months in the SOC-treated group).	[469]

**Abbreviations:** AEs, adverse events; BCNU, carmustine; BEV, bevacizumab; EGFR, epidermal growth factor receptor; EIAEDs, CYP3A4 enzyme-inducing antiepileptic drugs; GB, glioblastoma; LD, low dose; +MGMT, methylated MGMT promoter; -MGMT, unmethylated MGMT promoter; mOS, median overall survival; mPFS, median progression-free survival; ndGB, new diagnosed glioblastoma; QoL, quality of life; PFS6 progression-free survival at 6 months; rGB, recurrent glioblastoma; TMZ temozolomide; VEGFR vascular endothelial growth factor receptor; SOC, standard of care treatment; RT, radiotherapy.

**Table 4 ijms-25-02529-t004:** Selected phase II/III clinical trials on immunotherapy for GB.

Trial Number	Treatment	Trial Information	Outcome	Ref.
	**Allogenic DC vaccination**	Phase II for GB or grade 4 astrocytoma	OS (27.6 ± 2.4 months) was 75% greater in the vaccinated GB group.	[666]
NCT01213407	**Audencel** (tumor lysate-charged autologous DCs)**+SOC**	Randomized phase II in ndGB	Combined treatment had no clinical benefits. PFS (28.4% vs. 24.5%) and mOS (≈18.3 months) did not differ significantly between control and vaccine groups.	[667]
NCT01006044	**Autologous DCV** (in tumor resection)+**SOC**	Phase II in ndGB	Treatment was feasible and safe. Increase in specific immune response (proliferation or cytokine production) was detected in 11/27 evaluated patients. No correlation between immune response and survival was found. mPFS and mOS were 12.7 and 23.4 months, respectively.	[668]
NCT03400917	**AV-GBM-1**	Phase II in ndGB	AV-GBM-1 treatment had numerous AEs affecting the CNS. mPFS was longer than historical benchmarks, but no mOS improvement was noted. mPFS and mOS from ITT enrollment were 10.4 and 16.0 months, respectively. Two-year OS was 27%.	[669]
NCT00639639	**Cytomegalovirus Pp65+SOC+TMZ (DD)**	Randomized phase II in ndGB	Patients showed markedly prolonged mPFS and mOS (25.3 and 41,1 months) compared with historical controls (8.0 and 19.2 months). Four patients remained progression-free 5 years later.	[670]
NCT00045968	**DCVax-L+SOC (ndGB)** **DCVax-L (rGB)**	Phase III in progressive ndGB or rGB	Significant extension of survival compared with matched external controls. mOS in ndGB (from randomization): 19.3 vs. 16.5 months mOS in rGB (from relapse): 13.2 vs. 7.8 months.	[671]
NCT01567202	**DCV loaded with GSC antigens**	Phase II for ndGB or rGB	Extension in OS to 13.7 months, up from 10.7 months. Low B7-H4 expression identified subgroups of GB patients more responsive to treatment.	[672]
NCT02798406	**DNX-2401+ pembrolizumab**	Phase II rGB	DNX-2401 followed by pembrolizumab was well tolerated with notable survival benefit in selected patients. mOS was 12.5 months; OS at 12 and 18 months was 54.5% and 20.8%, respectively.	[673,674]
UMIN000001426 (Japan Clinical trials)	**Fractionated RT+ TMZ+AFT**	Phase I/II in ndGB	The treatment was well tolerated and resulted in favorable survival. mPFS, mOS, 2- and 3-year survival rates were: 8.2 months, 22.2 months, 47%, and 38%, respectively.	[675]
UMIN000015995 (Japan Clinical trials)	**G47∆ Herpes virus**	Phase II in rGB	G47∆ had a significant survival benefit and good safety profile. mOS was 20.2 months after G47∆ initiation. Biopsies revealed increasing numbers of tumor-infiltrating CD4+/CD8+ lymphocytes and persistent low numbers of Foxp3+ cells.	[676]
2004-000464-28 (EU Clinical trials)	**Ganciclovir+sitimagene ceradenovec+ SOC**	Randomized phase III trial in ndGB	Ganciclovir and sitimagene ceradenovec after resection can increase time to death or to re-intervention, although the treatment did not improve OS (16.32 vs. 14.85 in controls treated with SOC).	[677]
NCT00293423	**HSPPC-96 vaccine**	Phase II in rGB	HSPPC-96 vaccine showed considerable efficacy in terms of OS (10.65 months) and PFS (4.78 months). The improved outcomes were associated with the number of doses and a higher absolute lymphocyte count.	[678,679]
NCT01814813	**HSPPC-96 vaccine+BEV**	Phase II in rGB	The study failed to demonstrate a survival benefit for patients treated with HSPPC-96 alone or in combination with BEV compared to BEV alone.	[680]
NCT01280552	**ICT-107 DCV**	Phase II in ndGB	PFS increased by 2.2 months, but no significant improvement in OS was shown. The HLA-A2 patient group had higher levels of immune response than the HLA-A1 patient group	[681]
ISRCTN84434175	**Ipilimumab+SOC**	Randomized phase II in ndGB	No improvement in PFS or OS was observed with the addition of ipilimumab to the SOC.	[682]
NCT02017717	**Nivolumab** **BEV**	Randomized phase III in rGB	mOS (13.4 vs. 14.9 months) was comparable between nivolumab and BEV in rGB patients. The objective response rate was higher with BEV (23.1%) vs. nivolumab (7.8%).	[663]
NCT02550249	**Nivolumab** administered pre- and post-surgical intervention.	Phase II for ndGB or rGB that required salvage surgical resection	No obvious survival benefit was substantiated (mPFS and OS were 4.1 and 7.3 months, respectively, in evaluable patients,). Notably, 3 patients treated with nivolumab remained alive after long-term follow-up.	[683]
NCT02667587	**Nivolumab+SOC**	Phase III for ndGB with +MGMT	Nivolumab added to SOC did not improve mOS (28.9 months vs. 32.1 months).	[664]
NCT02617589	**Nivolumab+SOC**	Phase III in ndGB with -MGMT	Nivolumab added to SOC did not improve mOS (13.4 months vs. 14.9 months).	[665]
NCT02337491	**Pembrolizumab vs. BEV**	Phase II in rGB	Pembrolizumab was well tolerated but was ineffective as monotherapy over BEV.	[662]
NCT03405792	**Pembrolizumab+ TTFields+TMZ** (maintenance)	Phase II in ndGB	The triple combination was well tolerated and demonstrated promising efficacy. mOS was 24.8 vs. 14.7 months in controls. 2-year OS was 52.4% vs. 12.0% in controls. Several patients (40%) with measurable disease achieved a partial to complete response.	[684]
NCT03532295	**Retifanlimab+ BEV+ Hypofractionated RT**	Phase II in rGB	Combined treatment was well-tolerated and had an encouraging OS of 9 months (71.4%) and PFS at the time of data cutoff.	[685]
NCT01498328	**Rindopepimut+ BEV**	Randomized phase II in rGB	PFS6 was 27% and the mOS was 12 months, which means a significant improvement compared with controls (11% and 8.8 months, respectively)	[686]
NCT00458601	**Rindopepimut+ SOC**	Phase II in ndGB expressing EGFRvIII	Anti-EGFRvIII antibody titers increased ≥4-fold in 85% of patients. PFS at 8.5 months from diagnosis was 66% and 3-year OS was 26%.	[687]
NCT01480479	**Rindopepimut+** **SOC** **SOC**	Randomized phase III in ndGB expressing EGFRvIII	The addition of rindopepimut to SOC did not increase mOS, which was ≈20.0 months.	[688]
NCT02455557	**SurVaxM+SOC**	Phase IIa in ndGB	No serious AEs were attributable to SurVaxM were shown. mPFS and mOS were 11.4 and 25.9 months, measured from first dose of SurVaxM. Specific immune response was evidenced.	[689]
NCT02414165	**Toca 511+ flucytosine** Control: **Lomustine, TMZ, or BEV**	Randomized phase II/III in astrocytoma or rGB	Toca 511 + flucytosine did not improve OS (11.1 vs. 12.22 months in control) or other efficacy end points.	[690]
NCT02511405	**VB-111+BEV** **BEV**	Randomized phase III in rGB	Previous phase II trial (NCT01260506) showed a significant improvement for VB-111 as monotherapy. On the contrary, here, dual administration of VB-111 and BEV failed to improve outcomes, and 67.5% of the patients developed grade ≥3 adverse reactions. The differences were attributed to changes in treatment regimen.	[691,692]

**Abbreviations**: AEs, adverse events; AFT; autologous formalin-fixed tumor vaccine; BEV, bevacizumab; DCV, dendritic cell vaccine; EGFRvIII, epidermal growth factor receptor variant III; GB, glioblastoma; +MGMT, methylated MGMT promoter; -MGMT, unmethylated MGMT promoter; mOS, median overall survival; mPFS, median progression-free survival; ndGB, new diagnosed glioblastoma; QoL, quality of life; PFS6, progression-free survival at 6 months; rGB, recurrent glioblastoma; TMZ temozolomide; TTFields, tumor treating fields; SOC, standard of care treatment; RT, radiotherapy.

## Data Availability

No new data were created or analyzed in this study. Data sharing is not applicable to this article.

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
