# Peer review of "Glioblastoma Therapy: Past, Present and Future"

_ijms, 2024, doi:10.3390/ijms25052529_

Round 1
Reviewer 1 Report
Comments and Suggestions for Authors
Frst that is a very impressive review that gathers all the possible data about glioblastoma therapy. However, it is over 90 pages long and over 1000 references, which is more than most books in that area. Maybe try to publish it as a book instead?
Otherwise, the paper is great but has several errors which need to be corrected before it may be accepted for publication in IJMS:
1. Line 2316 - you shouldn't cite the online web pages as they might disappear soon - please find any validation in the scientific resources. (applies to the whole text).
2. Line 2049 you mention the clinical trials - it is good to summarize them and provide a table with all the important data for each separate section.
3. Figures are in low resolution - can you upload high definition ones?
4. Line 772 "Other therapeutic options for rGB, most under clinical trials (see below), have been recently reviewed in [390,391]." so what is the aim of your work if such comprehensive reviews are already available? Please give some reasoning for such paper in the introduction.
Author Response
First of all, we appreciate your comments and suggestions very much. This revised version, thanks to your reviewing, is definitely better than the original. All pages indicated refer to Manuscript ID: ijms-2815021-R1.
- Line 2316 - you shouldn't cite the online web pages as they might disappear soon - please find any validation in the scientific resources. (applies to the whole text).
In the original version, the online web pages cited have been substituted by the indicated refs.:
www.novocure.com (under 3.2. Tumor-Treating Fields (TTFields)) see page 20.
Tran DD, Ghiaseddin AP, Chen DD, Le SB. Final analysis of 2-THE-TOP: A phase 2 study of TTFields (Optune) plus pembrolizumab plus maintenance temozolomide (TMZ) in patients with newly diagnosed glioblastoma. J Clinl Oncol. 2024, 41 (16, Supplem.).
www.fusfoundation.org (under 7.2. Focused Ultrasounds (LIFUs and HIFUs), see page 66.
At present, a number of clinical trials (NCT03551249, NCT04440358, NCT04417088, NCT05370508, NCT06039709) are ongoing or recruiting patients with GB for focused ultrasound treatment (ClinicalTrials.gov, February 4, 2024).
www.adacap.com (under 7.5. Targeted Radionuclide Therapy), see page 70.
Bolcaen J, Kleynhans J, Nair S, Verhoeven J, Goethals I, Sathekge M, Vandevoorde C, Ebenhan T. A perspective on the radiopharmaceutical requirements for imaging and therapy of glioblastoma. Theranostics. 2021 Jul 6;11(16):7911-7947. doi: 10.7150/thno.56639. PMID: 34335972; PMCID: PMC8315062
2. Line 2049 you mention the clinical trials - it is good to summarize them and provide a table with all the important data for each separate section.
We have followed your advice and 3 new tables have been added to the revised version.
Table 2. Phase III Trials that support the actual SOC in ndGB patients. Pages 15-16.
Table 3: Selected Phase II/III Trials for Target Therapies in GB. Pages 23-28.
Table 4. Selected Phase II/III Trials on Immunotherapy for GB. Pages 41-44.
3. Figures are in low resolution - can you upload high-definition ones?
High resolution and corrected Figs. have been added to the revised version.
4. Line 772 "Other therapeutic options for rGB, most under clinical trials (see below), have been recently reviewed in [390,391]." so what is the aim of your work if such comprehensive reviews are already available? Please give some reasoning for such paper in the introduction.
Our revised review is updated, the discussion is more detailed, and the therapeutic options for rGB are analyzed more in depth. Thus, in the attached revised version, we have deleted the indicated refs.
Reviewer 2 Report
Comments and Suggestions for Authors
This review examines the challenges in treating glioblastoma (GB), a highly aggressive brain tumor. Standard therapies offer limited success, prompting exploration of targeted, immunotherapeutic, nanotherapeutic, and non-ionizing energy-based approaches. Despite efforts, GB's adaptive nature and immune evasion remain formidable, necessitating ongoing research and clinical trials for innovative treatments and potential breakthroughs. The aim of this review is to discuss the advances and limitations of the current therapies, and to present novel approaches which are under development or following clinical trials.
The review is significant because it examines the progress or limitations made on a crucial issue. Although this is a lengthy and thorough review, there are a few things that I feel are still missing. The review should be published after incorporating my suggestions and recommendations.
1. The abstract looks like an introduction section with mostly common information. Authors are encouraged to emphasize key issues and the literature covered in this manuscript to enhance readers' comprehension of the scope of this extensive review article.
2. Considering the substantial length of this review, inclusion of a detailed table of contents section would enhance its accessibility and assist readers in efficiently navigating through the extensive content.
3. It is recommended to label Figure 1 and avoid relying solely on photographs.
4. The conclusion section requires substantial refinement. Currently, it primarily consists of background information on glioblastoma (GB) rather than addressing the stated objective of highlighting advancements, limitations, and future prospects in GB therapy, as outlined in the title "Glioblastoma Therapy: Past, Present, and Future." It is strongly recommended to revise the conclusion by emphasizing key advancements, limitations, and prospects in the realm of GB treatment.
5. In Section 7, dedicated to "Non-Ionizing Energies in GB Therapy," it would be valuable for the authors to incorporate a discussion on a recent study (https://doi.org/10.3389/fcell.2023.1067861) that explores the utilization of nanosecond pulsed high-power microwave irradiation to induce apoptosis in brain cancer cells.
6. Microwaves are recognized to exert notable effects on the human brain, with outcomes being either beneficial or detrimental depending upon factors such as microwave type, frequency, dosage, and power. It is important to discuss these details within the review article, underscoring the potential for achieving favorable effects or mitigating harm through the optimization of these parameters.
Hence, author should add some discussion on microwave radiations and brain in section 7 "Non-Ionizing Energies in GB Therapy.”
7. Incorporating a discussion on the potential use of natural products or flavonoids for the treatment of brain tumors would be beneficial for the manuscript. Addressing relevant studies and findings in this area could provide additional insights into alternative therapeutic approaches.
8. The manuscript contains several typos and grammatical errors. I recommend thorough proofreading to identify and correct these issues.
Comments on the Quality of English LanguageThe manuscript contains several typos and grammatical errors. I recommend thorough proofreading to identify and correct these issues.
Author Response
First of all, we appreciate your comments and suggestions very much. This revised version, thanks to your reviewing, is definitely better than the original. All pages indicated refer to Manuscript ID: ijms-2815021-R1.
1. The abstract looks like an introduction section with mostly common information. Authors are encouraged to emphasize key issues and the literature covered in this manuscript to enhance readers' comprehension of the scope of this extensive review article.
We have followed your suggestion and have changed the abstract as follows:
“Glioblastoma (GB) stands out as the most prevalent and lethal form of brain cancer. Although great efforts have been made by clinicians and researchers, no significant improvement in survival has been achieved since the Stupp protocol became the standard of care (SOC) in 2005. Despite multimodality treatments, recurrence is almost universal with survival rates under 2 years after diagnosis. Here, we discuss the recent progress in our understanding of GB pathophysiology, in special, the importance of glioma stem cells (GSCs), the tumor microenvironment conditions, and epigenetic mechanisms involved in GB growth, aggressiveness and recurrence. The discussion on therapeutic strategies first covers the SOC treatment and targeted therapies that have been tested to interfere at different signaling pathways (pRB/CDK4/RB1/P16ink4, TP53/MDM2/P14arf, PI3k/Akt-PTEN, RAS/RAF/MEK, PARP) involved in GB tumorigenesis, pathophysiology, and treatment resistance acquisition. Below, we analyze several immunotherapeutic approaches (i.e., checkpoint inhibitors, vaccines, CAR-T cells and NK cell-based therapy, oncolytic virotherapy) that have tried to enhance the immune response against GB, and thereby to avoid recidivism or increase survival of GB patients. Finally, we present treatment attempts made using nanotherapies (nanometric structures having active anti-GB agents such as antibodies, chemotherapeutic/anti-angiogenic drugs or sensitizers, radionuclides, and molecules which target GB cellular receptors or open the blood brain barrier), and non-ionizing energies (laser interstitial thermal therapy, high/low intensity focused ultrasounds, photodynamic/sonodynamic therapies and electroporation). The aim of this review is to discuss the advances and limitations of the current therapies, and to present novel approaches which are under development or following clinical trials”.
2. Considering the substantial length of this review, inclusion of a detailed table of contents section would enhance its accessibility and assist readers in efficiently navigating through the extensive content.
Thank you for the suggestion. We have included a table of contents and an index to each of the sections. See page 2.
As suggestion of the other referee, we have added 3 new tables to the revised version.
Table 2. Phase III Trials that support the actual SOC in ndGB patients. Pages 15-16.
Table 3: Selected Phase II/III Trials for Target Therapies in GB. Pages 23-28.
Table 4. Selected Phase II/III Trials on Immunotherapy for GB. Pages 41-44.
3. It is recommended to label Figure 1 and avoid relying solely on photographs.
Although you indication is not entirely clear to us, we have improved the Fig.’s content. Besides, and as suggested by referee 1, the resolution of all Figs. has been improved. Page 4.
4. The conclusion section requires substantial refinement. Currently, it primarily consists of background information on glioblastoma (GB) rather than addressing the stated objective of highlighting advancements, limitations, and future prospects in GB therapy, as outlined in the title "Glioblastoma Therapy: Past, Present, and Future."It is strongly recommended to revise the conclusion by emphasizing key advancements, limitations, and prospects in the realm of GB treatment.
We have changed the Conclusion section following your advice pages 70-71.
No other cancer has witnessed as many therapeutic attempts as GB. Unfortunately, although every treatment options in the Oncotherapy arsenal have been tried, GB almost always recurs. In fact, if you read Dr. Siddhartha Mukherjee’s book “The Emperor of All Maladies: A Biography of Cancer”, it is easy to think about GB as the emperor of all cancers.
Despite improvements in shorter-term survival rates, the 5-year survival rate after diagnosis remains relatively constant (only 5.8% of all patients). Key advancements, limitations, and perspectives can be summarized as follows:
- GB location, aggressivity and infiltrative biological behavior represent exceptional challenges. Surgeons face great difficulties to remove all cancerous cells without causing damage to critical brain regions. Thus, preserving brain function while treating the tumor becomes a delicate balance. Aggressive treatments might easily lead to cognitive impairments and/or neurological deficits.
- Standard treatment and targeted therapies have only modestly improved patient outcomes, a fact likely due to redundant compensatory mechanisms, insufficient target coverage (in part related to the BBB), and GSC ability to remain quiescent or upregulate their molecular defenses until cytotoxic drug concentrations decrease. Mechanisms favor treatment failure and tumor recurrence.
- Targeting of key contributors to GSC self-renewal, survival, and plasticity will contribute to avoid recurrences. ChemoID test (NCT03632135) uses CSCs isolated from tumor biopsies to select the most effective chemotherapeutic treatment for each patient [1122]. Strategies like this can assist in identifying the best treatment option and in properly designing recruitment for clinical trials.
- The BBB restrict the entry of many therapeutic agents into the brain, limiting the effectiveness of systemic treatments. Although the BBB and BTB can be disrupted in a variety of ways depending on the stage of the illness, its heterogeneous breakdown makes it quite difficult to achieve uniform drug concentrations inside the tumor. Strategies to improve drug delivery at bioefficacious concentrations include intranasal or intratumoral administrations (with limited acceptance in clinical practice), opening of the BBB with LIFU, and the use of nanocarriers. Especially interesting are multifunctional nanocarriers that allow the delivery of diverse molecules (chemotherapeutic and/or immunotherapeutic and/or imaging agents) and can also be used for theranostic purposes. In addition, receptor-based therapies have the potential to improve clinical outcomes. Nevertheless, most of this technology is, at this moment, at developing stages.
- While immunotherapies have revolutionized treatment for other cancers, in GB patients have not achieved similar positive responses. A fact associated with the unique challenges associated with the immunosuppressive TME. GB has an immune-privileged nature associated to cytotoxic T-cell exhaustion, limited lymphocytic infiltration, recruitment of pro-tumorigenic TAMs, downregulation of cancer cells’ MHC I complexes, and abundance of immune-inhibitory molecules such as IL-10 and TGFβ. In our opinion DC vaccination, oncolytic virotherapy, CAR-cell therapy or combined strategies (with ICIs) seem promising, and results would probably be enhanced by targeting antigens present in GCS. In any case, immunotherapy should be considered as part of the combined treatment needed to increase patient survival. The integration of immunotherapy with SOC for GB presents opportunities and challenges since radio/chemotherapy can substantially influence the immune response in two different ways. Radiotherapy can induce immunogenic cell death, thus facilitating the release of tumor antigens that could potentially amplify immunotherapy effectiveness. On the contrary, radio and chemotherapy can also induce lymphopenia and augment the expression of immunosuppressive molecules (PD-L1) within the TME, hence compromising the efficacy of immunotherapeutic treatments. The timing (adjuvant or neoadjuvant) for the best synergistic effect remains a problem to resolve.
- The implementation of multi-modal combined strategies is essential to increase survival and the quality of life of GB patients. The design of personalized treatments should be based on the genetic profiling and characteristics of this heterogeneous tumor, its specific location, the selection of the most promising targets, including the GSC, and the stimulation of the immune system. Personalized combined treatments offer advantages in terms of efficacy and reduced collateral damage, but the time needed to make decisions and cost represent significant challenges. The use of mathematical models of synergism/antagonism of drugs and pathways will help in predicting drug combinations for this multifactorial disease.
- We should reconsider what has not been done properly. To enter patients into a trial, the genetic characteristics of the tumor should be considered, and only include patients that present the corresponding therapeutic targets. Only in this way we will ensure that results become objective and significant. Administering immunotherapies concurrently with corticosteroids or other immunosuppressive treatments is nonsensical. Such combinations must be considered when analyzing clinical study results to prevent misleading conclusions.
- LITT, TRT, HIFU, PDT and SDT can be an alternative (particularly in patients with poor physical condition), a complement to radiotherapy, or become part of the SOC.
5. In Section 7, dedicated to "Non-Ionizing Energies in GB Therapy," it would be valuable for the authors to incorporate a discussion on a recent study (https://doi.org/10.3389/fcell.2023.1067861) that explores the utilization of nanosecond pulsed high-power microwave irradiation to induce apoptosis in brain cancer cells.
Since points 5 and 6 refer to the same topic, we will address them jointly in point
6. Microwaves are recognized to exert notable effects on the human brain, with outcomes being either beneficial or detrimental depending upon factors such as microwave type, frequency, dosage, and power. It is important to discuss these details within the review article, underscoring the potential for achieving favorable effects or mitigating harm through the optimization of these parameters. Hence, author should add some discussion on microwave radiations and brain in section 7 "Non-Ionizing Energies in GB Therapy.”
Thank you for bringing this work to our attention. Indeed, it is a very interesting strategy. We have changed the title of the section 7.4. Pulsed Electric Fields to 7.4. Microwaves and Pulsed Electric Fields, and have incorporated the following information (see pages 68-67):
“Microwaves are a form of electromagnetic radiation with wavelengths ranging from about 1 mm to 30 cm, corresponding to frequencies between 1 GHz and 300 GHz, respectively [1073]. Microwaves have been introduced in medicine for cancer diagnosis [1074] and treatment [1075,1076]. However, microwaves have well known adverse effects on the CNS, and can affect neurotransmitters release and, thereby, cause a delay in the signaling process [1077]. Despite these limitations, the research of Rana et al. is an excellent example of how microwave radiation can therapeutically be used to treat GBs [1078]. These authors have shown that a strong electric field (~23 kV/cm) of pulsed high-power microwave (HPM) irradiation causes ROS generation, DNA damage, p53 activation and death in exposed GB U87 cells. Importantly, these authors show that pulse dosage causing damage to GB cells and brain normal cells is different, thus representing a new therapeutic approach that deserves to be tested rapidly in in vivo models. In fact, pulsed microwave-induced thermoacoustic therapy has been recently proposed as a potential alternative modality to precisely and effectively eradicate orthotopic GB. Interestingly, a NP composed of polar amino acids and adenosine-based agonists has been developed having a high microwave absorbance and selective penetration of the BBB. The NP activates the adenosine receptor on the BBB to allow self-passage, and once it is accumulated in the TME the NP converts absorbed microwaves into ultrasonic shockwaves, which can mechanically destroy tumor cells with minimal damage to adjacent normal brain tissue due to the rapid decay of the ultrasonic shockwave intensity [1079]”.
7. Incorporating a discussion on the potential use of natural products or flavonoids for the treatment of brain tumors would be beneficial for the manuscript. Addressing relevant studies and findings in this area could provide additional insights into alternative therapeutic approaches.
We have introduced a new section, entitled: 4.6. Other assayed strategies, in which we have included the following information regarding your suggestion about polyphenols (pages 38-39).
“Some natural polyphenols (resveratrol, curcumin, silibinin, quercetin, etc.) have evidenced antiGB potential and synergize with radio/chemotherapy in vitro. Inhibition of proliferation, migration, cell invasion and angiogenesis are mechanisms proposed to explain a potential effect of polyphenols on reducing GB progression [650–655]. Specifically, metformin and resveratrol have been suggested to inhibit GB cell proliferation, invasion and migration by downregulating the PI3K/Akt pathway, activating mTOR, and increasing AMPK phosphorylation [656]. Rutin, epigallocatechin-3-gallate, quercetin and curcumin have shown synergistic effects with TMZ; whereas curcumin enhances the action of etoposide, paclitaxel, cisplatin, camptothecin, and doxorubicin [653,657–659]. Nevertheless, most of these proposed mechanisms only have in vitro support. It should be pointed that, under in vivo conditions, the antitumoral activity of polyphenols is limited due to their short half-life. Even when polyphenols are administered at high doses, their rapid metabolism precludes to reach efficacious anti-tumor concentrations in a growing cancer. In practice, after an oral dose as high as 100 mg/kg, plasma peak level of resveratrol was of 11 ± 4 μM at 10 min, pterostilbene was of 25 ± 6 μM at 10 min, EGCG was of 5 ± 2 μM at 30 min, curcumin was of 27 ± 6 μM at 15 min, quercetine was of 8 ± 2 μM at 15 min, and genistein was of 17 ± 5 μM at 10 min. Plasma levels of all these PFs were <1 μM at 60 min [660]. Exposure of many different cancer cells (i.e. U87 and LN229 GB cells) to these concentrations for just 1h does not affect their growth and viability. Our results are supported by Beylerli et al. that recently point out that, although potential beneficial effects exerted by polyphenols are promising, their efficacy in vivo is strongly limited by their bioavailability and BBB permeability [658]. Zanotto-Filho et al. showed that resveratrol, a BBB permeable drug, improved TMZ/curcumin efficacy in brain-implanted tumors in rats [659]. In these experiments 50 mg curcumin/kg x day and 10 mg resveratrol/kg x day were administered i.p., but it is uncertain at which concentration each polyphenol reached the tumor and for how long. Based on the above discussion, it is easy to deduce that the tumor levels of in vivo administered polyphenols will be very low and acting for a short period of time. Facts that raise many doubts about the real mechanisms involved in the anti-tumor effects. The effect of curcumin has also been assayed using its intratumoral injection [661]. In this case, curcumin (100 mg/kg) inhibited U87 xenografts growth by approx. 50%. However, all the mechanisms proposed for curcumin were studied under in vitro conditions and using unreliable conditions compared to the in vivo setting. Methods to improve polyphenol absorption, pharmacokinetics, and efficacy [662–664] are key in order to deliver therapeutic concentrations at specific target sites, and to increase the antitumoral bioefficacy of polyphenols in vivo.”.
8. The manuscript contains several typos and grammatical errors. I recommend thorough proofreading to identify and correct these issues.
The manuscript has been fully revised in order to correct typos and grammatical errors.
Thanks again for your comments.
Round 2
Reviewer 1 Report
Comments and Suggestions for Authors
The authors have answered all my questions aside from that: "Frst that is a very impressive review that gathers all the possible data about glioblastoma therapy. However, it is over 90 pages long and over 1000 references, which is more than most books in that area. Maybe try to publish it as a book instead?
"
Reviewer 2 Report
Comments and Suggestions for Authors
The authors have addressed all of my comments and concerns in the revised version. I recommend to accept the paper in its present form.